# Early detection and staging of chronic liver diseases with a protein MRI contrast agent

Mani Salarian[1], Ravi Chakra Turaga [2], Shenghui Xue [1], Maysam Nezafati[3], Khan Hekmatyar[4], Jingjuan Qiao[1], Yinwei Zhang[2], Shanshan Tan[1], Oluwatosin Y. Ibhagui[1], Yan Hai[5], Jibiao Li[6], Rao Mukkavilli[7], Malvika Sharma[2], Pardeep Mittal[8], Xiaoyi Min[5], Shella Keilholz[3], Liqing Yu[6], Gengshen Qin[5], Alton Brad Farris [9], Zhi-Ren Liu[2,7] & Jenny J. Yang[1,7]

Early diagnosis and noninvasive detection of liver fibrosis and its heterogeneity remain as major unmet medical needs for stopping further disease progression toward severe clinical consequences. Here we report a collagen type I targeting protein-based contrast agent (ProCA32.collagen1) with strong collagen I affinity. ProCA32.collagen1 possesses high relaxivities per particle ($r_1$ and $r_2$) at both 1.4 and 7.0 T, which enables the robust detection of early-stage (Ishak stage 3 of 6) liver fibrosis and nonalcoholic steatohepatitis (Ishak stage 1 of 6 or 1A Mild) in animal models via dual contrast modes. ProCA32.collagen1 also demonstrates vasculature changes associated with intrahepatic angiogenesis and portal hypertension during late-stage fibrosis, and heterogeneity via serial molecular imaging. ProCA32.collagen1 mitigates metal toxicity due to lower dosage and strong resistance to transmetallation and unprecedented metal selectivity for $Gd^{3+}$ over physiological metal ions with strong translational potential in facilitating effective treatment to halt further chronic liver disease progression.

[1] Department of Chemistry, Georgia State University, Atlanta, GA 30303, USA. [2] Department of Biology, Georgia State University, Atlanta, GA 30303, USA. [3] Department of Biomedical Engineering, Emory University and Georgia Institute of Technology, Atlanta, GA 30322, USA. [4] Bioimaging Research Center, University of Georgia, Athens, GA 30602, USA. [5] Department of Mathematics and Statistics, Georgia State University, Atlanta, GA 30303, USA. [6] Center for Molecular and Translational Medicine, Georgia State University, Atlanta, GA 30303, USA. [7] Center for Diagnostics and Therapeutics, Georgia State University, Atlanta, GA 30303, USA. [8] Medical College of Georgia, Augusta University, Augusta 30912, Georgia. [9] Department of Pathology and Laboratory Medicine, Emory University School of Medicine, Atlanta, GA 30307, USA. Correspondence and requests for materials should be addressed to J.J.Y. (email: jenny@gsu.edu)

Nonalcoholic fatty liver disease (NAFLD) and alcoholic liver disease (ALD) are common causes of chronic liver disease (CLD), cirrhosis, and hepatocellular carcinoma (HCC), which are major causes of morbidity and mortality worldwide[1–5]. CLD originates from a variety of causes such as viral hepatitis, metabolic dysfunction, as well as alcohol abuse and autoimmune disease, and almost all chronic liver injuries cause liver fibrosis[6,7]. Nonalcoholic steatohepatitis (NASH) and hepatic fibrosis can develop in patients with any type of CLD, including alcoholic liver disease, hepatitis C, hepatitis B, NAFLD and autoimmune hepatitis. The Centers for Disease Control and Prevention (CDC) previously reported that 19,388 people died in the US from alcohol related liver diseases in 2014[6]. Moreover, from 2000 to 2015, death rates for chronic liver disease and cirrhosis in the United States increased 31% among persons aged 45–64 years[8]. Nonalcoholic fatty liver disease including NASH, is the primary cause of CLD in the United States, afflicting an estimated 80–100 million Americans (30–40% of the population). NAFLD and ALD can further progress to cirrhosis or hepatocellular carcinoma[5]. It has been projected that NASH occurs in 20% of patients with NAFLD (3–12% of the US population), and 30–40% of them with NASH will develop fibrosis[5,9,10].

Biopsy is the gold standard for diagnosis and staging of CLD as it relies on stage-dependent characteristic patterns of collagen expression. However, it has many limitations such as sampling errors, and high interobserver variability with 33–50% error rates even for diagnosis of advanced stages of liver fibrosis such as cirrhosis likely due to heterogeneity[2]. Earlier studies have reported that imaging modalities can detect morphological characteristics of liver cirrhosis such as surface nodularity; however, in some cases, the cirrhotic liver can appear completely normal, and biopsy is required for confirmation of the diagnosis[11,12].

Tremendous effort has been devoted to the development of noninvasive imaging techniques associated with fibrosis and NASH, such as ultrasound; MR apparent diffusion coefficient (ADC); magnetic resonance elastography (MRE); T1, T2, relaxation in the rotating frame ($T_{1\rho}$)[13]; magnetization transfer[14]; and proton density fat fraction. However, these techniques cannot provide accurate detection of early stage liver fibrosis and NASH.

A prerequisite for early detection and efficient treatment of liver fibrosis and NASH is a reliable noninvasive diagnostic method to accurately stage NASH and fibrosis progression and response to therapy. Magnetic resonance imaging (MRI) offers several unique advantages compared to other clinical imaging modalities with its deep tissue penetration, high spatial resolution, and coverage of the entire liver. MRI does not require the use of ionizing radiation, and it is well suited for detection and monitoring the progression and regression of NASH and fibrosis[15,16]. However, MRI cannot unambiguously detect early stage patient liver fibrosis and NASH with regional heterogeneity due to major limitations such as lack of sensitive MRI contrast agents.

To differentiate invisible fibrotic cells from heterogeneous tissue background lacking a clear boundary requires novel MRI contrast agents and imaging methodology to have a strong specificity for both organ and diseased cells as well as improved relaxation properties. Iron oxide-based contrast agents such as ferumoxides create negative (dark) $T_2/T_2^{\star}$ effects that reportedly yield images with signal voids and very limited accuracy[17,18]. The majority of clinically approved $Gd^{3+}$ MRI contrast agents have $r_1$ and $r_2$ relaxivity values, but only $r_1$ is strong enough to lead to a positive (bright) imaging[19]. To date, two clinical agents Gd-BOPTA (MultiHance) and Gd-EOB-DTPA (Eovist, US; Primovist, Europe) exhibit liver distribution and are able to detect late-stage fibrosis (cirrhosis) in vivo using T1-weighted sequence at a short time point[20,21]. However, there is an unmet medical need to quantify the severity or the stage of fibrosis when making decisions regarding diagnoses, and prognoses of CLDs.

The recent development of collagen-targeted contrast agents based on approved Gd-DTPA (Magnevist) appears to image liver fibrosis with improved sensitivity and specificity but these results depend on the animal models evaluated[3], while safety concerns about Magnevist have prevented the efforts to develop a safe MRI contrast agent capable of detecting early stage fibrosis, NASH and delineating disease heterogeneity[22]. To address this, we reasoned that precision T1 and T2 imaging capability ($r_1$ and $r_2$) of stage-dependent characteristic expression patterns of collagen type I would lead to the detection of early stage of the disease and its heterogeneity associated with late-stage liver fibrosis.

In this study, we report the development of a protein MRI contrast agent (ProCA) based on rat α-parvalbumin expressed in *Escherichia coli* for molecular imaging of collagen I levels (ProCA32.collagen1) in three models of CLDs[23]. ProCA32.collagen1 demonstrates high dual relaxivity values for $r_1$ and $r_2$ per $Gd^{3+}$ at both 1.4 and 7.0 T, as well as a strong collagen-targeting capability. ProCA32.collagen1 enables the first robust detection of early and late-stage liver fibrosis and early stage NASH in addition to heterogeneous expression of collagen by multiple imaging techniques. Furthermore, it provides the first demonstration of possible vascular and architectural alterations caused by intrahepatic angiogenic and portal hypertension during late-stage fibrosis.

## Results

**Design of ProCA32.collagen1**. Collagen-targeted protein MRI contrast agent, ProCA32.collagen1 was designed by engineering a collagen type I targeting peptide moiety to the C-terminal of protein contrast agent, ProCA32 (with two $Gd^{3+}$ binding sites) (Fig. 1a). In the modeled structure, lysine residues are positioned as anchor points for polyethylene glycol (PEG) modification. A flexible hinge was used to maximize targeting capacity and relaxivity, while maintaining metal binding capability[23]. PEGylation was used to improve relaxivities by tuning correlation time. It also increases stability and blood retention time[24]. The designed ProCA32.collagen1 was bacterially expressed, purified, modified, and formulated (Supplementary Fig. 1). Inductively coupled plasma optical emission spectrometry (ICP-OES) analysis of the complex indicated the formation a 2:1 Gd-ProCA32.collagen1 complex[23–25].

Collagen type I binding affinity of ProCA32.collagen1 was determined using indirect enzyme-linked immunosorbent assay (ELISA) (Fig. 1b). The contrast agent exhibited high affinity to collagen type I with a dissociation constant of $K_d = 1.42 \pm 0.2\ \mu M$ and a 1:1 binding model[26,27]. ProCA32 without the collagen I targeting moiety did not show any specific binding to collagen I.

The determined relaxivity values of $r_1$ and $r_2$ for ProCA32.collagen1 were $34.0 \pm 0.12\ mM^{-1}\ s^{-1}$ and $50.0 \pm 0.16\ mM^{-1}\ s^{-1}$ per $Gd^{3+}$, respectively at 37 °C and 1.4 T, or $68.0 \pm 0.25\ mM^{-1}\ s^{-1}$ and $100.0 \pm 0.32\ mM^{-1} s^{-1}$ per particle, respectively (Table 1, Supplementary Fig. 2a, b, c). ProCA32.collagen1 exhibited the highest $r_1$ ($21.3 \pm 0.5\ mM^{-1}\ s^{-1}$ per $Gd^{3+}$ or $42.6 \pm 1\ mM^{-1}\ s^{-1}$ per participle) and $r_2$ ($108.5 \pm 1.2\ mM^{-1}\ s^{-1}$ per $Gd^{3+}$ or $217.0 \pm 2.4\ mM^{-1}\ s^{-1}$, per particle) compared to clinical contrast agents, at higher magnetic field strength of 7.0 T and 37 °C[28,29]. ProCA32.collagen1's $r_1$ and $r_2$ relaxivity values are 5–10 and 12–17 times greater than those of clinical contrast agents at 1.4 and 7.0 T, respectively. Therefore, this targeted contrast agent can be applied to both low- and high-magnetic field strengths with both T1- and T2-weighted molecular imaging. The water number $q$ in the first coordination shell determined by luminescence resonance energy transfer (LRET) using $Tb^{3+}$ luminescence life time decay, was 0.5 (Supplementary Fig. 2d and e).

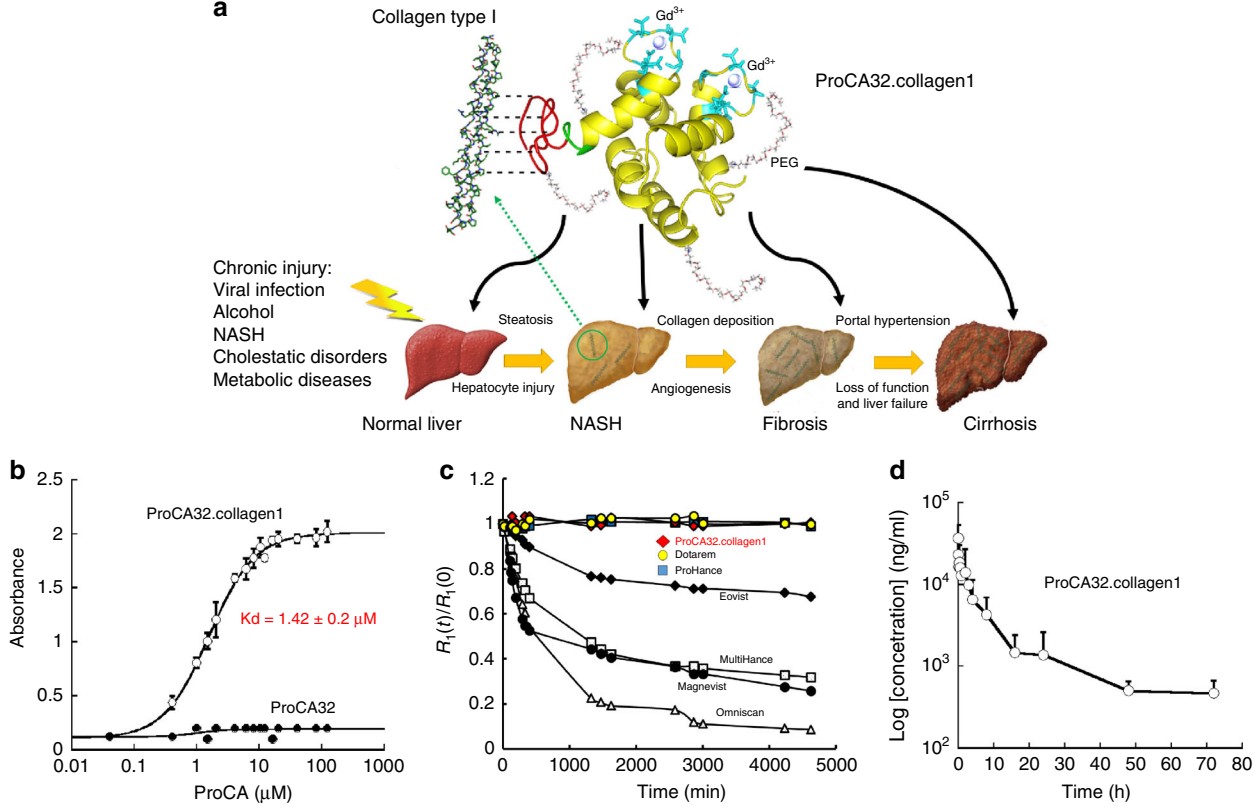

**Fig. 1** Development of ProCA32.collagen1 and its biophysical characteristics. **a** The model structure and development of ProCA32.collagen1 by engineering collagen type I targeting moiety (GGGKKWHCYTYFPHHYCVYG, red) at C-terminal of ProCA32 using a flexible hinge (green) and PEGylation. **b** Measurement of the dissociation constant of ProCA32.collagen1 to collagen type I using indirect ELISA assay. No collagen I binding was observed for ProCA32 (PEGylated, nontargeted agent). **c** The relaxation rates change of clinical contrast agents (black diamond Eovist; black circle Magnevist; white square MultiHance; white triangle Omniscan; blue square ProHance; yellow circle Dotarem) and red lozenge ProCA32.collagen1 in the presence of ZnCl$_2$ at different time points up to 4 days. **d** Pharmacokinetic of ProCA32.collagen1. Gd$^{3+}$ concentration in serum collected after injection of ProCA32.collagen1 was determined by ICP-OES. Error bars indicate standard deviations of six separate measurements in $n = 6$, biologically independent animals

**Table 1 Summary of $r_1$ and $r_2$ relaxivity of ProCA32.collagen1 and non-PEGylated ProCA32.collagen1 and their comparison with ProCA32 and clinical contrast agents at 1.4 and 7.0 T, 37 °C**

| Contrast agent[29,30] | $r_1$ at 1.4 T | $r_1$ at 7.0 T | $r_2$ at 1.4 T | $r_2$ at 7.0 T |
|---|---|---|---|---|
| Magnevist (Gd-DTPA) | 3.3 | 5.1 | 3.9 | 9.4 ± 1.3 |
| Eovist (EOB-DTPA) | 5.38 ± 0.02 | 5.37 | 6.54 ± 0.06 | 7.01 |
| Dotarem (DOTA) | 3.9 ± 0.2 | N/A | 3.2 ± 0.7 | N/A |
| MultiHance (BOPTA) | 6.20 | N/A | 8.7 | N/A |
| ProHance (HP-DO3A) | 4.39 | N/A | 5.0 | N/A |
| ProCA32 | 33.14 ± 0.32 | 18.9 | 44.61 ± 0.12 | 48.6 ± 0.1 |
| Non-PEGylated ProCA32.collagen1 | 30 ± 0.06 | N/A | 42 ± 0.1 | N/A |
| ProCA32.collagen1 | 34 ± 0.12 | 21.3 ± 0.5 | 50 ± 0.16 | 108.5 ± 1.2 |

$r_1$ and $r_2$ values are reported per Gd$^{3+}$

We examined metal binding affinity and selectivity of ProCA32.collagen1. The Gd$^{3+}$ binding affinity of ProCA32.collagen1 was calculated to be $2.0 ± 0.25 × 10^{-22}$ M, which is comparable to the approved clinical contrast agents (Table 2 and Supplementary Fig. 3). Since competition of physiological metal ions such as Zn$^{2+}$ and Ca$^{2+}$[30,31] is considered to be one of the most important risk factors for Gd$^{3+}$ release in vivo, we then determined metal binding constants for these metal ions[23,32]. As shown in Table 2, ProCA32.collagen1 exhibited $10^8$–$10^{16}$-fold greater metal selectivity (kinetic stability) for Gd$^{3+}$ over Ca$^{2+}$ and Zn$^{2+}$ compared with clinically approved contrast agents such as Dotarem.

Transmetallation studies demonstrated that the relaxation rates of clinical contrast agents gradually decreased upon incubation with 2.5 mM zinc[23] (Fig. 1c). In contrast, the relaxation rates of ProCA32.collagen1 remained unchanged suggesting that ProCA32.collagen1 resists transmetallation. Further, ProCA32.collagen1 remained intact without cleavage upon incubation with human serum at 37 °C for 13 days (Supplementary Fig. 4b). Pharmacokinetic properties of the contrast agent were also assessed (Fig. 1d). Moreover, ProCA32.collagen1 clinical chemistry test values were within the normal range, demonstrating normal function of organs including kidney and liver (Supplementary Fig. 5). No tissue damage or major Gd$^{3+}$ accumulation was observed (Supplementary Fig. 6).

**Table 2 Comparison of metal binding affinities (Gd$^{3+}$, Tb$^{3+}$, Zn$^{2+}$, and Ca$^{2+}$) and selectivity of ProCA32.collagen1 with clinical contrast agents**

| Contrast agent[29,30] | Log ($K_{Tb}$) | Log ($K_{Gd}$) | Log ($K_{Ca}$) | Log ($K_{Zn}$) | Log ($K_{Gd}/K_{Ca}$) | Log ($K_{Gd}/K_{Zn}$) |
|---|---|---|---|---|---|---|
| Magnevist (Gd-DTPA) | 22 | 22.46 | 10.75 | 18.6 | 12.24 | 4.13 |
| Eovist (EOB-DTPA)[a] | N/A | 23.6 | 11.82 | 18.78 | 12.22 | 5.18 |
| Dotarem (DOTA) | N/A | 24.7 | 17.23 | 21.05 | 7.46 | 3.65 |
| MultiHance (BOPTA)[a] | N/A | 21.91 | N/A | 17.04 | N/A | 4.87 |
| ProHance (HP-DO3A) | N/A | 23.8 | 14.83 | 19.37 | 10.07 | 4.37 |
| ProCA32[a] | 21.08 | 22.44 | 9.55 | 8.77 | 13.1 | 14.3 |
| Non PEGylated ProCA32.collagen1 | 22.53 | 22.54 | 8.71 | 6.36 | 14.7 | 16.2 |
| ProCA32.collagen1[a] | 22.80 | 22.30 | 8.70 | 6.12 | 14.4 | 16.2 |

[a]Liver-specific contrast agents

**Detection of early and late stages liver fibrosis and NASH**. We next evaluated whether molecular MRI with ProCA32.collagen1 could unambiguously and quantitatively detect both early and late stage liver fibrosis and NASH in vivo in different mouse models, taking advantage of the high $r_1$ and $r_2$ values (Figs. 2 and 3, Supplementary Fig. 8). Figure 2a shows R1 map MRI images collected pre, and 24 h after, intravenous (I.V.) injection of contrast agents (100 μL, 5 mM, 0.02 mmol/kg) of ProCA32.collagen1, ProCA32 (nontargeted), and Eovist in TAA/Alcohol-induced liver fibrosis mouse model. After 24 h post injection, ProCA32.collagen1 demonstrated an increase in R1 to 2.4 (ΔR1 ~0.78 s$^{-1}$, an approximate 50% increase in the liver area) in early stage fibrotic liver. This change is nearly doubled with ΔR1 enhancement to 1.40 s$^{-1}$ (a 78% increase) compared to preinjection for late-stage fibrotic liver. The R1 increase at 24 h post injection was highly specific, and gradually decreased after 48 h due to excretion. In contrast, the targeting agent did not result in a significant change in normal liver (ΔR1 < 0.2 s$^{-1}$ and 10% increase in liver area) at 24 h time point. Consistent with the R1 map, T2 map results also exhibited the same pattern at 24 h for both early and late stages of fibrosis, with higher changes for the late-stage fibrotic liver (Fig. 2b, Supplementary Fig. 8a).

ProCA32.collagen1 is also capable of detecting early-stage NASH in nonalcoholic fatty liver in NASH diet-induced mouse model using both R1 and T2 maps (Fig. 3a, b, Supplementary Fig. 8c). At 24 h post-injection of ProCA32.collagen1, liver with NASH exhibited ΔR1 of 1.02 s$^{-1}$ which is 4-fold higher than ΔR1 of 0.25 s$^{-1}$ for normal liver (Fig. 3a). Consistently, ΔR2 determined by T2 map is 11.1 s$^{-1}$ instead of 3.8 s$^{-1}$ of normal liver at the same time point (Fig. 3b, Supplementary Fig. 8c). In contrast, Eovist did not demonstrate significant enhancement in R1 map. In addition, ProCA32 without collagen binding capability did not result in any significant enhancement for R1 map changes at this time point (Fig. 2a).

**Histological validation, and biodistribution**. Detailed histological analysis and hydroxyproline content (Supplementary Fig. 16) confirmed early stage liver fibrosis (Ishak stage 3 of 6) and early stage NASH (Mild-1A or Ishak stage 1 of 6) based on the Ishak and NASH Clinical Research Network (NASH CRN) scoring systems with the help of a pathologist. Late-stage fibrosis (Ishak stage 5 of 6) was also confirmed by histology analysis for all animal models (Figs. 2f, 3f, g). Late-stage liver fibrosis (Ishak stage 5 of 6) induced by diethylnitrosamine (DEN) was also confirmed by histology analysis. Furthermore, ProCA32.collagen1 distribution in normal and fibrotic liver was well-correlated with the stage of the disease (Figs. 2d, e and 3d, e). During the late stage of liver fibrosis in both NASH diet and TAA/alcoholic models, αSMA levels (brown) were much higher compared to early-stage fibrosis, NASH and normal liver (Figs. 2f and 3f). Sirius red, and H&E

staining of early stage NASH compared to normal liver demonstrate the presence of microvesicular and macrovesicular steatosis and collagen, as indications of the disease stage (Fig. 3g).

We have also reported the application of serial molecular imaging (SMI) over time of a targeted contrast agent to reveal additional differences between early and late stages of fibrosis and NASH. The serial change of R1 values can be seen in Fig. 4b, d, where all R1 values in both early stage of the disease and normal liver increased to 2.0–2.4 s$^{-1}$, 3 h post injection of ProCA32.collagen1. However, at 24 h post injection, R1 values of normal liver decreased but fibrotic and livers with NASH showed an increase to 2.4 and 2.8 s$^{-1}$, respectively. These values decreased for both livers as the contrast agent was washed out of the liver 48 h post injection. However, at 3 h post injection of ProCA32.collagen1, R1 of late-stage fibrotic livers for both animal models increased to >3.5–4.1 s$^{-1}$ with ΔR1 ≥ 2.0 s$^{-1}$ (Fig. 4a, c). The Increase of R1 at 3 h post-injection for late-stage fibrosis likely suggests the existence of intrahepatic angiogenesis. Histogram analysis further confirms the large difference between early-stage and late-stage liver fibrosis in TAA/Alcohol model at different time points (Supplementary Fig. 14). We have observed significantly increased vessel formation stained by CD31 with quantitative analysis supporting the vessel changes due to intrahepatic angiogenesis in late-stage fibrosis (Fig. 4g).

We further reported the existence of portal hypertension associated with late-stage fibrosis in mouse liver which may justify the slow washout of ProCA32.collagen1 from the liver (Fig. 4e, f, Fig. 6c, d).

**Quantitative mapping of collagen heterogeneity**. Detection of liver fibrosis heterogeneity can be regarded as one of the major limitations of liver biopsy and other imaging modalities[33–36]. Collagen heterogeneity induced by DEN was observed with ProCA32.collagen1-enabled SMI (Fig. 5). The DEN-induced mouse model exhibited strong collagen heterogeneity mimicking patient cirrhosis[11,12]. Figure 5b indicated that the right area of the mouse (defined by A2: Area 2) liver has a collagen proportionate area (CPA) of 12% (Sirius red staining) while the left area (A1) has a CPA of 6.7%. All T1-, T2-weighted, and T1-inversion recovery images exhibited time-dependent liver enhancement at 3 and 24 h post injection of ProCA32.collagen1 with the maximum enhancement at 3 h post injection, specifically in A2, confirming the histology data (Fig. 5a, b). R1 maps and its corresponding histogram analysis further confirm collagen heterogeneity (Supplementary Fig. 15).

In order to demonstrate the time-dependent collagen heterogeneity detection by ProCA32.collagen1 and quantitatively assess the extent of enhancement, a color matrix was created (Fig. 5d, e). The first column shows the enhancement 3 h post injection compared to Prescan (called 3 h). The second column shows the

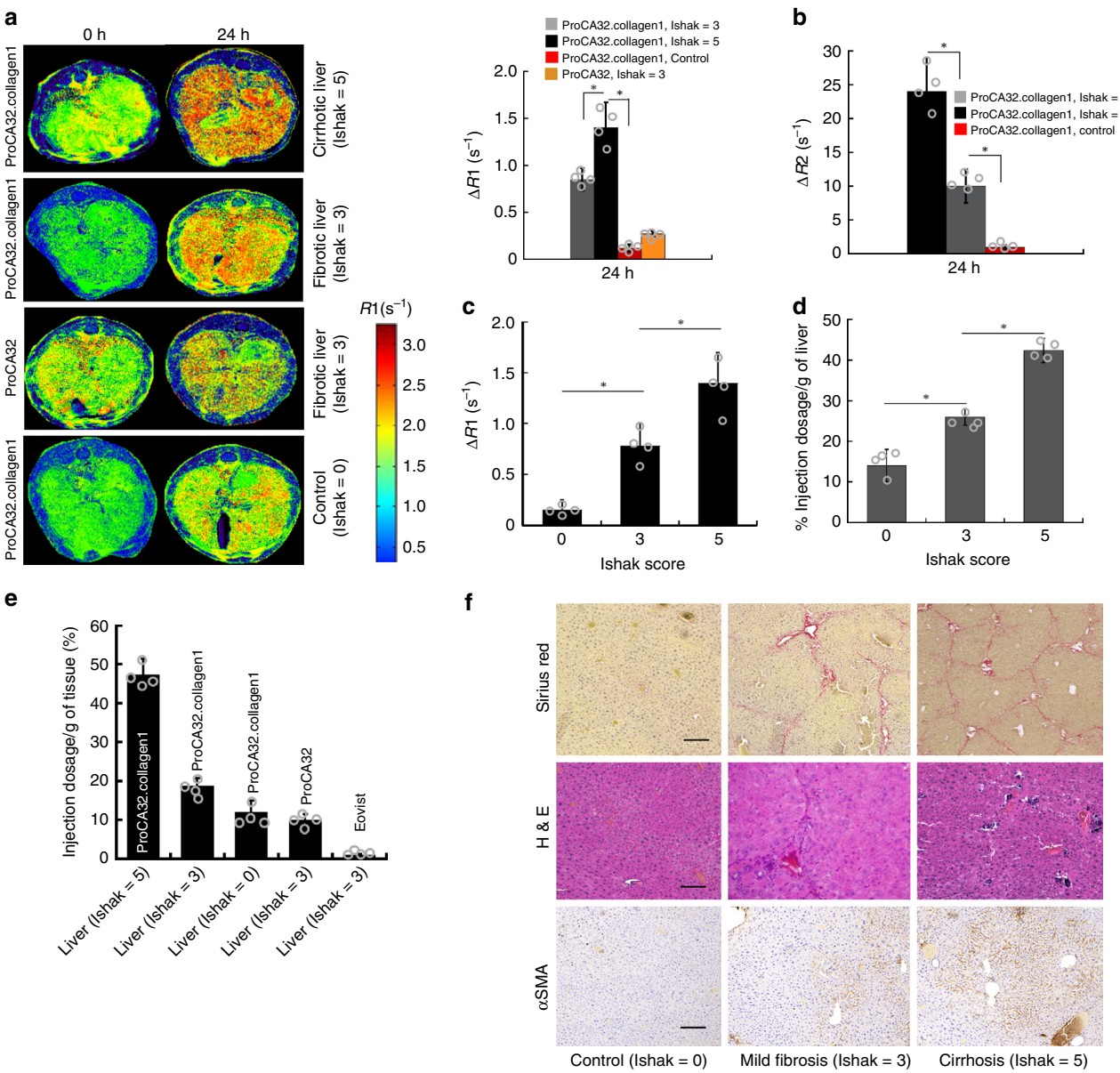

**Fig. 2** Early stage detection of liver fibrosis in TAA/Alcohol model. **a** R1 maps and ΔR1 values of normal (Ishak stage 0 of 6), early-stage (Ishak stage 3 of 6), and late-stage (Ishak stage 5 of 6) liver fibrosis before and 24 h after injection of ProCA32.collagen1, and ProCA32. **b** ΔR2 values derived from T2 maps demonstrating the highest enhancement for late-stage liver fibrosis at 24 h post injection. **c** ProCA32.collagen1 can distinguish early-stage liver fibrosis from normal and late-stage fibrotic liver with ΔR1 derived from a R1 map at 24 h time point. **d** Correlation between $Gd^{3+}$ concentration in liver with Ishak score at 24 h time point. **e** Percent injection dosage of ProCA32.collagen1 based on [$Gd^{3+}$] in fibrotic and normal livers at 24 h time point. **f** Sirius red, H&E and αSMA staining of early- and late-stage fibrotic liver compared to normal liver confirms the stage of liver fibrosis. scale bar, 100 μm, *$P < 0.05$, student's $t$ test, all data are represented as mean ± SD, $n = 4$ biologically independent animals in each group

enhancement 24 h post injection compared to Prescan (called 24 h) and third and fourth columns represent the enhancement 24 h post injection compared to 3 h. If the voxels remained enhanced it was called *Maintained* and if the voxels had decrease in intensity, they were called *Washout*. The matrix shows a quantitative voxel analysis of MRI images by exhibiting the number of voxels and their percentage of increase in intensity at each time point post injection of ProCA32.collagen1 compared to preinjection which is an indication of the dynamic property and targeting capability of ProCA32.collagen1 based on collagen distribution differences in the liver. In this analysis, the short-T1 inversion recovery with long TE demonstrated the highest sensitivity in terms of number of voxels with ~100% increase in intensity for detecting collagen heterogeneity among other pulse sequences.

**Disease staging with ProCA32.collagen1**. In order to demonstrate different uptake and washout rates of ProCA32.collagen1 based on the stage of the disease and post injection time point, percentage of R1 and R2 enhancement curves were plotted based on R1 and R2 values from MR mapping. To assess the uptake, time point before injection was used as a reference to calculate an area under the curve (AUC) in the time window 0–48 h (AUC_0–48) for the entire liver in each MRI slice in NASH-diet model (Fig. 6a, b). To assess the washout, 24 h time point was selected, and the corresponding AUC was calculated (Fig. 6c, d).

AUCs calculated from ΔR1 and ΔR2 were represented as boxplots for better demonstration of ProCA32.collagen1 ability to distinguish different stages of disease in NASH diet model (Fig. 6e). The difference in AUC_3–48 for normal vs. early, early vs. late and

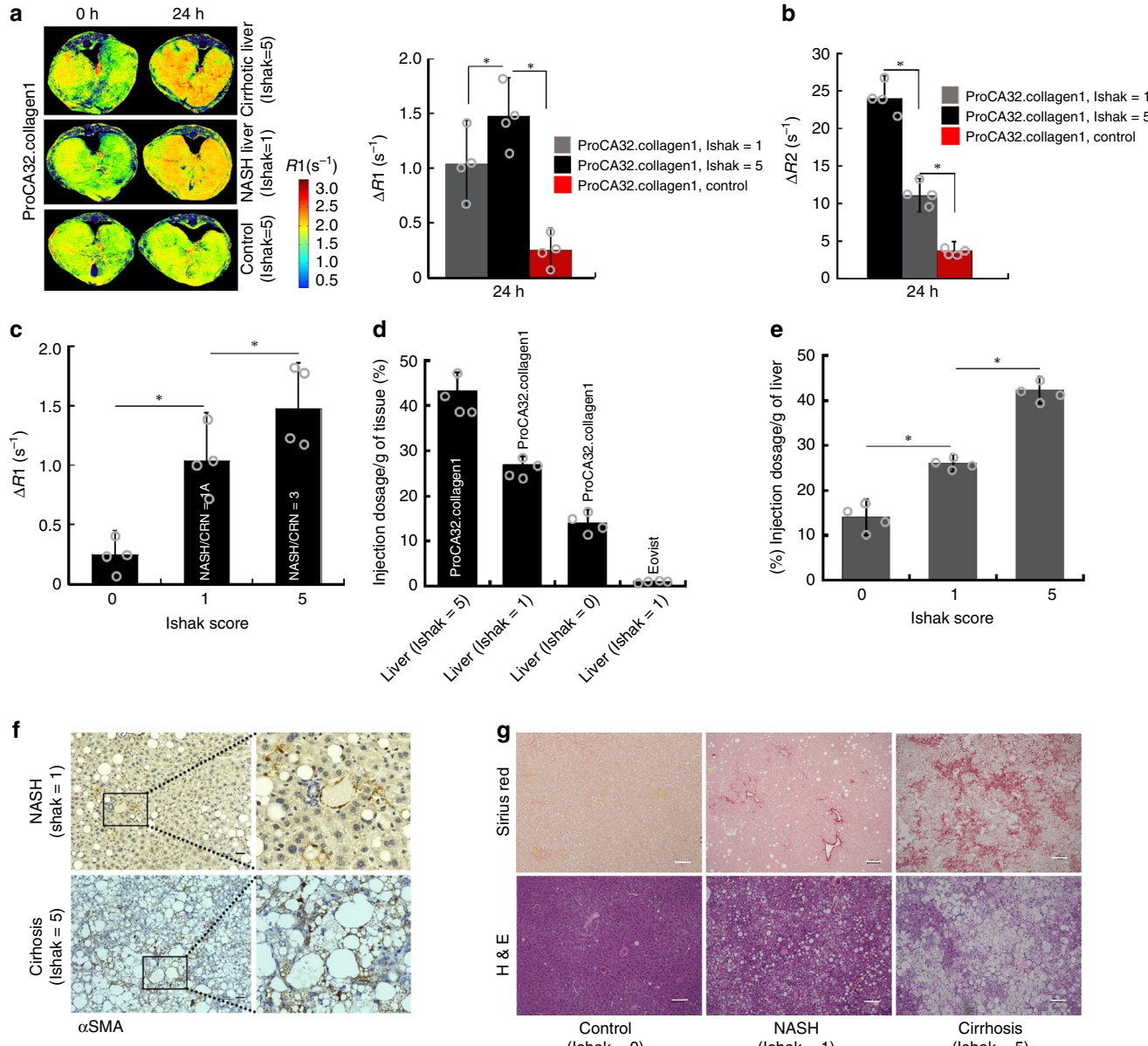

**Fig. 3** Early stage detection of NASH in NASH diet model. **a** R1 maps of NASH liver (Ishak stage 1 of 6 or Mild-1A in NASH CRN system), late-stage liver fibrosis (Ishak stage 5 of 6 or 3 in NASH CRN system), and normal liver (Ishak stage 0 of 6) before and 24 h after injection of ProCA32.collagen1. **b** $\Delta$R2 values derived from T2 map are consistent with $\Delta$R1 values showing the highest enhancement for late-stage liver fibrosis. **c** ProCA32.collagen1 can distinguish early-stage NASH from normal and late-stage fibrotic liver with $\Delta$R1 at 24 h time point. **d** Correlation between $Gd^{3+}$ concentration in liver with Ishak scores at 24 h time point. **e** Percent injection dosage of ProCA32.collagen1 based on $[Gd^{3+}]$ in NASH, late-stage fibrotic and normal liver at 24 h time point. **f** $\alpha$SMA staining of early-stage NASH and late-stage fibrosis demonstrates the degree of steatosis which is much higher in late-stage fibrosis (scale bar, 50 µm). **g** Sirius red, and H&E staining of early-stage NASH compared to normal liver demonstrate the presence of steatosis and collagen, as indications of the disease stage. scale bar, 100 µm; *$P < 0.05$, unpaired two-tailed student's $t$ test; all data are represented as mean ± SD, $n = 4$ biologically independent animals in each group

normal vs. late were all statistically significant ($p < 0.01$). In addition, scatter-plots (Fig. 6f) showed that AUC_0–48 are positively correlated with CPA measured from histology (Supplementary Fig. 9b) showing the potential of ProCA32.collagen1 in staging NASH and late-stage fibrosis (Fig. 6f) as a noninvasive technique. The Pearson correlations between LogitCPA and AUC_0–48 were 0.88 ($p$ value = 0.00016) and 0.89 ($p$ value = 0.00013) for $\Delta$R1 and $\Delta$R2, respectively. ProCA32.collagen1 was also capable of distinguishing normal liver from early- and late-stage fibrosis in TAA/alcohol model (Supplementary Fig. 12) based on receiver operating characteristic (ROC) analysis.

## Discussion

ProCA32.collagen1 exhibits higher relaxivity properties for both $r_1$ and $r_2$ compared to ProCA32 (Table 1). These increases of $r_1$ and $r_2$ are likely due to effect of PEG modification and targeting moiety on rotational correlation time ($\tau_R$). ProCA32.collagen1 also exhibits the highest binding affinity to collagen type I ($K_d$ of 1.4 µM) among all other MR imaging agents available[3], due to the rational design of the targeting moiety. ProCA32.collagen1 also exhibits in vivo binding to collagen I in mouse tissues compared to ProCA32 without any binding (Supplementary Fig. 11). This strong binding affinity to collagen type I provides required

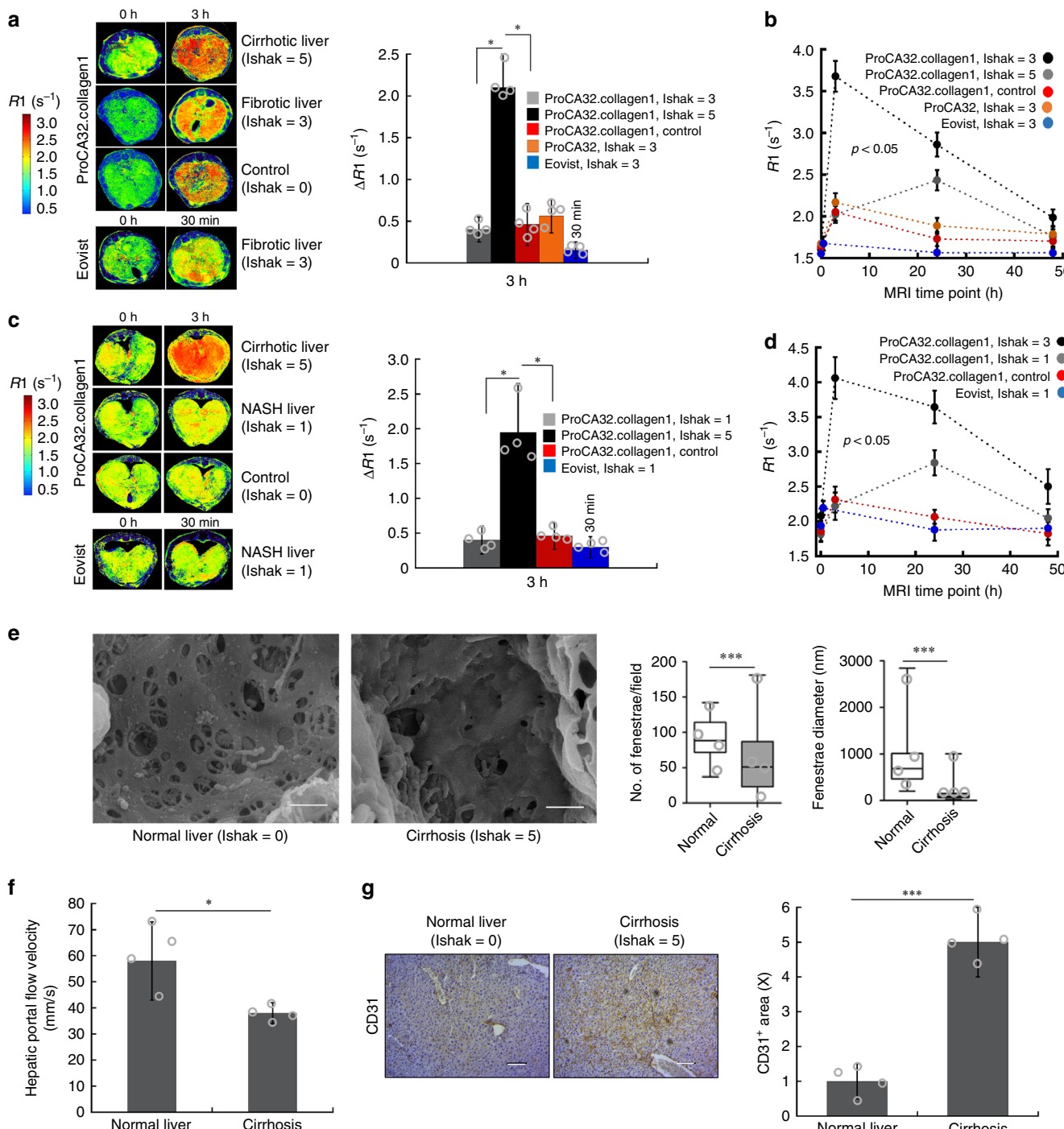

**Fig. 4** Detection of portal hypertension during liver cirrhosis in TAA/alcohol- and NASH diet-induced cirrhosis. **a** R1 map and ΔR1 values of early-stage (Ishak stage 3 of 6), late-stage (Ishak stage 5 of 6), and normal liver (Ishak stage 0 of 6) before and 3 h after injection of ProCA32.collagen1 and Eovist (30 min) in TAA/alcohol model. **b** R1 changes of liver over different MRI time points after injection of ProCA32.collagen1, ProCA32, and Eovist in TAA/Alcohol model ($P < 0.05$, paired student's $t$ test). **c** R1 map and ΔR1 values of NASH (Ishak 1 of 6), late-stage (Ishak stage 5 of 6), and normal liver (Ishak stage 0 of 6) before and 3 h after injection of ProCA32.collagen1 and Eovist (30 min) in NASH diet model. **d** R1 changes of liver over different time points after injection of ProCA32.collagen1 and Eovist in NASH diet model ($P < 0.05$, paired student's $t$ test). **e** Representative SEM images of sections from mice with late-stage liver fibrosis in TAA/alcohol model. Quantitation of number and size of fenestrations of liver sinusoids in mice with late-stage liver fibrosis measured by manually counting/measuring number and the diameters of fenestration in the SEM images (scale bar, 500 nm). **f** Velocity of portal vein blood flow as measured by Doppler ultrasound imaging shows high-portal hypertension detected at 3 h after injection of ProCA32.collagen1 in late-stage liver fibrosis in TAA/alcohol model. **g** Representative images of IHC stains of CD31 and quantitation of CD31 IHC stains of liver tissue in mice with late-stage liver fibrosis in TAA/alcohol model confirming intrahepatic angiogenesis. scale bar, 100 μm; $*P < 0.05$, $***P < 0.001$, unpaired two-tailed student's $t$ test; all data are represented as mean ± SD, $n = 4$ biologically independent animals in each group

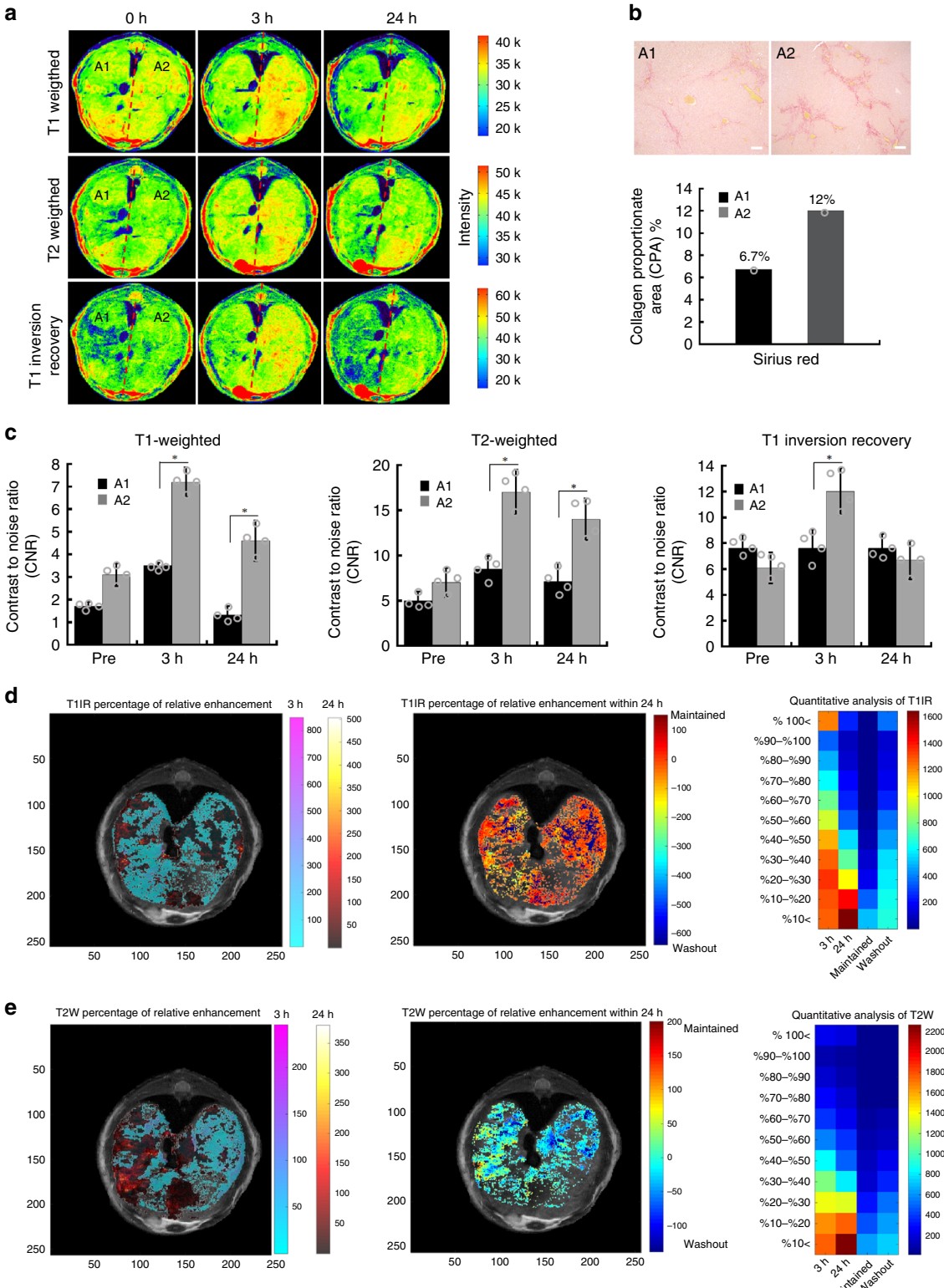

**Fig. 5** Demonstration of time-dependent mapping of collagen heterogeneity in DEN model. **a** T1-, T2-weighted, and T1-inversion recovery images of DEN-induced diseased liver before and 3 and 24 h post injection of ProCA32.collagen1 demonstrating collagen heterogeneity of liver in Area 2 (A2). **b** Sirius red staining and CPA analysis confirmed that A2 has more collagen than A1 which correlates with MRI (scale bar, 100 μm). **c** T1-, T2-weighted, and T1-inversion recovery images all demonstrated a higher contrast to noise ratio (CNR) in A2 of liver 3 h after ProCA32.collagen1 injection compared to A1. Data are represented as mean ± SD, n = 4 independent images. **d** Percentage of relative enhancement and quantitative voxel analysis of T1-inversion recovery images demonstrating liver regions enhanced at 3 and 24 h after injection of ProCA32.collagen1 compared to pre-injection (color scale represents number of voxels). **e** Percentage of relative enhancement and quantitative voxel analysis of T2-weighted images of liver demonstrating regions enhanced at 3 and 24 h after injection of ProCA32.collagen1 compared to pre-injection (color scale represents number of voxels)

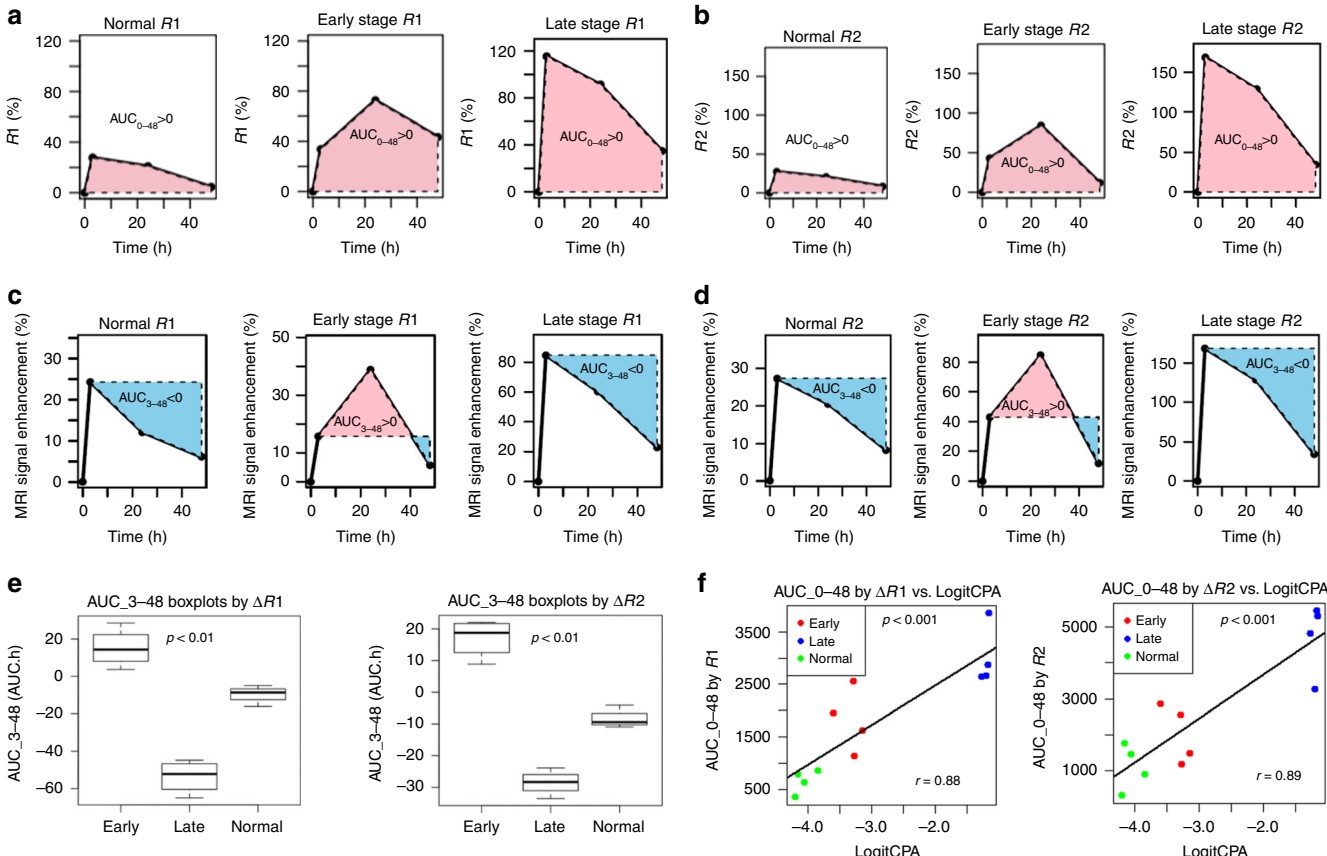

**Fig. 6** Assessment of contrast agent uptake and staging of NASH and liver cirrhosis in NASH diet model. **a, b** Area under the curve (AUC_0–48 R1 and R2) analysis with ProCA32.collagen1 enhanced MR in NASH diet model. **c, d** Area under the curve (AUC_3–48 R1 and R2) analysis demonstrates the washout rate of ProCA32.collagen1 3 h post injection in late-stage fibrosis based on decreasing R1 and R2 values and negative value of AUC compared to normal liver and early stage fibrotic liver. **e** AUC_3–48 h boxplot analysis demonstrating the ability of ProCA32.collagen1 to distinguish normal vs. early stage NASH and early stage NASH vs. late-stage fibrosis in NASH diet model ($P < 0.01$, unpaired two-tailed student's t test; the midline is the median of the data, with the upper and lower limits of the box being the third and first quartile, 75th and 25th percentile, respectively. The whiskers extends up to 1.5 times the interquartile range and show the minimum and maximum as they are all within that distance). **f** Scatter-plots of Logit-transformed CPA against AUC_0–48 showed that AUC_0–48 and CPA levels are well correlated in NASH diet model. $P < 0.001$, unpaired two-tailed student's t test; all data are represented as mean ± SD, $n = 4$ biologically independent animals in each group

sensitivity to detect early stage secretion of collagen in the liver (1–20 nmolg$^{-1}$)[37]. Our developed ProCA32.collagen1 has shown serum stability and no Gd$^{3+}$ deposition in brain (Supplementary Fig. 6d) that are equivalent to the macrocyclic agents.

Early stage fibrosis and NASH can be reversed if it is detected early[38,39]. However, current techniques including FibroScan and MRE as well as MRI with current clinical contrast agents, failed to detect early stages of NASH and fibrosis[40]. We have shown that early-stage alcohol-induced liver fibrosis (Ishak stage 3 of 6) and early stage NASH (Mild-1A, zone 3, perisinusoidal or Ishak stage 1 of 6) can all be detected using both R1 and R2 maps (24 h post injection). The detected MRI signals for both early and late stages correlated with results from histology analysis and CPA quantification, and were further validated by Gd$^{3+}$ detection with ICP-OES (Figs. 2c, d, 3c, d, Supplementary Fig. 9). As seen in Fig. 6e and Supplementary Fig. 12, ProCA32.collagen1 can reliably distinguish early stage NASH and fibrosis vs. late-stage fibrosis vs normal liver using both $r_1$ and $r_2$ properties. Another major advantage of ProCA32.collagen1, is its potential to stage and predict fibrosis scores in NASH diet model as revealed by correlation of AUC_0–48 of ΔR1 and ΔR2 values from MR mapping with CPA levels. This robust detection of early stages of the disease is due to five-fold increase in $r_1$ relaxivity over clinically approved contrast agents and the agent's collagen targeting capability.

ProCA32.collagen1 also has a high-$r_2$ value that enables the application of several imaging methodologies which reduces the possibility of detecting artifacts. Because of high $r_1$ and $r_2$, short-T1 inversion recovery with long TE of 32.67 ms was used to increase the sensitivity and achieve a high contrast between normal areas of liver and fibrotic regions. Using this pulse sequence, in the first step, liver signal was suppressed by selecting appropriate inversion times and short liver T1 because of high $r_1$ of contrast agent, then the second step suppression occurs with long TE because of short T2 of liver due to high-$r_2$ value of ProCA32.collagen1 (Supplementary Fig. 10). The effect of short-T1 inversion recovery with long-TE methodology can also be observed in detecting collagen heterogeneity as it shows higher sensitivity compared to other imaging pulse sequences (Fig. 5d) based on number of voxels that had ~100% increase in intensity.

At 3 h post injection in late-stage fibrosis, ProCA32.collagen1 with SMI exhibits organ distribution and retention time for possible detection of vascular changes due to intrahepatic angiogenesis, and portal hypertension[41], as well as collagen heterogeneity in liver. In addition, pharmacokinetic data (Table 3) suggest the long terminal half-life of ~10 h for ProCA32.collagen1 in blood and higher exposure time (AUC was significantly higher than ProCA32) with volume of distribution ($V_{ss}$) of 1.77 L/kg (two-fold higher than that of ProCA32), which are the indications

of targeting ability of contrast agent and its distribution and penetration in tissues.

The presence of portal hypertension can be observed in scanning electron microscopy (SEM) images and blood flow (Fig. 4e, f). Normal liver sinusoidal endothelial cells (SEC) have fenestrae. However, in late-stage fibrosis, SECs lose fenestrae when activated and express CD31 (Fig. 4e, g). Defenestration, as seen from SEM images, leads to portal hypertension in late-stage fibrosis, as the blood flow out of the liver is significantly reduced (Fig. 4f), therefore, it can be speculated that the increase of R1 and R2 at 3 h post injection of ProCA32.collagen1 and slow washout (Fig. 6c, d) is attributed to portal hypertension.

We have also shown that ProCA32.collagen1 at 3 h post injection, can demonstrate collagen distribution differences under

MRI in three different animal models of NASH diet, TAA/alcohol and DEN (Fig. 7a, Supplementary Fig. 7). This information might be essential for understanding the molecular mechanism of fibrosis formation and determining effective treatment as well as image-guided biopsy. In addition, ProCA32.collagen1 is capable of monitoring liver fibrosis regression and treatment in cirrhotic mice generated from TAA/alcohol model (Supplementary Fig. 17). Our data suggest that ProCA32.collagen1 can distinguish pirfenidone-treated liver from normal liver. This capability can be an essential step prerequisite to human clinical trials, to monitor treatment efficacy. Taken together, ProCA32.collagen1 has been shown to have strong human translational potential and application because of its high binding affinity and specificity to collagen in human HCC tissues with cirrhosis (Fig. 7c) and significantly reduced dose because of high relaxivity and the greatest metal selectivity than all other clinically approved $Gd^{3+}$-based contrast agents. The contrast agent is expected to overcome the major clinical barriers in early diagnosis, noninvasive detection and staging of chronic liver diseases, and have strong translational potential for monitoring treatment efficacy.

### Table 3 Summary of pharmacokinetic parameters in mouse for ProCA32.collagen1 and ProCA32

| Contrast Agent | ProCA32 | ProCA32.collagen1 |
|---|---|---|
| $t_{1/2\beta}$ (h) | 8.09 | 9.93 |
| $V_c$ (L/kg) | 0.20 | 1.53 |
| $V_{ss}$ (L/kg) | 0.76 | 1.77 |
| CL (mL/min/kg) | 0.11 | 0.36 |
| MRT (h) | 13.90 | 14.51 |
| AUC (ng h/mL blood) | 30773.00 | 140073.00 |

## Methods

**Protein expression, purification, and lysine PEGylation.** In the contrast agent design of ProCA32.collagen1, collagen type I targeting peptide (GGGKKWHCY-TYFPHHYCVYG) for human type I collagen was linked to the C-terminal of ProCA32 by a flexible hinge and surface modification by PEGylation. ProCA32.

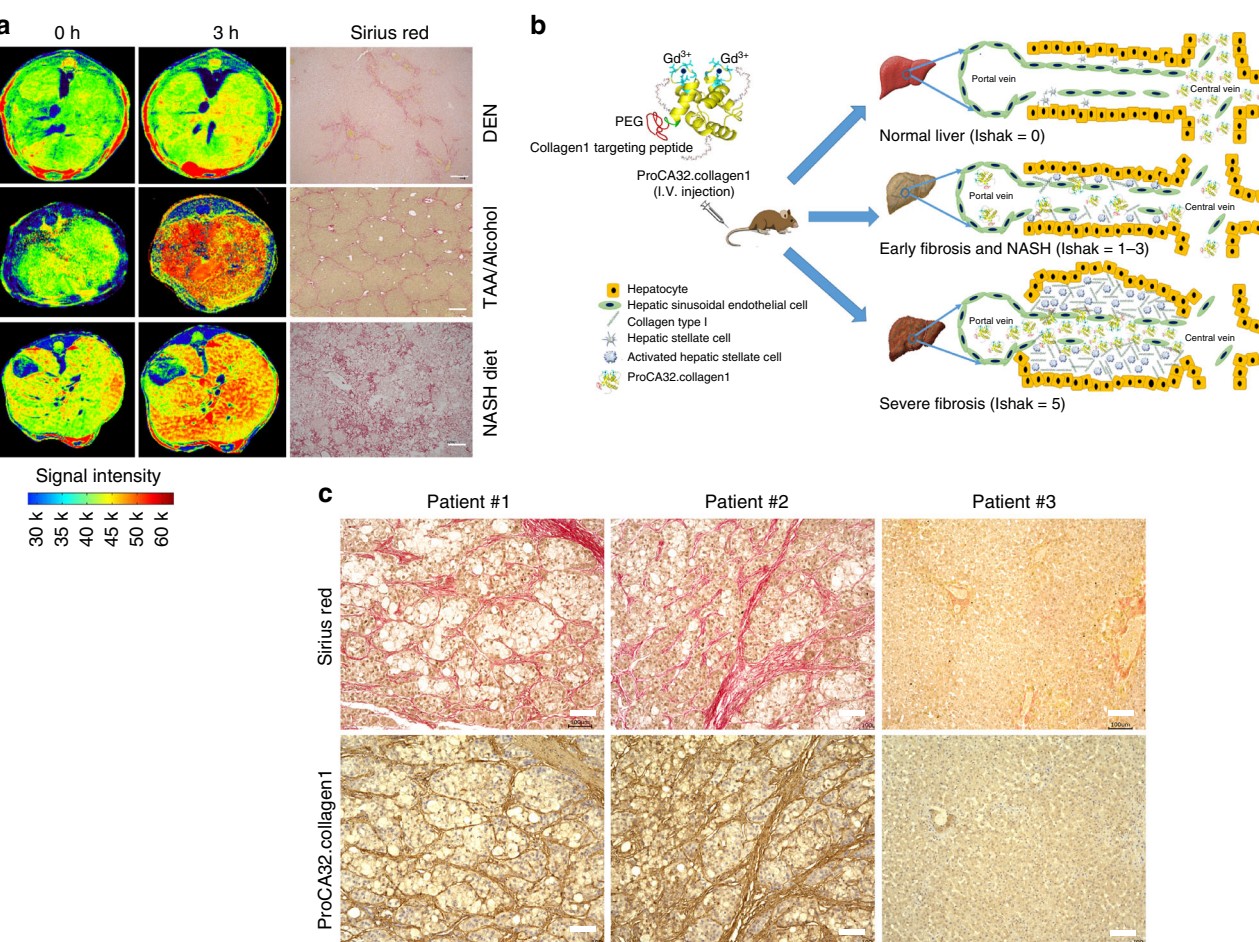

**Fig. 7** Human translational potential of ProCA32.collagen1. **a** ProCA32.collagen1 enhanced MRI can demonstrate different collagen distribution in DEN-, NASH diet-, and TAA/alcohol-induced models. **b** Schematic illustration of ProCA32.collagen1 distribution and suggested mechanism in different stages of fibrosis. **c** Demonstration of three hepatocellular carcinoma (HCC) and normal human liver tissue microarray stained with Sirius red (red) and ProCA32. collagen1 (brown) exhibiting the strong binding of the contrast agent to collagen human patient tissue (all scale bars, 100 μm)

collagen1 was expressed in Escherichia coli BL21 (DE3) cell strain and was purified using established procedures[23]. Purified ProCA32.collagen1 was confirmed by sodium dodecyl sulphate polyacrylamide gel electrophoresis (SDS-PAGE), ultra-violet visible (UV-vis) spectrometry, and electrospray ionization mass spectrometry. $Gd^{3+}$ was loaded to ProCA32.collagen1 at a 2:1 ratio. Other metals in ProCA32.collagen1 such as $Ca^{2+}$ were removed by chelex-100 and the metal content in ProCA32.collagen1 was analyzed by ICP-OES. The protein was PEGylated with methoxy succinimidyl carboxymethyl ester (M-SCM-2000) reagent with molecular weight of 2 kDa (JenKem Technology) and further purified. PEGylated products were analyzed by SDS/PAGE with protein staining by Coomassie Brilliant Blue and PEG staining by $I_2$. The protein absorbance was measured with UV–vis spectrometry by monitoring the Tryptophan (Trp) signal in the protein. The extinction coefficient of the protein was calculated based on the protein sequence and the final concentration was measured using Beer–Lambert law.

**Mice experiments.** All animal experiments were carried out in accordance with the NIH Guide for the Care and Use of Laboratory Animals and approved by Institutional Animal Care and Use Committee (IACUC) of Georgia State University and the University of Georgia.

**TAA/alcohol-induced liver fibrosis and cirrhosis.** To induce early stage fibrosis, BALB/c mice ($n = 4$) were treated with a twice weekly injection of thioacetamide (TAA) (200 mg/kg) and 10% ethanol in their drinking water for 6 weeks. To induce cirrhosis, BALB/c mice (6–7-week-old mice, $n = 4$) were treated with twice-weekly i.p. injections of TAA (200 mg/kg) and 10% ethanol in their drinking water for 12 weeks. The body weights were recorded every week or every four days. At the end of the experiments, animals were sacrificed. Livers, other organs, and blood samples were collected. Liver weights were measured, and liver pictures were taken. Tissue sections were prepared and analyzed by IF, IHC, or H&E stains using commercially available antibodies as indicated. Serum samples were prepared from collected blood samples. The serum samples were analyzed through an external company, Comparative Clinical Pathology.

**DEN-induced HCC.** For all of our studies, we employed female and male mice with C57BL/6 genetic background. Animals were housed in a pathogen-free animal facility under a 12 h light/dark cycle and fed standard rodent chow and water ad libitum. In order to induce HCC, 12-day-old mice were treated with a single dose of DEN (Sigma–Aldrich # N0756) dissolved in saline at a dose of 25 mg/kg body weight by i.p. injection on day 12. Mice in one randomly preassigned group ($n = 4$) were killed 10 months after DEN administration for histological and biochemical analyses. Mice were randomly distributed in various groups having equal males and females. Immediately after euthanizing, livers were removed, weighed and lobes were fixed in formalin and embedded in paraffin. Sections were stained with hematoxylin and eosin and examined microscopically.

**NASH diet-induced NASH and cirrhosis.** Two different groups of mice ($n = 4$ per group) on NASH diet were chosen for our studies in development of NASH and cirrhosis in mice with NAFLD. To develop early stage NASH, WT C57BL/6 mice were fed with NASH diet, which includes western diet (D12079B) and fructose in drinking water (42 g/L) starting from 6 week old for 6 months. Fructose in drinking water was autoclaved and changed twice a week. They were on western diet all the time. To develop cirrhosis, Liver-Specific Comparative Gene Identification-58 (CGI-58) knocked out (LivKO) mice were treated with the same exact procedure[42,43], and the levels of serum markers in blood circulation were compared with normal mice (Supplementary Fig. 13).

**Metal-binding affinity studies.** In order to investigate the $Gd^{3+}$ binding affinity of ProCA32.collagen1, a buffer system was used in which $Tb^{3+}$ binding affinity can be measured since $Gd^{3+}$ is spectroscopically silent. Then the $Gd^{3+}$-binding affinity was calculated using a competition assay[23]. Determining $Tb^{3+}$ binding affinity of ProCA32.collagen1 was based on the $Tb^{3+}$ LRET experiment in which 30 μM ProCA32.collagen1 was prepared in 5 mM DTPA, 50 mM HEPES, and 150 mM NaCl at pH 7.2. The protein-$Tb^{3+}$ LRET emission spectra were collected between 520 and 580 nm using an excitation wavelength of 280 nm. The free $Tb^{3+}$ concentrations ($[Tb]_{free}$) in each titration point were calculated by Eq. (1). The dissociation constant between $Tb^{3+}$ and ProCA32.collagen1 ($K_{d\ Tb,ProCA}$) was calculated by the Hill equation (Eq. (2)).

$$[Tb]_{Free} = \frac{K_{d\,Tb,DTPA} \times [Tb - DTPA]}{[DTPA]_{Free}}. \tag{1}$$

$$f = \frac{[Tb]_{Free}^{n}}{K_{d\,Tb,ProCA}^{n} + [Tb]_{Free}^{n}}. \tag{2}$$

$[Tb^{3+}]_{free}$ is the free $Tb^{3+}$ concentration calculated from the buffer system, $K_{d\ Tb,\ DTPA}$ is the dissociation constant of $Tb^{3+}$ and DTPA, and the dissociation constants of $Tb^{3+}$ to DTPA were obtained from National Institute of Standards and Technology Standard Reference Database[44]. [Tb-DTPA] is the concentration

of Tb-DTPA complex that is formed during titrations, $[DTPA]_{free}$ is the free DTPA in the buffer, $f$ is the fractional change of the LRET signal at each titration point, $n$ is the hill number, and $K_{d\ Tb,\ ProCA}$ is the dissociation constant between $Tb^{3+}$ and ProCA32.collagen1.

A competition assay was performed to measure the binding affinity of ProCA32.collagen1 for $Gd^{3+}$. The $Tb^{3+}$ fluorescence spectra were recorded by tryptophan excitation at 280 nm and emission from 500 to 650 nm. In all, 10 μM of ProCA32.collagen1 was used and 20 μM $Tb^{3+}$ was incubated with different concentrations of $GdCl_3$ from 0 to 1000 μM at room temperature overnight. The $Tb^{3+}$-FRET changes were measured by the emission at 545 nm. The apparent dissociation constants ($K_{d\ app}$) were calculated by fitting the plot of LRET peak intensities over different concentrations of $Gd^{3+}$, Eq. (3) and the dissociation constants of $Gd^{3+}$ to ProCA32.collagen1 ($K_{d\ Gd,ProCA}$) were calculated by Eq. (4).

$$f = \frac{\left([Tb]_T + [Gd]_T + K_{d\,app}\right) - \sqrt{\left([Tb]_T + [Gd]_T + K_{d\,app}\right)^2 - 4 \times [Tb]_T \times [Gd]_T}}{2 \times [Tb]_T}, \tag{3}$$

$$K_{d\,Gd,\,ProCA} = K_{d\,app} \times \frac{K_{d\,Tb,\,ProCA}}{K_{d\,Tb,\,ProCA} + [Tb]_T}, \tag{4}$$

where $f$ is the fractional change of the LRET signal, $[Tb]_T$ is the total $Tb^{3+}$ concentration, $[Gd]_T$ is the total $Gd^{3+}$ concentration at each titration point, and $K_{d\ Gd,ProCA}$ is the dissociation constant between $Gd^{3+}$ and ProCA32.collagen1 determined by Eq. (2).

The $K_{d\ app}$ for ProCA32.collagen1 was calculated first and then this value was used to calculate the actual dissociation constant using the Eq. (4).

For determining the calcium binding affinity of ProCA32.collagen1, 10 μM of ProCA32.collagen1 was added to the calcium–EGTA buffer system containing 50 mM HEPES, 150 mM NaCl, and 5 mM EGTA at pH 7.2. The system was titrated with different concentrations of $CaCl_2$ to alter the concentration ratio between the Ca-EGTA ([Ca-EGTA]) and free EGTA ($[EGTA]_{free}$). The tryptophan (Trp) fluorescence changes were monitored under the emission spectra between 300 and 390 nm as excited at 280 nm. The free calcium concentration at each titration point was calculated by Eq. (5)

$$[Ca]_{Free} = K_{d\,Ca,\,EGTA} \times \frac{[Ca - EGTA]}{[Tb]_{Free}}. \tag{5}$$

Furthermore, free calcium concentrations in the buffer were tightly monitored and calculated using an equation derived from Tsein's equation[32] with some modifications and the $K_d$ of EGTA-$Ca^{2+} = 1.51 \times 10^{-7}$ M was obtained from NIST. The $K_d$ of $Ca^{2+}$ to ProCA32.collagen1 ($K_{d\ Ca,ProCA}$) was determined by Eq. (6).

$$f = \frac{[Ca]_{Free}^{n}}{K_{d\,Ca,\,ProCA}^{n} + [Ca]_{Free}^{n}}, \tag{6}$$

where $f$ is the fractional change of Trp fluorescence intensity, and $[Ca]_{free}$ is the free $Ca^{2+}$ concentration at each titration point determined by Eq. (5).

The zinc-binding affinity of ProCA32.collagen1 was determined using a similar assay by competition titration using Fluozin1 dye[25,45]. The fluorescence of 2 μM Fluozin-1 was excited at 495 nm and the emission spectra were collected between 500 and 600 nm in the presence of 2 μM $Zn^{2+}$ and different concentrations of ProCA32.collagen1. The apparent dissociation constant ($K_{d\ app}$) was calculated by Eq. (7).

$$f = \frac{\left([Zn]_T + [ProCA]_T + K_{d\,app}\right) - \sqrt{\left([Zn]_T + [ProCA]_T + K_{d\,app}\right)^2 - 4 \times [Zn]_T \times [ProCA]_T}}{2 \times [Zn]_T}, \tag{7}$$

where $f$ is the fractional change of the fluorescence intensity of Fluozin-1, $[Zn]_T$ is the total $Zn^{2+}$ concentration, and $[ProCA]_T$ is the total concentration of protein contrast agent at each titration point. The dissociation constants between ProCAs and $Zn^{2+}$ ($K_{d\ Zn,ProCA}$) were then calculated by Eq. (8).

$$K_{d\,Zn,\,ProCA} = K_{d\,app} \times \frac{K_{d\,Zn,\,Fluozin}}{K_{d\,Zn,\,Fluozin} + [Fluozin]_T}, \tag{8}$$

where $K_{d\ app}$ is determined by Eq. (7), $K_{d\ Zn,\ Fluozin}$ is the $Zn^{2+}$ affinity to Fluozin-1 and $[Fluozin]_T$ is the total Fluozin-1 concentration.

**Measurement of water coordination number.** To determine the number of coordination water molecules in the inner sphere of $Gd^{3+}$-ProCA32.collagen1, the difference in $Tb^{3+}$ luminescence decay between $H_2O$ and $D_2O$ was calculated[46]. A fluorescence spectrophotometer with Xenon Flash light source was used to measure the $Tb^{3+}$ lifetime with a 10-mm pathlength quartz cell at ambient temperature. $Tb^{3+}$-ProCA32.collagen1 complexes were prepared in both $H_2O$ and $D_2O$. Then with an excitation at 265 mm with Xenon Flash lamp (PTI), $Tb^{3+}$ emission decay over time was monitored at 545 nm. Luminescence decay data were then fitted with a monoexponential decay equation. A standard curve of water number ($q$) over $\Delta K_{obs}$ (the difference of the decay constant between $H_2O$ and $D_2O$) was created

using well-characterized chelators, such as $Tb^{3+}$-DTPA ($n = 1$), $Tb^{3+}$-EDTA ($n = 3$ when [EGTA]:[$Tb^{3+}$] = 1:1), $Tb^{3+}$-NTA, and $Tb^{3+}$ in aqueous solution ($n = 9$). The water number of $Tb^{3+}$-ProCA32.collagen1 and $Tb^{3+}$-ProCA32 was calculated by fitting $\Delta K_{obs}$ into the standard curve. The water numbers for Gd-DTPA and Eovist were obtained from literature reports[47–49].

**Serum and transmetallation studies.** For serum stability experiments, human serum (sterile-filtered, human male AB plasma with composition of hemoglobin ≤ 30 mg/dL, and impurities of endotoxin ≤ 10 EU/mL) was purchased from MilliporeSigma, catalog # H4522. Then $Gd^{3+}$-loaded ProCA32.collagen1 (250 µM) was mixed with human serum in 1:1 ratio, and then incubated at 37 °C for different time points up to 13 days. SDS-PAGE gel was performed to monitor the contrast agent band with Ponceau S solution and Coomassie Brilliant Blue staining. The serum samples purchased from MilliporeSigma were collected with the donor being informed completely and with their consent. The study has been determined by the Institutional Review Board (IRB) of Georgia State University to be exempt from federal regulations and it was determined that it meets the organization's ethical standards. Transmetallation of ProCA32.collagen1 and clinical MRI contrast agents were evaluated by measuring the relaxation rate changes in the presence of $Zn^{2+}$ and phosphate over time. The relaxation rate change of ProCA32.collagen1 was monitored with 50 µM of the contrast agent, 100 µM $Gd^{3+}$, 100 µM $Zn^{2+}$, and 1.2 mM $PO_4^{3-}$. Then the percentage of decrease in relaxation rate was calculated using the last time point after 4 days compared to their initial values[50].

**Immunofluorescence imaging.** Immunofluorescence staining was performed on fibrotic tissues collected from the mice livers post injection of ProCA32 and ProCA32.collagen1 at 24 h time point. Cryosections of fibrotic liver tissues were cut at a thickness of 5 µm. The tissues were fixed with 4% (vol/vol) formaldehyde for 10 min at room temperature and then washed three times with Tris-buffered saline with Tween 20 (TBST). Tissues were then blocked at room temperature for 1 h with 1% bovine serum albumin (BSA) in TBST, and then incubated with two primary antibodies for 1.5 h. A ProCA32.collagen1 rabbit antibody prepared by our lab was used as a primary antibody for the contrast agent at 1:100 dilution. Another primary antibody for collagen type I, Anti-Collagen I antibody [COL-1] (ab34710) was used with 1:500 dilution. After washing three times with TBST, the slides were incubated with Alexa Fluor 488-labeled goat anti-rabbit IgG (Invitrogen; catalog #R37116) as a secondary antibody for collagen type I with 1:200 dilution. Alexa Fluor 594-labeled goat anti-rabbit IgG (Invitrogen; catalog #A-11012) was used as a secondary antibody for ProCA32.collagen1 (1:200 dilution). The nuclei of the cells were stained by DAPI (blue, ThermoFisher scientific, catalog #62248, 1:1000 dilution).

**Histology analysis.** For immunohistochemistry, paraffin-embedded liver sections were cut (4 µm) and mounted on slides (Probe On; Fisher Scientific). The sections were dried, deparaffinized, and hydrated with xylene and graded ethanol. Formalin-fixed samples were then stained with Sirius Red according to standard procedures. Sirius Red stained slides were analyzed by a pathologist, at Emory University School of Medicine, to score the amount of liver disease based on Ishak scoring system[51,52]. For H&E Staining, livers were fixed in 10% (vol/vol) formalin. Then they were dehydrated in increasing concentrations of alcohol, cleared in xylene, and embedded in paraffin. Serial 5-µm sections were prepared, stained with H&E, and processed for light-microscopic examination. Alpha-smooth muscle actin stain was performed in immersion fixed paraffin-embedded sections of mouse fibrotic and normal liver tissues using Anti-alpha smooth muscle Actin antibody (Catalog # ab7817) with concentration of 0.1–0.5 µg/ml. Heat mediated antigen retrieval was performed in citrate buffer pH = 6 before using the standard IHC staining protocol. An experienced pathologist was blinded to the groups of H&E staining to evaluate the organ toxicity of ProCA32.collagen1. For CD31 staining (mouse CD31, catalog# 561813, Millipore), the quantity of CD31 IHC is presented as fold changes in CD31 stains compared to that of non-fibrotic mice (normal).

Four human HCC and normal precut liver tissue microarray (formalin-fixed paraffin-embedded), each containing 20 cases of liver tumor with 4 normal tissues from autopsy, duplicated cores per case were purchased from US Biomax, Inc., catalog # LV482 and used to stain for collagen and ProCA32.collagen1. Each slide had 48 cores, 24 cases, with core dimeter of 1.5 mm and thickness of 5 µm. The tissue samples were collected with the donor being informed completely and with their consent. The study has been determined by IRB of Georgia State University to be exempt from federal regulations and it was determined that it meets the organization's ethical standards. In this procedure, the contrast agent was incubated with the ProCA32.collagen1 antibody (1:2 ratio, polyclonal rabbit-anti-ProCA32 antibody) and then the preincubated mixture was added into the slides. Patient #1 and #2 are shown on adjacent slides and Patient #3 is the control (Fig. 7c). The CPA, as determined by the % area stained with Sirius Red, was quantified from the histology images using ImageJ as per standard procedures[52]. The percentage of Sirius red stained areas was calculated based on the entire slide scan. The results are mean values across 10 different slides from different locations of mouse liver (Supplementary Fig. 9).

**MRI scan.** All mice were imaged on a 7-T Agilent MRI scanner at the University of Georgia. Animals were anesthetized using isoflurane and their respiration rate was monitored and the temperature was maintained with a small animal physiological monitoring system. Anesthesia was adjusted to maintain a respiration rate of 65 ± 5 breaths per minute. Fast spin echo-based inversion recovery sequence was used to derive T1 map and spin echo multi-slice sequence for T2 quantitation was used before and after intravenous (I.V.) administration of 0.02 mmol/kg of ProCA32.collagen1, ProCA32 or Eovist. T1 map images were acquired with inversion recovery times of 10, 222, 435, 648, 861, 1074, 1287, and 1500 ms. Other acquisition parameters include: repetition time of TR = 5000 ms, Effective TE = 32.67 ms, field of view (FOV) = 35 × 35 mm, matrix = 256 × 256, slice thickness = 1.0 mm, and 12 image slices with no gap, the total acquisition time was 21 min. T2 map images were collected before and after the contrast agent injection with the same parameters as above except TR = 2000 ms, different echo times, TE (8, 16, 24, 32, 40, 48, 56, 64, 72, 80, 88, 96 ms), the total acquisition time was 8 min. Both R1 and T2 maps were generated using a custom written MATLAB (Mathworks, Natick, MA) program. T2-weighted images (fast spin echo multi slice) were collected with acquisition parameters as follows: TR = 4000 ms, effective TE = 40 ms, field of view, FOV = 35 × 35 mm, matrix = 256 × 256, slice thickness = 1.0 mm, and 12 image slices with no gap, the total acquisition time was 4 min. T1-wieghted images (spin echo multi-slice) were collected with the same acquisition parameters except: TR = 500 ms, and TE = 14.89 ms, the total acquisition time was 2 min. T1 inversion recovery with inversion time of 10 ms was also collected with the same acquisition parameters as T1 map. All scans were performed with the an average of 2 for each.

**Histogram and voxel analysis of MRI images.** Quantitative analysis of acquired MRI data was performed using MATLAB. A set of custom written scripts was prepared to create the histograms from R1 map images. By creating the histograms, a quantitative measurement for the R1 values in R1 maps can be achieved in each time point after injection of the contrast agent. For R1 map histogram analysis, first the R1 value of each voxel was calculated. Then, the minimum and maximum R1 value in each image was determined. The interval between the maximum and minimum value was divided into 500 equal-sized bins. The voxels which were within each bin were counted. The 0 h time point was chosen to be the baseline, and other time points (3 and 24 h) were plotted based on 0 h time point.

To demonstrate the time-dependent enhancement of contrast agent, the relative enhancement was calculated using the initial T1-, T2-weighted, and T1-inversion recovery MRI image as the baseline (0 h time point). The intensity of each voxel was subtracted from the initial value then divided by the initial value, and then the obtained number multiplied by 100 to get the percentage of relative enhancement at various time points. To identify areas that remained enhanced or the contrast agent is washed out, another variable was defined. In this variable, the desired time point value was subtracted from its previous time point and then divided by the same value, then multiplied by 100. If a positive value is achieved, it can be concluded that the voxel enhancement was unchanged or increased, whereas a negative value indicated that the contrast agent washed out. To have a better quantitative analysis, a matrix was created, which demonstrates the percentage of enhancement vs time points. In this matrix, 12 bins were created, where the initial bin was the voxels with values of less than 10% change and the last bin included voxels with more than 100% change. Then the voxels within each bin were counted. As previously noted, this matrix represented positive values or changes for 3 and 24 h, maintained and negative washout values.

**Determination of $r_1$, and $r_2$ and $\Delta$R1 and $\Delta$R2 analysis.** The ability of a contrast agent to change a relaxation rate is represented quantitatively as relaxivity, $r_1$ or $r_2$, where the subscript refers to either the longitudinal ($1/T_1$) or the transverse rate ($1/T_2$). Relaxivity is simply the change in relaxation rate after the introduction of the contrast agent ($\Delta(1/T)_1$) normalized to the concentration of contrast agent or metal ion (M) as in Eq. (9)

$$r_1 = \frac{\Delta(1/T_1)}{[M]}. \tag{9}$$

The relaxivities of protein ($r_1$ and $r_2$) were determined using different concentrations of $GdCl_3$ and protein (2:1) at 37 °C with 1.4 T Bruker Minispec using saturation recovery and CPMG sequence, respectively. Using the equation below the relaxation rate for both $T_1$ and $T_2$ were measured. The slope of the curve will be ($r_1$) and ($r_2$) relaxivities. Furthermore, the $T_1$ and $T_2$ of Eovist and ProCA32.collagen1 were also measured by using a 7 T Agilent scanner using saturation recovery and spin echo sequence. Values for $r_1$ and $r_2$ were calculated by Eq. (10).

$$r_i = \left(\frac{1}{T_{is}} - \frac{1}{T_{1ib}}\right)/[Gd^{3+}] \quad i = 1, 2. \tag{10}$$

In addition, the change in liver longitudinal relaxation rate $\Delta$R1 and $\Delta$R2 defined as R1,2 $_{post}$ − R1,2 $_{pre}$ was calculated.

**Enzyme-linked immunosorbent assay.** Collagen type I solution from rat tail (Sigma-Aldrich) was coated in 96-well plates at 4 °C overnight. After being blocked with 5% BSA, a series of different concentrations of ProCA32.collagen1 was incubated with the collagen type I solution. After incubation at 4 °C overnight, the

unbounded ProCA32.collagen1 was washed away with 1× TBST. The homemade polyclonal rabbit-anti-ProCA32 antibody was applied with 1: 1000 dilution as the primary antibody against protein contrast agent. A stabilized goat-anti-rabbit HRP-conjugated antibody (Pierce) was used as the secondary antibody (1:200). After a robust wash with 1× TBST, the remaining secondary antibody in the 96 well plate was visualized using 1-Step™ Ultra TMB-ELISA Substrate Solution (Thermo Fisher Scientific). The absorbance intensity was detected by the FLUOstar OPTIMA plate reader at an absorbance wavelength of 450 nm.

**Organ distribution analysis**. Different organ tissue samples from mice were collected after euthanasia. The collected tissues (~0.2 g) were digested with 70% (wt/vol) ICP-grade $HNO_3$ at 40 °C overnight. The $HNO_3$ solution containing the digested tissues were collected the next day, filtered, and diluted with 2% (wt/vol) $HNO_3$ to 8 mL. Then each sample containing different organ was analyzed by ICP-OES to measure the $Gd^{3+}$ concentration at wavelength of 342.246 nm. $YCl_3$ at 2 ppm was used as internal reference and different $Gd^{3+}$ solutions with different concentrations ranging from 5 to 1000 ppb were used as reference.

**AUC analysis**. Percentage of R1 and R2 enhancement for each post-injection time-point coming from R1 and R2 values in MR mapping was calculated as: % enhancement = (postinjection signal − preinjection baseline signal)/(preinjection baseline signal) × 100. First, the percentage of R1 and R2 enhancement curves were plotted and then the areas under the entire curve (AUC_0–48) were calculated. AUC values were calculated for each mouse from normal, early and late stage groups of NASH diet model. These represented the average percentage of enhancement ($n = 4$) (Fig. 6a, b). We also calculated AUC_3–48, area under the percentage of R1 and R2 enhancement curve in the time window 3–48 h, using the 3 h data point as a reference for each subject. Logit CPA is the logit transformation of CPA values and can be calculated as LogitCPA = ln [(CPA/100)/(1 − CPA/100)] (Fig. 6f). Statistical significance was tested by unpaired two-tailed Student's $t$ test (Excel), with $p < 0.05$ as significant. ROC was calculated using SAS 9.4 (SAS Institute) and R 3.2.2 software.

**CNR, ΔR1 and ΔR2 calculations**. All CNR, R1, ΔR1, R2, and ΔR2 values of all plots represent mean values across multiple MRI slices in each group. The total animal numbers in each group are denoted in each figure legend. Data were statistically analyzed by comparing different groups. The p values were calculated using unpaired two-tailed Student $t$ test and have been denoted in figure legends. Statistical analysis was performed in GraphPad Prism 5 (GraphPad Software). The mouse group sizes were set to support statistically valid data and to minimize the use of the animal. Contrast to noise ratio (CNR) in the liver was calculated by choosing a region of interest (ROI) within the liver, estimating the mean intensity within the ROI, and then subtracting it by the mean intensity of muscle. Then this value was divided by the standard deviation of a region outside the mouse torso.

**Pharmacokinetic studies**. Composite plasma concentration-time profiles following intravenous injection of ProCA32.collagen1, and ProCA32 in mice ($n = 6$) were analyzed using noncompartmental analysis tool of PhoenixWinNonlin software (Version 6.2). $Gd^{3+}$ concentration in serum collected after injection of ProCA32.collagen1 at different time points was measured by ICP-OES at 342 nm up to 7 days. After injection, the $Gd^{3+}$ concentration in serum decreased until it was below the detection limit of ICP-OES. The area under the concentration-time curve ($AUC_{last}$ and $AUC_{inf}$) was calculated by linear trapezoidal rule. Values of clearance and volume of distribution were estimated values. The elimination rate constant value ($k_{el}$) was obtained by linear regression of the log-linear terminal phase of the concentration–time profile using at least 3 nonzero declining concentrations in terminal phase with a correlation coefficient of >0.8. The terminal half-life value ($t_{1/2}$) was calculated using the Eq. (11)

$$t_{1/2} = 0.693/k_{el}. \quad (11)$$

**Measurement of hydroxyproline content in liver tissues**. In the Hydroxyproline Assay Kit (Sigma-Aldrich, St. Louis, MO), hydroxyproline concentration was determined by the reaction of oxidized hydroxyproline with 4-(dimethylamino) benzaldehyde (DMAB), which results in a colorimetric (560 nm) product, proportional to the hydroxyproline present. Hydroxyproline standards for colorimetric detection were prepared as follows: First, 10 μL of the 1 mg/mL hydroxyproline standard solution was diluted with 90 μL of water to prepare a 0.1 mg/mL standard solution. Then, 0, 2, 4, 6, 8, and 10 μL of the 0.1 mg/mL hydroxyproline standard solution was added into a 96-well plate, generating 0 (blank), 0.2, 0.4, 0.6, 0.8, and 1.0 μg/well standards. For sample preparation, 10 mg of liver tissue was homogenized in 100 μL of water and then transferred to a pressure-tight polypropylene vial with PTFE-lined cap. Then, 100 μL of concentrated hydrochloric acid (HCl, ~12 M) was added, to hydrolyze at 120 °C for 3 h. Solutions were mixed and centrifuged at 10,000 × g for 3 min. About 10–50 μL of supernatant was transferred to a 96-well plate. All wells were evaporated to dryness under vacuum. The assay reagents were prepared as follows: Chloramine T/oxidation buffer mixture − 100 μL is required for each reaction well. For each well, 6 μL of Chloramine T concentrate was added to 94 μL of oxidation buffer and mixed well. Diluted DMAB

reagent—100 μL is required for each reaction well. For each well, 50 μL of DMAB was added and concentrated to 50 μL of perchloric acid/isopropanol solution and mixed well. Then, 100 μL of the Chloramine T/oxidation buffer mixture was added to each sample and standard well and incubated at room temperature for 5 min. Totally, 100 μL of the Diluted DMAB reagent was added to each sample and standard well and incubated for 90 min at 60 °C, and finally the absorbance was measured at 560 nm (A560).

**Reporting summary**. Further information on research design is available in the Nature Research Reporting Summary linked to this article.

## Data availability
All data used for this paper are available from the authors upon request.

## Code availability
The custom codes created to generate histograms of T1 maps or R1 maps as well as percentage of relative enhancement and quantitative voxel analysis of T1 and T2 MR Images are available at GitHub, https://github.com/msalarian2013/MRI-Codes along with a "Readme" file for instructions. For generating the histograms and images with custom codes, MATLAB 2016 (Mathworks, Natick, MA) with academic license, ImageJ 1.51j8 which is publicly accessible at https://imagej.nih.gov/ij/download.html and MRIcron version 1 which is publicly accessible at http://people.cas.sc.edu/rorden/mricron/install.html were used.

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

## Acknowledgements

We thank Dr. Peter Caravan for his discussion and advice. Dr. Micheal Kirberger for carefully editing the paper. Drs. Qun Zhao, and Yubin Zhou for their helpful discussion. This work was supported in part by a Molecular Basis of Disease fellowship (to M. Salarian) and National Institute of Health (NIH) Research Grants EB007268, CA183376 and AA025863 (to J.J.Y.), CA118113 (to Z.-R. L.) and S10RR023706 (instrumentation grant for the University of Georgia Bio-Imaging Research Center).

## Author contributions

M.Salarian, and J.J.Y. designed the research and experiments; R.C.T., M. Salarian, Y.Z., J.Q., J.L., and M. Sharma created animal models; M. Salarian, S.X., K.H., and M.N. analyzed the MRI data; A.B.F., L.Y., and Z.R.L. reviewed the histopathology and provided critical suggestions for animal models; S.K. provided the critical suggestions regarding MR imaging and data analysis; S.T. helped with the ELISA experiment and data analysis; M. Salarian and O.Y.I. purified the protein; P.M. provided the clinical contrast agent and suggestion for MR data analysis; Y.H., G.Q., and X.M. provided the suggestions and performed the statistical analysis; R.M. helped with the pharmacokinetic data analysis and M. Salarian. and J.J.Y. wrote the paper.

## Additional information

**Competing interests:** The authors declare no competing interests.

