## [Peer Review File · Nature Communications]

Reviewers' comments:

Reviewer #1 (Remarks to the Author):

General comments

The use of Gd-containing proteins as efficient MRI CAs has already been reported by the same group. Different proteins have been tested finally selecting Parvalbumin (rat, ProCA-30 and humanized, ProCa-32). Gd-containing proteins have been shown to act as good reporters in several targeted applications (1)Pu, Fan, et al. "GRPR-targeted protein contrast agents for molecular imaging of receptor expression in cancers by MRI." Scientific reports 5 (2015): 16214. 2)Pu, Fan, Shenghui Xue, and Jenny J. Yang. "ProCA1. GRPR: a new imaging agent in cancer detection." (2016): 449-452. 3)Pu, Fan, et al. "Prostate-specific membrane antigen targeted protein contrast agents for molecular imaging of prostate cancer by MRI." Nanoscale 8.25 (2016): 12668-12682. 4)Qiao, Jingjuan, et al. "Molecular imaging of EGFR/HER2 cancer biomarkers by protein MRI contrast agents." JBIC Journal of Biological Inorganic Chemistry 19.2 (2014): 259-270.). The collagen targeted MRI CA herein reported used as targeting vector the peptide previously reported by Caravan et al.

Overall the work is carefully done and the conclusions look sound.

Below specific comments are reported.

Line # Comment

33 the Title (INTRODUCTION) is missing

68 Gd-DTPA appears

69 remove the comma after "while"

74 a complete description of the product should be given including the nature of the protein and its origin (human or animal)

85 lysine residues are probably intended as anchor point for PEG, not for collagen binding (consider revising)

89 A deeper analytical characterization should be reported, e.g., calculated MW, Size-exclusion chromatography, SDS-PAGE, etc. For the assessment of Gd content, it is not clear how the protein content is determined. Possibly the correct ratio is 2:1 Gd-molecule (not 1:2)

90-91 The dose in Fig 1e (0.0013 mmol/kg) is purely speculative; it should not be presented together with experimental data. Moreover, the conversion does not seem to be correct. Considering the HED factor of 12.3 between mouse and man, the adopted dose 0.02 mmol/kg corresponds to 0.0016 mmol/kg

94 and Fig. 1b not clear what Lysine-ProCA32.collagen1 is, nor if there is a difference between Lysine PEGylated ProCA32.collagen1 and ProCA32.collagen1. A consistent terminology should be used throughout the paper. See also in Fig. 1d-1e, where PEG-ProCA32.collagen1 appears to be different from ProCA32.collagen1. Neither ProCA32-P40 is clearly described.

96 Check “Ominiscan” instead of Omniscan (also in Fig.1c). Experimental details and methods used to obtain Fig 1c are missing. The methods in “Laurent S, Vander Elst L, Henoumont C, & Muller RN. (2010). How to measure the transmetalation of a gadolinium complex. Contrast media & molecular imaging, 5(6), 305-308.” should be referred to.

97 the tested product (ProCA32.collagen1) has a lower than reported Gd content (see line 89). Why? Is there a difference in Gd affinity for the two chelating sites? All results should refer to the same batch, after full characterization.

100-101 See above, 90-91

101

114-117 check “raveled”

The measured hydration number of 0.5 for the Tb-containing system may not be transferable to Gd-containing protein

116-117 A unique and consistent terminology should be defined and used throughout the paper

125 This sentence is not correct, since ProCA32.collagen1 is not among clinical GBCAs. Rewrite as “...ProCA32.collagen1 has a greater metal selectivity for Gd³⁺ than clinical gadolinium-based contrast agents.”

128-129 Actually from Fig 1f one can see that the main protein band is decreasing and possibly some lower MW bands appear. The image should reproduce the whole migration lanes, in order to allow the reader to assess the stability of the main product.

130-134 These sentences cannot be evaluated without Supplementary materials and related study methods.

135-136 The comment of possible reduced dosage does not pertain to the table of Fig. 1e. Possibly to the Relaxivity table (1d), but it is not clear on which data the ratio is calculated

141 Was the dose of Eovist also 0.02 mmol/kg? The standard dose, after correction for body surface, is 0.3 mmol/kg. This dose would compensate for the lower relaxivity of Eovist and yield a more meaningful comparison.

166-207 Figure 2 has > 50 panels, conveying confused and unreadable information. The legends are little useful

248-267 Eovist is tested only in animals with Ishak =1 or =3, never with Ishak =5, which are the only cases in Fig. 3a-d where the response to ProCA32.collagen1 is significantly higher. Therefore a comparison is not possible

- 309 T1 inversion
- 318 Discussion (not Dissuasion)
- 328 “precision is doubled” where is it shown?
- 330-332 From previous description (see 105-106), it seemed that ProCA32-P40 contains PEG, differing only in the lack of the collagen binding peptide. A clearer compound identification should be provided. The contribution of PEG is expected to influence tR , more than second sphere water molecules
- 380-381 See above 130-134. Comparison with other agents should imply the same administration protocol and analytical methods
- 382 “molecular biomarkers” too generic, better “collagen”
- 384 See above: the dosage in humans is speculative
- 402 i.p. injection? delete
- 404 The TAA dose (not concentration) is not well defined and not consistent with above 200 mg/kg
- 500 How was the antibody prepared? Is a rabbit antiserum?
- 502-3 The COL-1 antibody is from mouse. The corresponding secondary Ab is not reported

Reviewer #2 (Remarks to the Author):

This work describes a novel imaging approach with an innovative MRI contrast agent to visualize fibrosis in mouse models.

1. I have a series of concerns regarding the animal models of NASH/ASH/fibrosis used in this manuscript. Throughout the manuscript, it remains quite blurry, which models are presented (“early fibrosis”, “late fibrosis”). The figure legends need to be clear on these aspects (e.g., figure 2). Moreover, it is simply not true that a single injection of DEN in c57bl/6 mice would induce liver cirrhosis after 10 months. This is a non-fibrotic / non-cirrhotic model of HCC. If the authors observe cirrhosis, we are likely looking at artifacts related to tumor development (e.g., pathogenic angiogenesis, leading to Sirius Red stained areas). Same is true for “NASH diet induced cirrhosis” – this will simply not happen after 6 months of Western diet.

2. No translational data about the applicability of this agent in humans (toxicity, dosing, distribution, specificity) are provided. At least some work with primary human cells / tissues confirming targeting specificity need to be provided.

3. The true challenge in clinical trials is monitoring treatment efficacy non-invasively. There is a large series of effective antifibrotic treatments in NAFLD under development that we effective in mouse models (e.g., elafibranor, selonsertib, cenicriviroc). The authors need to demonstrate that the imaging can be performed longitudinally and is capable of monitoring fibrosis regression (or reduced fibrosis development) in mouse models.

4. Minor: Fibrosis need to be assessed by additional methods, including hydroxyproline and collagen Western blot.

Reviewer #3 (Remarks to the Author):

Format:

Please be advised that the manuscript (“Article”) format does not abide by Nature recommendations:

- Abstract >150 words
- Overall Text >3000 words
- Methods >3000 words
- References > 50
- Large number of sub-panels per figure (as many as 10 sub-panels occur in figure 3). Legends are very long >500 words

General Comments:

The manuscript describes a novel protein-based MR contrast agent that is specific to collagen 1. The motivation is to develop an MRI-sensitive marker for staging liver fibrosis. The research topic has important implication, since many current MRI methods lack the sensitivity to detect early fibrotic

changes in a quantitative manner. A main conclusion is that the change in MR relaxation rates (pre to post injection) at 24hrs allows the ability to distinguish normal, early stage, and late stage fibrosis, while an earlier time point (3hrs post-injection) is able reveal facets of angiogenesis and portal hypertension, indicative of more advanced disease.

Results and Claims: In my opinion, there is an excess of data and claims presented in this manuscript, which distracts from the overall objective and my understanding of the essential findings. I believe it is longer than it needs to be, and can be written more concisely. Several claims are given in the manuscript that point to novel methods for detecting and staging fibrosis, but it is never clear which finding is preferred or advantageous in a clinical setting. I feel there needs to be a more specific and in-depth analysis of 1-2 primary claims the authors feel most strongly about. Moreover, from the manuscript, it is difficult to piece together a singular procedure for using ProCA32collagen1 and an MR method in clinical application. Ultimately, my question is still, “Once injected, what single MR method should I perform, and when? And what primary data analysis method should I perform to best elucidate fibrosis stage?” If this still needs investigation, you should state the need for further work.

Pharmacokinetics of ProCA32collagen1: You should consider adding more pharmacokinetic data into the main document, and not in supplemental data. For a new agent, I think it’s crucial to discuss distribution kinetics, elimination half-life, and clearance, in addition to relaxivity, stability, and safety. I’m curious to know the short time-scale wash-in/wash-out characteristics of this agent in both normal and fibrotic tissue environments. Given a high R1 for late stage fibrosis and low R1 for normals at 3hrs, does this imply faster wash-in/wash-out for normals? How long will ProCA32collagen1 bind to collagen markers in fibrotic livers before being eliminated?

Pathology and Histology: The pathology work in this study is extensive and well-done. I am curious whether you analyzed any gross liver specimens from the mouse models. I think a valuable qualitative and quantitative comparison would be fibrosis seen on gross specimens versus slice-matched post-injection (24hrs) MRI R1 and R2 maps. This would truly validate contrast agent uptake, distribution, and wash-out in fibrotic and remote liver regions. I feel biopsy and histology proves the presence of fibrosis degree, but it is limited in emphasizing the geographic match between pathology and MR imaging data.

Time points: MR imaging of ProCA32collagen1 distribution was performed at 3, 24, and 48hrs, along with baseline measurements. Please state the logic for this procedure, especially in light of the agent’s half-life (~9hrs). Even though the use of animal models may preclude imaging at earlier time points, almost all current dynamic MR imaging occurs < 1hr post-injection. The use of the word “dynamic” (e.g DMI and DCE) should be tempered when considering 3, 24, and 48hrs post-injection time-points; this word implies imaging over a short time scale, usually over minutes. The authors

should at least address the possibility of missing the peak signal or max R1 time points, which may affect AUC and other results presented in Fig 3 and 4.

Methods: Some descriptions of experimental methods contain explicit data results. For example, the “statistical analysis” section describes the AUC results. Other method descriptions contain results shown in supplemental data figures. This may be ok, but I leave it to the Editors to determine whether these should be referenced in the text. In fact, the supplemental data contain lots of methods descriptions not referred to in main document.

From the histogram analysis of R1, R2, delta R1, and delta R2, there seems to be large variance in the images (Fig 3h, Fig 4e), yet very small error bars on plots. It is a bit unclear whether mean and standard deviation at each time point represents the entire imaging slice or volume. Further, there is no mention of a segmentation algorithm.

Style: I also recommend a further review to improve readability, word usage, and some grammar. There are many statements and phrases that can be eliminated to shorten the manuscript, without losing effect. Furthermore, authors should refrain from using exaggerated language like “dramatic” and “strikingly” when describing results. Emphasis should be placed on whether results are statistically sound.

More specific comments and inquiries follow below.

Specific Comments (from merged PDF file – Page.LineNumber)

Abstract

P1.L21 – do you mean “detection of liver fibrosis”? Other MR techniques are used to detect fatty liver disease and its heterogeneity

Introduction

P2.L41 – “which is consisted of...” does not make sense.

P2.L42 – use of “which” makes this a run on sentence. Consider breaking up these statements.

P2.L44 – consider improving the word usage of the statement beginning with “which”

P2.L48 – not all methods are based on liver stiffness.

P2.L49 – do you mean “MRI” apparent diffusion coefficient?

P2.L50 – PDFF is not used for fibrosis detection. It’s a fat quantification technique

P2.L51 – many of these techniques do provide information about disease heterogeneity.

P2.L52 – you do not need to re-state “non-invasive” as a pre-requisite for a non-invasive tool. The pre-requisite is accurate and precise staging, to allow detection and monitoring of disease and therapy response (which implies frequent and safe studies)

P2.L54 – Not sure what is meant by “deep tissue penetration”. While high resolution can be achieved with MRI, it is lower than CT, so not really a clear advantage. Same goes with “full liver coverage”.

P2.L55 – improved “detection” is not a direct consequence of MRI being non-ionizing.

P2.L57 – not only because of “limitation of currently available contrast agents”. There are other reasons too.

P3.L63 – That Gd only has positive (T1) contrast is not entirely true. The same contrast agents are used for dynamic susceptibility weighted (T2*) contrast (negative contrast). It merely depends on the pulse sequence used.

P3.L65 – Needs more clarity. Most Gd agents distribute to the liver, albeit extracellularly. Moreover, relaxivity is static for a given field strength and tissue, so an increase in this parameter is not the cause of bright T1 signal. Also, “short arterial and short hep uptake” is confusing.

P3.L70 – clarity: “underscore the pressing to”?

P3.L71 – are the authors proposing a dual imaging protocol for fibrosis detection? I think it is sufficient to propose the exploration of both imaging mechanisms of this protein contrast agent. But how do you know which is better?

P3.L76 – do you mean 1.4T?

P3.L78 – word usage: “Besides”

Results

P4.L86 – Judging by the references in this section, much of the methodological description of design seems based on prior publications, not an original result.

P4.L87 – stability and blood retention time is not referenced in Fig 1d.

P5.L105 – once again, this result seems based on a prior publication (ref 27,28), not original to this paper.

P5.L108 – what is the difference between the expressions “per Gd” vs “per particle”?

P6.L139 – Is detection based on heightened relaxivity or collagen binding affinity? I would think the latter.

P6.L139 – suffice to say R1 maps (no need to include inversion recovery)

P6.L139 – what is the rationale for collecting at 24hrs? Was it arbitrary?

P6.L140 – how was a dose of 0.02mmol/kg determined? This was ~20x what’s stated in Fig 1e. Was it the same for Eovist?

P6.L142 – a change in R1 of 0.78 is not “dramatic”. R1=2.4 is T1=416ms, which is not “dramatic”.

P6.L144 – was mouse fibrosis uniform throughout liver? If not, there may be an internal control of fibrosis degree.

P6.L148 – why is T2 maps shown and not R2maps (Fig 2b)?

Figure 2 – figure has 14 panels, which is extensive in my view. You may want smaller figures for each section.

Figure 2a – not a tremendous visual difference between Ishak 5 and 3 post-contrast images. Is delta R1 averaged voxel-wise across the entire liver? Pre-contrast R1 looks variable between grades too.

Figure 2g – appearance between Ishak 1 and 5 look very similar pre and post. Difference seems very dependent on ROI placement.

P7.L153 – Eovist is nearly completely eliminated by 24 hrs (and will only “enhance” functioning liver cells). Not an essential comparison.

P10.L211 – again, was fibrosis uniform throughout liver?

P10.L212 – was distribution measures and pathology results taken at 24 or 48hrs post-injection? Please state.

P10.L219 – run-on sentence; consider modifying.

P11.L222 – what does AUC of signal enhancement signify?

P11.L224 – distinguished by r1 and r2 (relaxivity), or AUC differences?

Figure 2l – why are %signal enhancement (PSE) plot labeled “Normal R1”, etc? PSE implies MR signal intensity, not R1 (which is a parametric measurement).

P11.L227 – missing the word “late” before “stage”?

P11.L227 – ROC analysis result: shouldn’t this be placed in the previous section, “Robust detection of early and late stages liver fibrosis and NASH with dual contrast property”?

P11.L237 – why does early fibrosis R1 increase at 24hr relative to 3hr (while late fibrosis decreases)? What's the rationale or mechanism? Are you possibly missing the peak R1 change of early fibrosis?

P11.L240 – are these 3hr findings specific to this time point only, or even earlier time points as well? Please describe the mechanism of these results (possibly in the discussion section) in terms of the pharmacokinetics of the agent. This “suggestion” from these results is not clear to me.

Figure 3f – the existence of portal hypertension in cirrhotic livers is common, and not related to the “injection of ProCA32.collagen1”. I can not piece together how the presence of a contrast agent bound to collagen in liver is indicative to the pressure of blood flow through the portal vein.

Figure 3a – some areas of “normal” liver seem to have quite high R1 values (similar to early and late fibrosis) at 3hrs. Why is that? What does it indicate?

Figure 3c and d – NASH liver (Ishak 5) is reported at $R1 \sim 4s^{-1}$ at 3hr (Fig 3d), but images on Fig 3c look less “red” than fibrotic liver (Ishak 5) on Fig 3a. The color scale on Fig 3c also seems off ($R1=4s^{-1}$ should be very dark red, which isn't seen on NASH images)

P14.L269 – I don't follow how the “targeting ability” of the contrast agent reflects the existence of portal hypertension. Is it purely based on slow wash out? Then it's not really due to the targeting ability.

P14.L270 – there are several claims in this sentence. You may want to be more concise, otherwise it gets confusing. Is it the time-dependent R1 contrast profiles that distinguish fibrosis stage, or delta R1 (and R2) at 24hrs?

P14.L272 – Suggesting the contrast agent reveals patterns in cirrhotic liver may be true, but it's a qualitative assessment, which may require a separate investigation. It is valid to speculate on the distribution differences in each disease model in the discussion section.

P14.L278 – Do all mice in the DEN group exhibit this difference in CPA (12% vs. 6.7%)? Is this considered mild, moderate, or severe fibrosis?

P14.L279 – Why was T1, T2, and T1 IR performed here, and not R1 and R2 mapping (consistent with earlier analysis)?

Figure 4a – It is not obvious where right and left liver segments are divided. This may make enhancement differences clearer to the reader. T1, T2, and T1 IR are very different imaging techniques; yet the colorscale in figure is identical. Are the intensities comparable between images shown? Please state how scaling was performed or include a colorbar.

Figure 4c – How is “contrast to noise ratio” defined? Both T1 and T2-weighted images seem to show positive signal enhancement. Shouldn't you see stronger negative signal between segments on T2 using a very high $r2$ contrast agent?

Figure 4d – need color bar scale for R1. Also should show right vs. left segment R1.

P16.L311 – please revise this sentence; it is unclear. There are many uses of the word “or” that makes the claims confusing.

P16.L313 – why was this analysis of enhancement over time only performed with the DEN model, and not other mouse models.

Figure 4fg – what do the two color bars on one image signify? Are you showing both 3 and 24hrs in one image? Also, the colorbar associated with “maintained” and “washout” is unclear... dark red indicates >100% relative enhancement, yet it's defined as “maintained”? The color matrix is somewhat unclear to me.

P17.L316 – from Fig4c, it seems T2w has higher sensitivity for heterogeneity (difference b/w left-right segments?) than IR. Please define how you are determining sensitivity. How are you quantifying which method (T1 or T2 or IR) is more “sensitive”?

Discussion

P17.L323 – What were the data results for correlation with histology, CPA, and Gd3+?

P17.L324 – Is ICP-OES a common term? Not defined anywhere. Was Gd3+ detection at 48hrs?

P17.L325 – Is it due to 5x increase in relaxivity, or collagen binding affinity? I would think the latter, since the mechanism seems to be whether the CA is present in liver tissue or not, right?

P17.L326 – IR + long TE may need to be elaborated. Depending on inversion time, IR will enhance T1 differences, while long TE enhances T2 differences.

P17.L327 – It is not clear how the use of enhanced r2 properties, coupled with various imaging techniques (IR, T1 and T2-weighted) overcome small changes in “liver morphology”

P17.L328 – explain how “precision” is “doubled”. Precision of what? Does this mean some coefficient of variation metric is two-times lower by using this agent?

P17.L333 – Are these statements “likely due to...” an effect? Are they not definite?

P17.L336 – What is meant by “tissue penetration”? I don't see correlation values mentioned between CPA and delta R1 and R2. Is correlation based of visual inspection only?

P17.P339 – It is important to precisely define “dynamic molecular imaging (DMI)”. It's not an intuitive concept. You mention high “spatial and temporal resolution”, but what is considered high for DMI? 0, 3, 24, 48hrs seem very spaced out time points. It will be important to fully discuss how “early” (i.e. +3hrs) time points point to angiogenesis. Where is contrast being “retained” in this scenario vs. normal liver, and why is that indicative to new vessel formation.

P18.L346 – what is SEM? Acronym not defined.

P18.L350 – is your claim that a “dramatic” increase in R2 and R2 at 3hrs indicative of portal hypertension? Why? What increase is considered “dramatic”? Presumably, ProCA32collagen1 concentration in liver is high at 3hrs (vs. normal and early-stage). Doesn't this mean that contrast

agent wash-in and retention (collagen affinity) is greater in cirrhotics vs. early-stage? Do you suspect a more gradual CA uptake in early stage disease? Also, please explain why R1 and R2 appear to continue increasing in early-stage from 3 to 24 hrs.

P18.L352 – This is a one sentence paragraph and seems out of place.

P18.L358 – It sounds like you’re distinguishing DCE from DMI. Please explicitly state the difference, since it’s not obvious.

P18.L362 – The connection between improved “penetration capability” and the formation of “new vascular structures” is unclear. The agent binds to collagen, so how does this point to vascular information?

P18.L367 – Please state/disclose these precise pattern differences in the Results, and discuss them here. Are they consistent? Is there a metric to quantitatively determine pattern differences?

P19.L370 – This is a run-on sentence. Also, are you claiming that AUC₀₋₄₈ of both R1 and R2 is a robust metric to stage fibrosis? What are the threshold values these need to be to distinguish normal, early and late stage? Are these better than delta R1 and R2 at 24hrs for staging fibrosis?

P19.L376 – Some institutions have migrated from MultiHance to Prohance or Dotarem. Eovist is usually used in specialized cases (low overall usage), and is fairly stable.

Methods

P24.L509 – Is CPA calculated over the entire liver, individual slices, or just one segment? In other words, how big was the CPA sample? What was the CPA for the various mouse models?

P25.L522 – You state 0.02mmol/kg (Line 140). Is 5mM equivalent to that? Perhaps be consistent here.

P25.L523 – Please comment on the choice of inversion times. Do you feel there is enough T1 sampling resolution? Your results show that your R1 range is ~ 1.5 to $\sim 4s^{-1}$ (or $T1 \sim 250ms$ to $666ms$), which means the ideal null point is around 170 to 460ms. Also, were the inversion times acquired in separate acquisitions? Spin echo or gradient echo? What was the scan time? Similar questions for T2 mapping.

P25.L529 – What model was used for curve fitting T1 and T2? Was it performed pixel by pixel?

P25.L534 – What is meant by “intensity MRI images”? Are these the R1 and R2 maps? If so, these are somewhat quantitative, not qualitative.

P25.L536 – Not sure I follow. Are you saying each image voxel has a min and max R1 value?

P25.L537 – Please explain “seeding” over the “interval”. What is the interval?

P26.L542 – I assume T1, T2, and IR are different acquisitions than R1 and R2 mapping. What were the parameters? So, if I understand, at each time point (0, 3, 24, 48 hrs), T1, T2, IR, R1-mapping, and R2-mapping was acquired? Please state how many scans were performed at each time point, and how long each scan was?

P26.L544 – awkward sentence (“To make sure the enhanced area remained enhanced or the contrast agent is washed out...”)

P26.L547 – Not sure if you can conclude this; it depends on imaging technique. For T2-weighting, more negative contrast may indicate more T2 effect, hence greater agent concentration. A negative T1 contrast may also indicate increase concentration, since a T2 effect dominates. I can presume you may not have to worry about the latter case, since delta R2 were not extremely high, but you may want to acknowledge that this is possible.

P26.L549 – It's ok to formulate a unique analysis like this. But since it's challenging for readers to grasp initially, please summarize the overall results concisely (in the Results). What significance does it have?

P26.L560 – relaxivities, not “relaxation rates”. “GdCl₃ and protein (2:1)”: how does this mimic ProCa32collagen in vivo?

P26.L563 – How were Eovist and ProCA32collagen dilutions prepared for ex vivo measurements? In saline, plasma, serum? How was it different than the 1.4T preparation?

P27.L577 – euthanasia at 48hr post-injection?

P27.L583 – this stats section only deals with AUC analysis. Please re-state section heading.

P27.L584 – PSE was already defined and stated earlier in Methods (line 543).

P27.L586 – why only NASH model? How was AUC measured... trapezoidal rule?

P27.L587 – why wouldn't AUC_3-24, slope_3-24, or slope_0-3 show differences too? Were all possibilities tested to determine the appropriate metric?

P28.L591 – Most of the statements in this section are results, not methods.

P28.L598 – Were the values from an ROI of the entire liver? Please state ROI size and any segmentation routines (automatic or manual).

P28.L600 – some analysis was paired, correct? Such as R1 time course within the same animal model.

P28.L603 – why weren't the pharmacokinetic results included in the main document?

P28.L605 – how many sampling points? What is meant by n=3-6?

Supplemental Figures

Figure S4 – a. show colorbar scale on image; c. which tissues/values represent CNR calculation?
Normal vs. cirrhotic? Left vs. right? Which ROIs?

Figure S6 – Is clearance via the kidney? Also consider comparing Table c with current clinical agents from literature. I suggest including pharmacokinetic results in the main document.

Dr. Jenny J. Yang
Regents Professor of Biochemistry and Biophysics
Associate Director, Center for
Diagnostics and Therapeutics
Department of Chemistry
Georgia State University
P.O. Box 4098
Atlanta, Georgia 30302-4098
February 18, 2019

Reviewer #1

General comments: *The use of Gd-containing proteins as efficient MRI CAs has already been reported by the same group. Different proteins have been tested finally selecting Parvalbumin (rat, ProCA-30 and humanized, ProCa-32). Gd-containing proteins have been shown to act as good reporters in several targeted applications (1)Pu, Fan, et al. "GRPR-targeted protein contrast agents for molecular imaging of receptor expression in cancers by MRI." Scientific reports 5 (2015): 16214. 2)Pu, Fan, Shenghui Xue, and Jenny J. Yang. "ProCA1. GRPR: a new imaging agent in cancer detection." (2016): 449-452. 3)Pu, Fan, et al. "Prostate-specific membrane antigen targeted protein contrast agents for molecular imaging of prostate cancer by MRI." Nanoscale 8.25 (2016): 12668-12682.4)Qiao, Jingjuan, et al. "Molecular imaging of EGFR/HER2 cancer biomarkers by protein MRI contrast agents." JBIC Journal of Biological Inorganic Chemistry 19.2 (2014): 259-270.). The collagen targeted MRI CA herein reported used as targeting vector the peptide previously reported by Caravan et al. Overall the work is carefully done and the conclusions look sound. Below specific comments are reported.*

Specific Comment 1: *33 the Title (INTRODUCTION) is missing*

Response: Corrected (line 34).

Specific Comment 2: *68 Gd-DTPA appears*

Response: Corrected (line 72).

Specific Comment 3: *69 remove the comma after "while"*

Response: Corrected (line 73).

Specific Comment 4: *74 a complete description of the product should be given including the nature of the protein and its origin (human or animal)*

Response: It was addressed in the manuscript (line 78).

Specific Comment 5: 85 lysine residues are probably intended as anchor point for PEG, not for collagen binding (consider revising)

Response: The sentence was revised in the manuscript according to the Reviewer's suggestion (line 88).

Specific Comment 6: 89 A deeper analytical characterization should be reported, e.g., calculated MW, Size-exclusion chromatography, SDS-PAGE, etc. For the assessment of Gd content, it is not clear how the protein content is determined. Possibly the correct ratio is 2:1 Gd-molecule (not 1:2)

Response: A complete analytical characterization of the protein including the calculated MW using Electrospray ionization (ESI) mass spectrometry, ion exchange chromatography for protein purification, SDS-PAGE showing the expression of the protein and SDS page demonstrating the purified protein band were all added to the manuscript in Supplementary Materials Figure S1. The ratio is in fact 2:1 for Gd-ProCA32.collagen1 complex and this mistake was corrected in the manuscript (line 92). Metals in ProCA32.collagen1 were removed by chelex-100 and metal content in ProCA32.collagen1 was analyzed by ICP-OES. The protein concentration was analyzed by UV-Vis spectrometry by monitoring Trp signal and obtaining the extinction coefficient from the protein sequence (line 445-447).

Specific Comment 7: 90-91 The dose in Fig 1e (0.0013 mmol/kg) is purely speculative; it should not be presented together with experimental data. Moreover, the conversion does not seem to be correct. Considering the HED factor of 12.3 between mouse and man, the adopted dose 0.02 mmol/kg corresponds to 0.0016 mmol/kg

Responses: Thank you for your correction. We have removed injection dosage in the column in Fig. 1e to avoid the presentation of speculative data according to your suggestion.

Specific Comment 8: 94 and Fig. 1b not clear what Lysine-ProCA32.collagen1 is, nor if there is a difference between Lysine PEGylated ProCA32.collagen1 and ProCA32.collagen1. A consistent terminology should be used throughout the paper. See also in Fig. 1d-1e, where PEG-ProCA32.collagen1 appears to be different from ProCA32.collagen1. Neither ProCA32-P40 is clearly described.

Response: As suggested, a consistent terminology was used throughout the paper. Wherever ProCA32 and ProCA32.collagen1 names are mentioned, they are referring to PEGylated contrast agents on Lysine residues. If "ProCA32.collagen1" is not PEGylated on Lysine residues it is referred to as "Non-PEGylated ProCA32.collagen1". These names are edited in figures legends and text for further clarifications.

Specific Comment 9: 96 Check "Ominiscan" instead of Omniscan (also in Fig.1c). Experimental details and methods used to obtain Fig 1c are missing. The methods in "Laurent S, Vander Elst L, Henoumont C, & Muller RN. (2010). How to measure the transmetallation of a gadolinium complex. Contrast media & molecular imaging, 5(6), 305-308." should be referred to.

Response: The mistake “Ominiscan” was corrected in the manuscript (line 106). Experimental details and methods used to obtain the Transmetallation studies in Fig. 1c have been added to Supplementary Materials (line 546-552) and as suggested, the above reference was cited (Reference 63).

***Specific Comment 10:** 97 the tested product (ProCA32.collagen1) has a lower than reported Gd content (see line 89). Why? Is there a difference in Gd affinity for the two chelating sites? All results should refer to the same batch, after full characterization. 100-101 See above, 90-91 101.*

Response: All experiments in this manuscript was performed with 2:1 Gd-ProCA32.collagen1 complex except the transmetallation studies which were performed with a 1:1 Gd-ProCA32.collagen1 complex. This ratio was used to demonstrate a more meaningful comparison with clinical contrast agents since in all of them Gd³⁺ is loaded in a 1:1 ratio with chelators. However, as suggested we performed the transmetallation experiment with 2:1 Gd-ProCA32.collagen1 complex as well and the results are shown in Fig. 1c. Due to cooperative binding, there is no difference in affinity between two binding sites.

***Specific Comment 11:** 114-117 check “raveled”*

Response: The mistake was corrected in the manuscript.

***Specific Comment 12:** The measured hydration number of 0.5 for the Tb-containing system may not be transferable to Gd-containing protein*

Response: We have determined water coordination number in ProCA32.collagen1 by Terbium Lifetime Luminescence. The number of coordination water molecule in the inner sphere of Gd³⁺-ProCA32.collagen1 was determined by the difference in Tb³⁺ luminescence decay between H₂O and D₂O. This system was used since both Gd³⁺ and Tb³⁺ have very similar coordination chemistry, and the Tb³⁺ system is commonly used for determining the water hydration number in clinical contrast agents such as Gd-DTPA and Gd³⁺ chelators. Below are the relevant, associated references including our publication used to measure the water number:

1. Chang CA, Brittain HG, Telsler J, Tweedle MF (1990) pH Dependence of relaxivities and hydration numbers of gadolinium (III) complexes of linear amino carboxylates. Inorg Chem 29: 4468
2. Zhang X, Chang CA, Brittain HG, Tweedle MF (1992) pH Dependence of relaxivities and hydration numbers of gadolinium (III) complexes of macrocyclic aminocarboxylates. Inorg Chem 31: 5597
3. Shenghui Xue et al, Protein MRI contrast agent with unprecedented metal selectivity and sensitivity for liver cancer imaging Proc Natl Acad Sci U S A. 2015 May 26;112(21):6607-12.
4. Ronald M. Supkowski a, William DeW. Horrocks, Jr. b, On the determination of the number of water molecules, q, coordinated to europium(III) ions in solution from luminescence decay lifetimes, Inorganica Chimica Acta 340 (2002) 44-48

Specific Comment 13: 116-117 *A unique and consistent terminology should be defined and used throughout the paper*

Response: We have carefully revised the manuscript to ensure that a unique and consistent terminology was defined and used throughout the paper.

Specific Comment 14: 125 *This sentence is not correct, since ProCA32.collagen1 is not among clinical GBCAs. Rewrite as "...ProCA32.collagen1 has a greater metal selectivity for Gd³⁺ than clinical gadolinium-based contrast agents."*

Response: The sentence was rewritten according to Reviewer's suggestion (line 130, 416).

Specific Comment 15: 128-129 *Actually from Fig 1f one can see that the main protein band is decreasing and possibly some lower MW bands appear. The image should reproduce the whole migration lanes, in order to allow the reader to assess the stability of the main product.*

Response: As suggested, the whole SDS page with Ponceau S staining was shown in the manuscript (Fig. S4b) to more clearly demonstrate the stability of the contrast agent.

Specific Comment 16: 130-134 *These sentences cannot be evaluated without Supplementary materials and related study methods.*

Response: We have carefully rewritten these sentences to accurately represent our Supplementary Data (line 137).

Specific Comment 17: 135-136 *The comment of possible reduced dosage does not pertain to the table of Fig. 1e. Possibly to the Relaxivity table (1d), but it is not clear on which data the ratio is calculated*

Response: The reduced dosage was referred to projected human dosage, however, the injection dosage data were removed from Fig. 1e as suggested previously and the possible reduced dosage was attributed to higher relaxivity in the manuscript. Human equivalent dose (HED) was calculated using the equation below:

$$\text{HED (mg/kg)} = \text{Animal dose (mg/kg)} \times (\text{Animal } K_m / \text{Human } K_m)$$

As the K_m factor for each species is constant, the K_m ratio is used to simplify calculations. Hence, equation above is modified as:

$$\text{HED (mg/kg)} = \text{Animal dose (mg/kg)} \times K_m \text{ ratio} = 0.02 \times 3/37 = 0.0016 \text{ mmol/kg}$$

Specific Comment 18: 141 *Was the dose of Eovist also 0.02 mmol/kg? The standard dose, after correction for body surface, is 0.3 mmol/kg. This dose would compensate for the lower relaxivity of Eovist and yield a more meaningful comparison.*

Response: To better compare the MRI and Gd³⁺ accumulation results of the two contrast agents, the injection dosage of Eovist used was the same as ProCA32.collagen1 (5 mL, 100 µL). If we used higher injection dosage for Eovist (0.3 mmol/kg), we could not compare the Gd³⁺ accumulation results in other organs which would have been possibly 15-fold higher.

Specific Comment 19: 166-207 Figure 2 has > 50 panels, conveying confused and unreadable information. The legends are little useful

Response: We have divided original Fig. 2 into 3 new Figures, 2, 3 and 6, according to animal models and statistical data to eliminate confusion and unreadable information. The figure legends were also rewritten to increase clarity.

Specific Comment 20: 248-267 Eovist is tested only in animals with Ishak =1 or =3, never with Ishak =5, which are the only cases in Fig. 3a-d where the response to ProCA32.collagen1 is significantly higher. Therefore a comparison is not possible

Response: The purpose of testing Eovist in animals with Ishak =1 or =3, was to demonstrate the inability of Eovist to detect early stage liver fibrosis and NASH and these data were compared with animal with Ishak =1 or =3 that were injected with ProCA32.collagen1.

Specific Comment 21: 309 T1 inversion

Response: Corrected (line 287).

Specific Comment 22: 318 Discussion (not Dissuasion)

Response: Corrected (line 330).

Specific Comment 23: 328 “precision is doubled” where is it shown?

Response: To avoid any confusion, the sentence was removed from the manuscript. It was meant to show that using multiple imaging sequences provide multiple ways to eliminate MRI artifacts associated with heterogeneous liver background (i.e. false positives) and increase accuracy (line 359).

Specific Comment 24: 330-332 From previous description (see 105-106), it seemed that ProCA32-P40 contains PEG, differing only in the lack of the collagen binding peptide. A clearer compound identification should be provided. The contribution of PEG is expected to influence tR , more than second sphere water molecules

Response: In order to avoid any confusion and for clarify, we used “ProCA32” instead of “ProCA32-P40” in the manuscript. As suggested, we modified the manuscript to reflect the contribution of PEG in influencing tR , rather than second sphere water molecules (line 332).

Specific Comment 25: 380-381 See above 130-134. Comparison with other agents should imply the same administration protocol and analytical methods

Response: We indeed applied the same administration protocol of tail vein injection and analytical methods such as organ digestion and ICP-OES to measure the Gd^{3+} concentration, biodistribution and organ deposition for both ProCA32.collagen1 and Eovist. Different metal binding assays were required to determine metal binding affinity due to special nature of proteins and small chelators

since classic pH potentiometric method is not applicable to protein, ProCA32.collagen1 Also, because pH titration is not suitable for determining K_d between proteins and metals, we were not able to compare them at the same conditions. Also, due to the lack of fluorescence signal, the Tb-buffer system cannot be applied to small chelators to determine binding constants. Gd-Tb competition can also not be applied to get the affinity of Gd^{3+} for small molecules.

However, we applied the same transmetallation method to directly compare their kinetic stability against endogenous zinc replacement or “inertness” (See Fig 1c). Clearly, ProCA32.collagen1 has superior metal stability and selectivity for Gd^{3+} over Zn^{2+} that is consistent with our metal binding assay.

Specific Comment 26: 382 “molecular biomarkers” too generic, better “collagen”

Response: As suggested, “Collagen” was used instead of “molecular biomarkers” (line 346).

Specific Comment 27: 384 See above: the dosage in humans is speculative

Response: As suggested, the data regarding the projected dosage of ProCA32.collagen1 in humans were removed from the manuscript.

Specific Comment 28: 402 i.p. injection? delete

Response: “i.p. injection” was deleted.

Specific Comment 29: 404 The TAA dose (not concentration) is not well defined and not consistent with above 200 mg/kg

Response: The TAA dose used was 200 mg/kg and this was corrected in the manuscript (line 453).

Specific Comment 30: 500 How was the antibody prepared? Is a rabbit antiserum?

Response: The self-generated antibody was purified from rabbit antiserum using PEGylated-ProCA32 as the immunogen.

Specific Comment 31: 502-3 The COL-1 antibody is from mouse. The corresponding secondary Ab is not reported

Response: Goat anti-Mouse IgG1 Cross-Adsorbed Alexa Fluor 488 was used as secondary antibody. The information was added to the “Methods” section, “Immunofluorescence Imaging” (line 560-562).

Reviewer #2

This work describes a novel imaging approach with an innovative MRI contrast agent to visualize fibrosis in mouse models.

Specific Comment 1: *I have a series of concerns regarding the animal models of NASH/ASH/fibrosis used in this manuscript. Throughout the manuscript, it remains quite blurry, which models are presented (“early fibrosis”, “late fibrosis”). The figure legends need to be clear on these aspects (e.g., figure 2). Moreover, it is simply not true that a single injection of DEN in c57bl/6 mice would induce liver cirrhosis after 10 months. This is a non-fibrotic / non-cirrhotic model of HCC. If the authors observe cirrhosis, we are likely looking at artifacts related to tumor development (e.g., pathogenic angiogenesis, leading to Sirius Red stained areas). Same is true for “NASH diet induced cirrhosis” – this will simply not happen after 6 months of Western diet.*

Response: This comment has several sub comments which are addressed point-by point below:

I have a series of concerns regarding the animal models of NASH/ASH/fibrosis used in this manuscript.

Reponses: No details are mentioned by the Reviewer to justify concern for the ASH model.

Throughout the manuscript, it remains quite blurry, which models are presented (“early fibrosis”, “late fibrosis”). The figure legends need to be clear on these aspects (e.g., figure 2).

Response: We have carefully rearranged figures and the legends in each figure were rewritten and shortened to clarify the data presented in each figure.

Moreover, it is simply not true that a single injection of DEN in c57bl/6 mice would induce liver cirrhosis after 10 months. This is a non-fibrotic / non-cirrhotic model of HCC. If the authors observe cirrhosis, we are likely looking at artifacts related to tumor development (e.g., pathogenic angiogenesis, leading to Sirius Red stained areas).

Response: We defined “Cirrhosis” as “a diffuse process characterized by fibrosis and the conversion of normal liver architecture into structurally abnormal nodules that affect the whole organ based on the original published definitions. Fibrosis is defined as the presence of excess collagen due to new fiber formation that causes only minor clinical symptoms or disturbance of liver cell function”. Thus, throughout the manuscript, we include histological analysis of collagen content to verify the degree of fibrosis in every animal model we used (Fig. 2f, 3f, 3g, 5b). References below used for these definitions:

1. Anthony PP et al, Bull World Health Organ. 1977; 55(4): 521–540.

2. Christian Liedtke et al, Fibrogenesis Tissue Repair. 2013; 6: 19.

Figure I. The entire slide scan of DEN-treated mouse liver stained by Sirius red (black arrows) shows heterogeneous accumulation of collagen across the liver (right lobe), as well as intra tumor crossing-networking Sirius Red stains representing collagens.

The main purpose here is to use a model to mimic cirrhosis conditions and associated collagen heterogeneity. To provide further clarity, we have replaced the term “cirrhosis” by “collagen heterogeneity” in the related section) in the revised manuscript for the DEN-induced model (line 249. For the DEN mice model, our histological results (Fig. 5b, Fig. I) show that there is extensive accumulation and deposition of collagen in the tumor region of the tissue and the entire heterogeneous distribution of collagen in the liver can be detected by our developed contrast agent, which is a key application in field of hepatology. We have also provided the entire scanned slide in Fig I which highlights the intra tumor collagen accumulation shown in red by Sirius red stain. In addition, the Sirius Red stains are typical collagen stains (typical crossing-networking pattern), not artifact of pathological angiogenesis that has no crossing-networking stain pattern.

Same is true for “NASH diet induced cirrhosis” – this will simply not happen after 6 months of Western diet.

Response: It is not true that NASH diet was used to induce cirrhosis in normal, wild type mice. Instead, we have clearly stated in the “Methods” section (line 472), that liver-specific Comparative Gene Identification-58 (CGI-58) knocked out (LivKO) mice were used which develop severe hepatic cirrhosis even on chow diet (reference 1) and the NASH diet facilitates the development of hepatic fibrosis to develop cirrhosis conditions. In previous publications (Feng Guo et al, Journal of Lipid Research, Volume 54, 2013; J. Mark Brown et al, Journal of Lipid Research Volume 51, 2010), it has been clearly demonstrated that CGI-58 deficiency in the liver directly causes not only hepatic steatosis but also steatohepatitis and fibrosis in mice. Our histological results (Fig. 3g) confirmed by pathologist, Dr. Alton Brad Farris (Emory University), clearly demonstrate that the knockdown of CGI-58 gene is responsible for development of cirrhosis. While we agree that NASH-diet alone cannot create liver cirrhosis, in our reported study here we use CGI KO mice with NASH diet that will create cirrhosis as verified by histological studies and our previous publications.

NASH-diet alone only causes mild hepatic fibrosis at the time point used here. Furthermore, we have successfully established an animal model of hepatic fibrosis after such a short duration of NASH diet treatment, and the contrast agent was able to detect it in this animal model. The literature describing this model is shown below:

1. Feng Guo et al, Journal of Lipid Research, (54), 2013; 2109-2573

Furthermore, it is worth mentioning that, we have successfully developed an early stage NASH model in normal, wild type mice that is verified by histology results. Our developed contrast agent is able to detect it in this animal model.

In addition, “Western diet” was not used. We did not use just Western diet to induce fibrosis. Instead, we used Western diet with fructose in drinking water (line 470) which is called NASH diet. NASH diet has been extensively reported to have much greater capability to induce fibrosis than just Western diet (see related references 2-4). The references below can explain the difference between diets:

2. Jennie Ka Ching Lau et al, J Pathol 2017; 241: 36–44

3. Samar H. Ibrahim et al, Dig Dis Sci. 61(5): 1325–1336, 2016

4. Jesse D. Riordan et al, Mamm Genome 25:473–486, 2014

Specific Comment 2: *No translational data about the applicability of this agent in humans (toxicity, dosing, distribution, specificity) are provided. At least some work with primary human cells / tissues confirming targeting specificity need to be provided.*

Response: The whole purpose of creating this MRI contrast agent is to develop future translational applications in humans, given the high translational potential of MRI. In this manuscript, we have provided extensive results to support potential human applicability of our developed contrast agents especially for reduced metal toxicity (Fig.1e, 1c, 1f, Supplemental 7d), reduced dosage, and biodistribution (Supplemental 5a). To address one of the major causes of nephrogenic systemic fibrosis (NSF) and brain deposition associated with free Gd³⁺ toxicity, we determined that the Gd³⁺ binding affinity of ProCA32.collagen1 is comparable to the approved clinical

contrast agents (**Fig. 1e**). ProCA32.collagen1 also exhibited 10^{14} -to 10^{16} -fold increases in metal selectivity (kinetic stability) for Gd^{3+} over Ca^{2+} and Zn^{2+} compared with all clinically approved contrast agents. To the best of our knowledge, ProCA32.collagen1 has the greatest metal selectivity among all other clinically approved Gd^{3+} -based contrast agents. In addition, the required injection dose is much lower than any clinical contrast agents.

We have cited and discussed references (Ref 12-14) in our manuscript to support human translation potential of collagen targeting. Collagen expression has been observed in several chronic human patient diseases including NASH and HCC (R A Standish et al, Gut 2006;55:569–578; Mette J. Nielsen et al, PLoS One. 2015; 10(9): e0137302; Ramón Bataller et al, J Clin Invest. 2005 Feb 1; 115(2): 209–218; Krishna Sumanth Nallagangula et al, Future Sci OA. 2018 Jan; 4(1): FSO250). As suggested, human HCC tissue samples were used (Fig II) and stained both with our contrast agent and Sirius red to demonstrate high collagen expression and ProCA32.collagen1 binding to collagen in human HCC samples. These results demonstrated the specificity of our contrast agent to bind to collagen in human HCC tissue arrays and the applicability of our contrast agent in humans. This figure has been added to our manuscript as Fig. 7c.

***Specific Comment 3:** The true challenge in clinical trials is monitoring treatment efficacy non-invasively. There is a large series of effective antifibrotic treatments in NAFLD under development that we effective in mouse models (e.g., elafibranor, selonsertib, cenicriviroc). The authors need to demonstrate that the imaging can be performed longitudinally and is capable of monitoring fibrosis regression (or reduced fibrosis development) in mouse models.*

Response: Our results in Fig. III in this cover letter show that our developed contrast agent is in fact capable of monitoring liver fibrosis regression and treatment. In order to test the efficacy response, cirrhotic mice generated from TAA/Alcohol model were treated with pirfenidone which has been tested previously (Oleksii Seniutkin et al, Toxicol Appl Pharmacol. 2018 Jan 15;339:1-9), and the mice were scanned before and after injection with the contrast agent, and the results were compared with normal mice.

Our data suggest that ProCA32.collagen1 can distinguish pirfenidone-treated liver from normal liver. We did not include these results in the manuscript since our goal here was to provide extensive results to demonstrate the novel capability of developed contrast agents for the “early detection and staging of liver diseases”. This is the essential step prerequisite to human clinical trials, to monitor treatment efficacy. In addition, we have shown that ProCA32.collagen1 can detect different stages of the disease in several models as you discussed here.

Specific Comment 4: *Minor: Fibrosis need to be assessed by additional methods, including hydroxyproline and collagen Western blot.*

Response: We have provided extensive assessment of fibrosis stage using Sirius Red, IHC staining and α SMA for each model (Fig 2f, 2j, 2k, 4b). The fibrosis stage was carefully analyzed by a pathologist, Dr. Alton Brad Farris, Emory University and statisticians Dr. Xiaoyin Meng and Dr. Gesheng Qin. As suggested, we have added additional results for hydroxyproline content for our TAA/Alcohol model in Supplementary Materials in Fig S16.

Figure III. A. R1 map images of normal vs Pirfenidone treated liver before and 3 and 24 hrs post injection of ProCA32.collagen1. **B.** Quantitative analysis of $\Delta R1$ and percentage of R1 increase rate 3 and 24 hrs post injection of ProCA32.collagen1 compared to Prescan. **C.** R1 changes of normal and drug treated liver over different MRI time points after injection of ProCA32.collagen1. **D.** Liver weights of the animals at end point of experiments and at end of fibrosis induction or after Pirfenidone treatment or buffer treatment. **E.** Liver enzyme levels after treatments with buffer (PBS), normal liver and Pirfenidone treated liver. **F.** Sirius red stains of livers treated in different conditions. **G.** Percentage of Sirius red stained areas in livers treated with buffer (PBS, cirrhosis), normal liver and Pirfenidone.

Reviewer #3

Comment 1: Please be advised that the manuscript (“Article”) format does not abide by Nature recommendations:

- Abstract >150 words
- Overall Text >3000 words
- Methods >3000 words
- References > 50
- Large number of sub-panels per figure (as many as 10 sub-panels occur in figure 3). Legends are very long >500 words

Response: As per the Editor’s suggestion, Dr. Michael Basson, we have transferred our manuscript from Nature Medicine Letter to Nature Communications. The guidelines of Nature Medicine Letter limited the number of figures to four. We have rearranged the figures and added new figure numbers to improve the readability of the manuscript and understanding the key findings. We have divided original Fig. 2 into 3 new Figures, 2, 3 and 6, according to animal models and statistical data to eliminate confusion and unreadable information. The figure legends were also rewritten to increase clarity.

General Comments: *The manuscript describes a novel protein-based MR contrast agent that is specific to collagen 1. The motivation is to develop an MRI-sensitive marker for staging liver fibrosis. The research topic has important implication, since many current MRI methods lack the sensitivity to detect early fibrotic changes in a quantitative manner. A main conclusion is that the change in MR relaxation rates (pre to post injection) at 24hrs allows the ability to distinguish normal, early stage, and late stage fibrosis, while an earlier time point (3hrs post-injection) is able reveal facets of angiogenesis and portal hypertension, indicative of more advanced disease.*

Comment 2: *Results and Claims: In my opinion, there is an excess of data and claims presented in this manuscript, which distracts from the overall objective and my understanding of the essential findings. I believe it is longer than it needs to be, and can be written more concisely. Several claims are given in the manuscript that point to novel methods for detecting and staging fibrosis, but it is never clear which finding is preferred or advantageous in a clinical setting. I feel there needs to be a more specific and in-depth analysis of 1-2 primary claims the authors feel most strongly about. Moreover, from the manuscript, it is difficult to piece together a singular procedure for using ProCA32collagen1 and an MR method in clinical application. Ultimately, my question is still, “Once injected, what single MR method should I perform, and when? And what primary data analysis method should I perform to best elucidate fibrosis stage?” If this still needs investigation, you should state the need for further work.*

Response: We have completely revised and rewritten some parts of the manuscript, in order to emphasize on the key findings and improve readability. Heterogeneous MR signals coming from fibrosis liver and tissue background are the major challenge and cause of low specificity of MRI with current clinical approved contrast agents in morphology detection. In addition to lack of disease molecular biomarker detection, all approved Gd^{3+} contrast agents have very small r_1 and r_2 relaxation values. Only r_1 contrast is able to provide some image enhancement in vivo. Thus, all clinical imaging method is built only on one single imaging technique, which leads to low

specificity in differentiation of fibrosis vs tissue background and inability to detect early-stage fibrosis.

Our developed contrast agent has several unique properties including dual relaxation and collagen binding. One of the major advantages of using our developed contrast agent, is that you can use multiple imaging pulse sequences and imaging methodology with advantages in early detection and staging fibrosis *in vivo* instead of one single method. By using only one single imaging technique, there is a high chance of imaging artifacts due to heterogeneous MR signal coming from liver background, the use of multiple imaging techniques provide complementary ways to validate and confirm the stage of the disease with significantly improved sensitivity and specificity. We have explicitly mentioned in our manuscript that both the T1 and T2 mapping 24 hrs time point post-injection of the contrast agent is used to stage the disease (normal vs early vs late) while the 3 h time point provides additional information such as portal hypertension and angiogenesis associated with late stage liver cirrhosis. Furthermore, we have shown a novel dynamic molecular imaging (DMI) methodology, taking into consideration the entire imaging response curve of T1 and T2 mapping, which provides the highest sensitivity and specificity for early detection and staging of fibrosis.

***Comment 3: Pharmacokinetics of ProCA32collagen1:** You should consider adding more pharmacokinetic data into the main document, and not in supplemental data. For a new agent, I think it's crucial to discuss distribution kinetics, elimination half-life, and clearance, in addition to relaxivity, stability, and safety. I'm curious to know the short time-scale wash-in/wash-out characteristics of this agent in both normal and fibrotic tissue environments. Given a high R1 for late stage fibrosis and low R1 for normals at 3hrs, does this imply faster wash-in/wash-out for normals? How long will ProCA32collagen1 bind to collagen markers in fibrotic livers before being eliminated?*

Response: Based on your suggestion, we have added the pharmacokinetic data in the main text (see Fig. 1f and 1g). The distribution of the contrast agent, elimination half-life, and clearance have all been reported in the manuscript in the supplemental information section: “Clearance of ProCA32.collagen1 was low at 0.36 mL/min/kg but more than 3 times higher than ProCA32-P40, and with very high exposure (>100000 ng.h/mL blood). ProCA32.collagen1 exhibited a terminal elimination half-life of 9.9 h, which was slightly higher than ProCA32-P40 (8.09 h), with a mean residence time of 14.5 h that was higher than 13.9 h of ProCA32-P40. In addition, volume of distribution (V_c) and volume of distribution at steady state (V_{dss}) were 1.53 and 1.77 L/kg, respectively, which were more than 2-7 times higher than ProCA32”.

We have performed more statistical analysis to demonstrate the uptake and washout rate of our contrast agent in both normal and different stages of fibrotic liver based on MR mapping data, and since R1 and R2 values will reflect the concentration of Gd^{3+} , they can be a good indication of uptake and washout of the contrast agent. Fig. IV (top) shows the overall uptake characteristics of this agent (0-48 hrs) in both normal and fibrotic tissue environments using our MRI data and percentage of signal enhancement in R1 map. Fig. IV (bottom) demonstrates the washout rate of the contrast agent between 3-48 hrs post-injection. The washout rate is: normal liver > early stage fibrosis > late stage cirrhosis. Uptake is as follows: late stage cirrhosis > early stage fibrosis > normal liver. We have added these data to our manuscript (see Fig 6c, and 6d). Based on AUC analysis in Fig. 6, after 48 hrs the majority of the contrast agent is washed out.

Comment 4: Pathology and Histology: The pathology work in this study is extensive and well-done. I am curious whether you analyzed any gross liver specimens from the mouse models. I think a valuable qualitative and quantitative comparison would be fibrosis seen on gross specimens versus slice-matched post-injection (24hrs) MRI R1 and R2 maps. This would truly validate contrast agent uptake, distribution, and wash-out in fibrotic and remote liver regions. I feel biopsy and histology proves the presence of fibrosis degree, but it is limited in emphasizing the geographic match between pathology and MR imaging data.

Response: We have compared the Gd^{3+} concentration in tissue (at 24 hrs time point) with Gd^{3+} concentration calculated from R1 map in MRI for both TAA/Alcohol and NASH diet models (see Fig. V). In addition, the MRI data presented in the manuscript main text is one slice, however the R1 and R2 values represent the mean values across all MRI slices as mentioned in the manuscript. Furthermore, the stage of fibrosis and collagen proportionate area calculations are all based on the entire slides from different locations of the mouse liver, not just a single slide.

Comment 5: *Time points: MR imaging of ProCA32collagen1 distribution was performed at 3, 24, and 48hrs, along with baseline measurements. Please state the logic for this procedure, especially in light of the agent’s half-life (~9hrs). Even though the use of animal models may preclude imaging at earlier time points, almost all current dynamic MR imaging occurs < 1hr post-injection. The use of the word “dynamic” (e.g DMI and DCE) should be tempered when considering 3, 24, and 48hrs post-injection time-points; this word implies imaging over a short time scale, usually over minutes. The authors should at least address the possibility of missing the peak signal or max R1 time points, which may affect AUC and other results presented in Fig 3 and 4.*

Response: It is important to note that ProCA32.collagen1 has much greater size than clinically approved contrast agents of small chelators with much longer half-life in blood (~10hrs). Thus, we performed the MRI scans at longer time points (longer than 3 hrs) since the half-life in liver tissues is likely to be longer than in the blood.

Dynamic Contrast Enhanced (DCE) imaging measures MRI signal changes such as T1 changes in tissues over time after bolus administration of gadolinium contrast agent. Dynamic refers to the

changes as a function of time. It does not imply “short time”. All clinical contrast agents that are based on small molecules (<1 kD in size) have much shorter half-lives (<10 min), and that is why the MR imaging occurs < 1 hrs post-injection for other MR imaging contrast agents. The short half-life of clinically approved contrast agents results in limitations in missing the peak signal and large errors in AUC calculation. Since our contrast agent has a half-life of ~9 hrs and has collagen binding capability, thus we tailored the acquisition time points to capture the entire curve without missing the peak signal or max R1 time points. We purposely use the new term “dynamic molecular imaging” (DMI) to avoid the confusion with DCE and highlight the targeting ability and collagen binding of our contrast agent with a new mechanism. Once the contrast agent is injected, liver will be enhanced along with the blood vessels due to the biodistribution of ProCA32.collagen1 in the blood vessel and sinusoid space in the liver. ProCA32.collagen1 gradually binds to collagen in the liver of diseased mice, which in turn prolong the half-time of contrast agent in the liver. Non-specifically distributed ProCA32.collagen1 was eliminated and ProCA32.collagen1 remained in the liver regions with overexpression of collagen.

This new dynamic process enhanced by our targeted contrast agent enables us to visualize differential washout rates based on the stage of the disease (see Fig. 6). This is consistent with heterogeneous MRI signal enhancement within 24 h post injection in Fig. 5d and 5e.

***Comment 6:** Methods: Some descriptions of experimental methods contain explicit data results. For example, the “statistical analysis” section describes the AUC results. Other method descriptions contain results shown in supplemental data figures. This may be ok, but I leave it to the Editors to determine whether these should be referenced in the text. In fact, the supplemental data contain lots of not referred to in main document.*

Response: We have carefully checked the experimental methods section and removed any results from this section. We made sure to only include description of methods used in the manuscript in this section. All methods descriptions were referred to in the main text wherever applicable.

***Comment 7:** From the histogram analysis of R1, R2, delta R1, and delta R2, there seems to be large variance in the images (Fig 3h, Fig 4e), yet very small error bars on plots. It is a bit unclear whether mean and standard deviation at each time point represents the entire imaging slice or volume. Further, there is no mention of a segmentation algorithm.*

Response: The signal was filtered to remove all the noise and artifacts. We performed manual segmentation to extract the liver from other organs in each slice and air, then the R1 and R2 values for entire liver in 12 image slices were measured for each voxel for both $\Delta R1$, and $\Delta R2$ calculations and histogram analysis.

***Comment 8:** Style: I also recommend a further review to improve readability, word usage, and some grammar. There are many statements and phrases that can be eliminated to shorten the manuscript, without losing effect. Furthermore, authors should refrain from using exaggerated language like “dramatic” and “strikingly” when describing results. Emphasis should be placed on whether results are statistically sound.*

Response: We have carefully rewritten some sections of the manuscript to shorten some parts and improve the readability as suggested. We have also removed any exaggerated language or wording in the text.

Abstract Comments:

Specific Comment 1: P1.L21 – do you mean “detection of liver fibrosis”? Other MR techniques are used to detect fatty liver disease and its heterogeneity

Response: We rephrased this sentence in the Abstract (line 23).

Introduction Comments:

Specific Comment 2: P2.L41 – “which is consisted of...” does not make sense.

Response: We rephrased this sentence in the Introduction (line 43).

Specific Comment 3: P2.L42 – use of “which” makes this a run on sentence. Consider breaking up these statements.

Response: We rephrased these statements (line 44).

Specific Comment 4: P2.L44 – consider improving the word usage of the statement beginning with “which”

Response: We rephrased this sentence (line 45).

Specific Comment 5: P2.L48 – not all methods are based on liver stiffness.

Response: We rephrased this sentence (line 53-54).

Specific Comment 6: P2.L49 – do you mean “MRI” apparent diffusion coefficient?

Response: The sentence was corrected: “ultrasound; MR apparent diffusion coefficient (ADC)” (line 54).

Specific Comment 7: P2.L50 – PDFF is not used for fibrosis detection. It’s a fat quantification technique

Response: We have corrected this statement by adding “early stage liver fibrosis and NASH” at the end of the sentence (line 56).

Specific Comment 8: P2.L51 – many of these techniques do provide information about disease heterogeneity.

Response: We have corrected this statement (line 56).

Specific Comment 9: P2.L52 – you do not need to re-state “non-invasive” as a pre-requisite for a non-invasive tool. The pre-requisite is accurate and precise staging, to allow detection and monitoring of disease and therapy response (which implies frequent and safe studies)

Response: We have corrected this statement (line 57).

Specific Comment 10: P2.L54 – Not sure what is meant by “deep tissue penetration”. While high resolution can be achieved with MRI, it is lower than CT, so not really a clear advantage. Same goes with “full liver coverage”.

Response: There is a difference between penetration or depth and resolution of an imaging modality. Depth of penetration, or the ability to observe deeper structures, is related to the frequency of the wave. Higher frequencies have less depth of penetration, while lower frequencies have greater depth of penetration. Shorter wavelengths or high frequency also give a better resolution, which is the ability to differentiate between two things. If the resolution is good, the picture will be clear, and any two objects will look like two objects. If the resolution is poor, the picture will be blurred, and the two objects will look like one.

High frequency: shorter wavelength, shorter penetration, good resolution

Low frequency: longer wavelength, longer penetration, poor resolution

The paper below can describe the differences in greater detail:

1. Paul J Cassidy and George K Radda et al, J R Soc Interface. 2005 Jun 22; 2(3): 133–144.

There are several papers and textbooks that refer to “deep tissue penetration” of MRI as an advantage:

2. Biomedical Nanomaterials Edited by Yuliang Zhao, and Youqing Shen, WILEY-VCH

3. Shizhu Chen et al, Eur. J. Nanomed. 2013.

MR techniques do offer the advantage of “full liver coverage”. Increasing the coverage of the liver volume is one of the main advantages in comparison to US elastography methods such as transient elastography (TE) and shear wave elastography (SWE). Please see references below:

4. Kenneth Coenegrachts, World J Radiol. 2009 Dec 31; 1(1): 72–85.

5. Gavin Low et al, World J Radiol. 2016 Jan 28; 8(1): 59–72.

Specific Comment 11: P2.L55 – improved “detection” is not a direct consequence of MRI being non-ionizing.

Response: We rephrased this statement (line 60).

***Specific Comment 12:** P2.L57 – not only because of “limitation of currently available contrast agents”. There are other reasons too.*

Response: We rephrased this statement (line 62).

***Specific Comment 13:** P3.L63 – That Gd only has positive (T1) contrast is not entirely true. The same contrast agents are used for dynamic susceptibility weighted (T2*) contrast (negative contrast). It merely depends on the pulse sequence used.*

Response: We rephrased this statement (line 66-67).

***Specific Comment 14:** P3.L65 – Needs more clarity. Most Gd agents distribute to the liver, albeit extracellularly. Moreover, relaxivity is static for a given field strength and tissue, so an increase in this parameter is not the cause of bright T1 signal. Also, “short arterial and short hep uptake” is confusing.*

Response: We rephrased and removed some parts of this statement for further clarity (line 67-69).

***Specific Comment 15:** P3.L70 – clarity: “underscore the pressing to”?*

Response: We rephrased this statement for further clarification (line 74).

***Specific Comment 16:** P3.L71 – are the authors proposing a dual imaging protocol for fibrosis detection? I think it is sufficient to propose the exploration of both imaging mechanisms of this protein contrast agent. But how do you know which is better?*

Response: Yes, we can use multiple imaging pulse sequences because this contrast agent has both r_1 and r_2 property. Using multiple imaging sequences provides multiple ways to eliminate MRI artifacts associated with heterogeneous liver background (i.e. false positives) and increase accuracy.

***Specific Comment 17:** P3.L76 – do you mean 1.4T?*

Response: Yes, we corrected the mistake (line 80).

***Specific Comment 18:** P3.L78 – word usage: “Besides”*

Response: We replaced the word as suggested (line 82).

Results Comments:

***Specific Comment 19:** P4.L86 – Judging by the references in this section, much of the methodological description of design seems based on prior publications, not an original result.*

Response: The targeted contrast agent was designed based on one of our previous publications (Reference 21) as we mentioned in our manuscript. Some of the methods used to characterize the designed contrast agent were used based on our previous publication.

Specific Comment 20: P4.L87 – stability and blood retention time is not referenced in Fig 1d.

Response: Reference 24 was used to mention the effect of PEGylation on stability, blood retention time and biocompatibilities in general. We removed “(Fig. 1d)” in the text. We added two references (29 and 30) to the table (Fig. 1d) for relaxivity of clinical contrast agents.

Specific Comment 21: P5.L105 – once again, this result seems based on a prior publication (ref 27,28), not original to this paper.

Response: This result is not based on prior publication. This is the original data reporting the binding affinity of ProCA32.collagen1 to collagen type I. References 26 and 27 are referring to the use of 1:1 binding model in fitting the data.

Specific Comment 22: P5.L108 – what is the difference between the expressions “per Gd” vs “per particle”?

Response: Contrast agents relaxivities, are typically expressed as either per Gd^{3+} or per particle, therefore, you can multiply the number of Gd^{3+} ions in a contrast agent to report this value per particle. For instance, if the relaxivity of a contrast agent is $20 \text{ mM}^{-1}\text{s}^{-1}$ per Gd^{3+} , and the contrast agent has two Gd^{3+} in its binding sites then the value per particle would be $40 \text{ mM}^{-1}\text{s}^{-1}$.

1. Zhuxian Zhou and Zheng-Rong Lu, Wiley Interdiscip Rev Nanomed Nanobiotechnol. 2013 Jan; 5(1): 1–18.

Specific Comment 23: P6.L139 – Is detection based on heightened relaxivity or collagen binding affinity? I would think the latter.

Response: The detection is based on both higher relaxivity and therefore higher sensitivity under MRI and binding to collagen I in the liver. However, upon binding there is a chance that the relaxivity of the contrast agent can increase even more which leads to even higher sensitivity for early stage detection. ProCA32.collagen1 binds to collagen in the liver, which extends its retention in the liver with much longer half-life in the liver compared with that of blood vessel. Once the contrast agent is injected, liver will be enhanced along with the blood vessels due to the biodistribution of ProCA32.collagen1 in the blood vessel and sinusoid space in the liver. ProCA32.collagen1 gradually binds to collagen in the liver of diseased mice, which extends/elongates the half-time of contrast agent in the liver. Collagen bound ProCA32.collagen1 remains in the liver regions.

Specific Comment 24: P6.L139 – suffice to say R1 maps (no need to include inversion recovery)

Response: We removed “inversion recovery” from the sentence (line 141).

Specific Comment 25: P6.L139 – what is the rationale for collecting at 24hrs? Was it arbitrary?

Response: This is the time point that the highest enhancement in liver is observed and targeting of the contrast agent occurs. Since our contrast agent has a half-life of ~9 hrs and has collagen binding capability, we tailored the acquisition time points to capture the entire curve without missing the peak signal.

Specific Comment 26: P6.L140 – how was a dose of 0.02mmol/kg determined? This was ~20x what's stated in Fig 1e. Was it the same for Eovist?

Response: This dose was calculated based on 100 μ L, 5 mM of ProCA32.collagen1 that was injected to mouse with weight of 25 g: $(5 \text{ mmol} \times 0.0001 \text{ L})/0.025 \text{ kg} = 0.02 \text{ mmol/kg}$. The dose mentioned in Fig. 1e was projected and speculated dosage that will be used in human (converted from mouse to human based on FDA guidelines), however due to the speculative nature of this value, we have removed it from Fig. 1e. Eovist dosage (0.02 mmol/kg) used in these experiments were kept the same as other contrast agents (ProCA32.collagen1 and ProCA32) for a more meaningful comparison.

Specific Comment 27: P6.L142 – a change in R1 of 0.78 is not “dramatic”. R1=2.4 is T1=416ms, which is not “dramatic”.

Response: We have removed the word “dramatic” from this sentence and some other sentences in the manuscript (line 143).

Specific Comment 28: P6.L144 – was mouse fibrosis uniform throughout liver? If not, there may be an internal control of fibrosis degree.

Response: The calculated CPA (Fig. S9) which is the percentage of Sirius red stained areas was based on the entire slide scan. The results are mean values across 10 different slides from different locations of mouse liver. In two of our models (TAA/Alcohol and NASH diet), fibrosis was relatively uniform across different slices of mouse livers. However, in the DEN model, collagen heterogeneity was observed in the first four slices of MRI, from which the first slice has been shown as representation (Fig 5a, 5d, 5e). The fibrosis stage and degree in all models was assessed and determined by pathologist, Dr. Alton Brad Farris (Emory University School of Medicine). In the DEN model, where there is collagen heterogeneity, the region that has the highest CPA and collagen will be considered to determine the final stage of fibrosis. Therefore, in DEN model, it was also determined that it is late stage cirrhosis.

Specific Comment 29: P6.L148 – why is T2 maps shown and not R2maps (Fig 2b)?

Response: T2 map and R2 map are basically representing the same information. $T2 \text{ map} = 1/R2 \text{ map}$.

Specific Comment 30: Figure 2 – figure has 14 panels, which is extensive in my view. You may want smaller figures for each section.

Response: As suggested, we reduced the number of panels in Fig. 2 and added new Figures for additional panels to improve the readability (line 156, 178, 316).

Specific Comment 31: *Figure 2a – not a tremendous visual difference between Ishak 5 and 3 post-contrast images. Is delta R1 averaged voxel-wise across the entire liver? Pre-contrast R1 looks variable between grades too.*

Response: Yes, the delta R1 was averaged voxel-wise across the entire liver. The observation that cirrhotic livers exhibit different R1 values compared to normal liver at pre-contrast has also been reported in several previous studies:

1. Rajarshi Banerjee et al, Journal of Hepatology 2014 vol. 60, 69–77.

2. Suk Keu Yeom et al, World J Hepatol. 2015 Aug 18; 7(17): 2069–2079.

3. Michael Haimerl et al, PLoS One. 2013; 8(12): e85658.

4. Suraj Sharma et al, World J Gastroenterol. 2014 Dec 7; 20(45): 16820–16830.

However, the main challenge in MR imaging is to distinguish early stage fibrosis from normal liver.

Specific Comment 32: *Figure 2g – appearance between Ishak 1 and 5 look very similar pre and post. Difference seems very dependent on ROI placement.*

Response: As mentioned previously, the delta R1 was averaged voxel-wise across the entire liver in 12 image slices to better represent these values for staging fibrosis. The ROI was selected to cover the entire liver and remove any other organs.

Specific Comment 33: *P7.L153 – Eovist is nearly completely eliminated by 24 hrs (and will only “enhance” functioning liver cells). Not an essential comparison.*

Response: As suggested, we have removed Eovist results in 24 hrs time point in Fig.2a and 3a, however, considering the short half-life of Eovist, we performed comparison at 30 min post injection as well (see Fig. 4a, 4c).

Specific Comment 34: *P10.L211 – again, was fibrosis uniform throughout liver?*

Response: In two of our models (TAA/Alcohol and NASH diet), fibrosis was uniform across different slices of mouse livers. The calculated CPA (Fig. S9) which is the percentage of Sirius red stained areas was based on the entire slide scan (line 580-582). The results are mean values across 10 different slides from different locations of mouse liver. However, in DEN model, collagen heterogeneity was observed in the first four slices of MRI which the first slice has been shown as representation (Fig 5a, 5d, 5e). The fibrosis stage and degree in all models was assessed and determined by pathologist, Dr. Alton Brad Farris Emory University School of Medicine). In DEN model, where there is collagen heterogeneity, the region that has the highest CPA and collagen will be considered to determine the final stage of fibrosis. Therefore, in DEN model, it was also determined that it is late stage cirrhosis. Please also see our Response to your comment 28.

Specific Comment 35: P10.L212 – was distribution measures and pathology results taken at 24 or 48hrs post-injection? Please state.

Response: Biodistribution results presented in Fig.S4a, Fig. 2d, e, and 3d, e and pathology results are performed at 24 hrs time point.

Specific Comment 36: P10.L219 – run-on sentence; consider modifying.

Response: We have modified the sentence as suggested (line 301-302).

Specific Comment 37: P11.L222 – what does AUC of signal enhancement signify?

Response: AUC (Area Under the Curve) demonstrates the uptake and washout of the probe at different time points. Positive regions (shown in pink) are referred to uptake and negative regions (shown in blue) are referenced to washout. R1 and R2 at different time points values were used to calculate the AUC.

Specific Comment 38: P11.L224 – distinguished by r_1 and r_2 (relaxivity), or AUC differences?

Response: Distinguishing different stages of NASH and fibrosis have been shown by AUC differences. AUC was calculated for both R1 and R2 curves. Since ProCA32.collagen1 has both r_1 and r_2 properties, therefore, AUC for both R1 and R2 can be calculated.

Specific Comment 39: Figure 2l – why are %signal enhancement (PSE) plot labeled “Normal R1”, etc? PSE implies MR signal intensity, not R1 (which is a parametric measurement).

Response: In Figures 6a, b, c, d, R1 and R2 values were measured, and percentage of signal enhancement refers to percentage of R1 and R2 increase or decrease in R1 and R2 maps. We corrected the mistake and modified the label for these figures. “Normal R1” refers to R1 value in normal mice. We have also corrected these statements in “Methods” section as well (line 647-649).

Specific Comment 40: P11.L227 – missing the word “late” before “stage”?

Response: We corrected the mistake and added “late” in the sentence (line 313).

Specific Comment 41: P11.L227 – ROC analysis result: shouldn't this be placed in the previous section, “Robust detection of early and late stages liver fibrosis and NASH with dual contrast property”?

Response: We respect your opinion and suggestion; however, we feel that these results from statistical analysis should be placed in “Assessment of ProCA32.collagen1 capability in early detection and staging chronic liver diseases” section since it demonstrates the contrast agent ability to stage the disease (line 301).

Specific Comment 42: *P11.L237 – why does early fibrosis R1 increase at 24hr relative to 3hr (while late fibrosis decreases)? What’s the rationale or mechanism? Are you possibly missing the peak R1 change of early fibrosis?*

Response: Since our contrast agent has a half-life of ~9 hrs and has collagen binding capability, we tailored the acquisition time points to capture the entire curve without missing the peak signal. Earlier time point of 3 hrs, is the time point when portal hypertension is observed and as a consequence, slow washout of the contrast agent occurs in late-stage fibrosis and that is why there is increase in R1 at 3 hrs time points in all animal models compared to 24-hrs time points. However, 24-hrs time point is the targeting phase of the contrast agent, and there is no portal hypertension observed in early-stage fibrosis, therefore the washout of the contrast is faster in early-stage fibrosis but ProCA32.collagen1 gradually binds to collagen in the liver of mice with early-stage fibrosis, which extends/elongates the half-time of contrast agent in the liver. Non-specifically distributed ProCA32.collagen1 is eliminated and ProCA32.collagen1 remains in the liver regions with overexpression of collagen.

As shown in Figs. 2a and 3a, at 24-hrs time point, the R1 of liver is still higher for late-stage fibrosis compared to early-stage which is an indication of targeting ability of the contrast agent and more collagen levels for the contrast agent to bind to as confirmed by histology analysis. The 24-hrs time point is too long to demonstrate any difference in washout rates, and based on the half-life of the contrast agent, 3-hrs time point was chosen to show the difference and the existence of portal hypertension in late-stage fibrosis.

Specific Comment 43: *P11.L240 – are these 3hr findings specific to this time point only, or even earlier time points as well? Please describe the mechanism of these results (possibly in the discussion section) in terms of the pharmacokinetics of the agent. This “suggestion” from these results is not clear to me.*

Response: The enhancement we observed at the 3 hrs point for late stage cirrhosis should also be true for earlier time points. We have revised the manuscript to make it clear that such enhancement is likely due to the formation of intrahepatic angiogenesis, the existence of portal hypertension that results in the slow washout rate of the contrast agent in addition to the collagen binding (in the “Discussion” section, line 380-384).

Specific Comment 44: *Figure 3f – the existence of portal hypertension in cirrhotic livers is common, and not related to the “injection of ProCA32.collagen1”. I can not piece together how the presence of a contrast agent bound to collagen in liver is indicative to the pressure of blood flow through the portal vein.*

Response: The late stage of fibrosis results in the decrease of the number and size of fenestrations of liver sinusoids in liver (Fig 4e) associated with hypertension which in turn results in slow wash out rate of the reagents in the liver (Fig. 6c, d). The accumulation of high ProCA32.collagen1 in the liver upon injection results in this large enhancement.

Specific Comment 45: *Figure 3a – some areas of “normal” liver seem to have quite high R1 values (similar to early and late fibrosis) at 3hrs. Why is that? What does it indicate?*

Response: The similar small enhancement of R1 value for normal liver and early and late stage fibrosis at 3 hrs post-injection of contrast agent suggests that ProCA32.collagen1 at early time point functions mainly as a blood pool agent. This is consistent with the decrease of signal after 24 hrs time point due to excretion of the contrast agent.

Specific Comment 46: Figure 3c and d – NASH liver (Ishak 5) is reported at R1 ~4s-1 at 3hr (Fig 3d), but images on Fig 3c look less “red” than fibrotic liver (Ishak 5) on Fig 3a. The color scale on Fig 3c also seems off (R1=4s-1 should be very dark red, which isn’t seen on NASH images)

Response: The R1 map images shown in the manuscript in new Fig.4a and 4c are representative images, however the R1 value was measured across the entire liver slices. Furthermore, these two models are very different in terms of collagen distribution difference and morphology of the liver. In Fig. 4c, liver of C57BL/6 CGI-58 knocked out mice were shown which the liver has accumulation of both collagen and fat, however, Fig. 4a demonstrates the BALB/c mice liver with accumulation of only collagen. For better representation of color scale in Fig. 4c, we have replaced that slice with the same corresponding slice in another mice in the same group.

Specific Comment 47: P14.L269 – I don’t follow how the “targeting ability” of the contrast agent reflects the existence of portal hypertension. Is it purely based on slow wash out? Then it’s not really due to the targeting ability.

Response: The reporting of portal hypotension at early time points is largely due to slow washout and has little to do with the targeting ability. We have modified this sentence to clear any confusion (line 247-248).

Specific Comment 48: P14.L270 – there are several claims in this sentence. You may want to be more concise, otherwise it gets confusing. Is it the time-dependent R1 contrast profiles that distinguish fibrosis stage, or delta R1 (and R2) at 24hrs?

Response: Delta R1 and R2 values are used to stage fibrosis as Fig. 6e and 6f and S12 demonstrate. We have modified this sentence as suggested (line 385-387).

Specific Comment 49: P14.L272 – Suggesting the contrast agent reveals patterns in cirrhotic liver may be true, but it’s a qualitative assessment, which may require a separate investigation. It is valid to speculate on the distribution differences in each disease model in the discussion section.

Response: We agree this important point and we have modified several sentences regarding the collagen pattern and discussed collagen distribution differences in the “Discussion” section (line 411-412).

Specific Comment 50: P14.L278 – Do all mice in the DEN group exhibit this difference in CPA (12% vs. 6.7%)? Is this considered mild, moderate, or severe fibrosis?

Response: The purpose of using this model was to show the ability of ProCA32.collagen1 in detecting heterogeneous distribution of collagen in diseased liver. This model is considered to be

severe fibrosis (cirrhosis) and the difference in CPA values have been observed in all slices in all mice.

***Specific Comment 51:** P14.L279 – Why was T1, T2, and T1 IR performed here, and not R1 and R2 mapping (consistent with earlier analysis)?*

Response: We have also performed R1 mapping, Fig. S15a is showing the results for DEN model. T1-, T2-weighted images were used since multiple imaging techniques can further confirm the results and provide more information.

***Specific Comment 52:** Figure 4a – It is not obvious where right and left liver segments are divided. This may make enhancement differences clearer to the reader. T1, T2, and T1 IR are very different imaging techniques; yet the colorscale in figure is identical. Are the intensities comparable between images shown? Please state how scaling was performed or include a colorbar.*

Response: We have provided dashed red line (Fig. 5a) to demonstrate the left and right areas as A1 (left) and A2 (right) in the liver. The intensities between each imaging pulse sequence are not comparable since they are different imaging techniques. We have provided a color bar for each image.

***Specific Comment 53:** Figure 4c – How is “contrast to noise ratio” defined? Both T1 and T2-weighted images seem to show positive signal enhancement. Shouldn’t you see stronger negative signal between segments on T2 using a very high r2 contrast agent?*

Response: Contrast to noise ratio (CNR) in the liver was calculated by choosing a region of interest (ROI) within the liver, estimating the mean intensity within the ROI, and then subtracting it by the mean intensity of muscle. Then this value was divided by the standard deviation (SD) of a region outside the mouse torso. We have defined contrast to noise ratio (CNR) in the “Methods” section (line 661-664). In fibrosis, usually there is an increase in both T1- and T2-weighted images and we did not observe any negative contrast.

***Specific Comment 54:** Figure 4d – need color bar scale for R1. Also should show right vs. left segment R1.*

Response: As suggested, we have included color bar for Fig. S15, however, this image is R1 map and it’s different from images in Fig. 5a, therefore, the enhancement is very different in the entire liver, and as a result, we did not perform any right or left segmentations.

***Specific Comment 55:** P16.L311 – please revise this sentence; it is unclear. There are many uses of the word “or” that makes the claims confusing.*

Response: As suggested, we revised the sentence (line 290).

***Specific Comment 56:** P16.L313 – why was this analysis of enhancement over time only performed with the DEN model, and not other mouse models.*

Response: DEN-induced model showed the heterogeneous distribution of collagen while other two models were relatively homogenous. Thus, we performed in-depth imaging analysis to demonstrate that our developed collagen targeted contrast agent has the capability to capture this important phenomenon mimicking human patients.

***Specific Comment 57:** Figure 4fg – what do the two color bars on one image signify? Are you showing both 3 and 24hrs in one image? Also, the colorbar associated with “maintained” and “washout” is unclear... dark red indicates >100% relative enhancement, yet it’s defined as “maintained”? The color matrix is somewhat unclear to me.*

Response: We superimposed the 3 and 24 hrs intensity maps, therefore we used two different Matlab colormaps (cool for 3 hrs and hot for 24 hrs). The color bar demonstrates the number of voxels that have a specific percentage of relative increase in intensity at both 3 and 24 hrs time points. In the color matrix, the first column shows the enhancement 3 hrs post injection compared to Prescan (called 3 hrs). Then the second column shows the enhancement 24 hrs post injection compared to Prescan (called 24 hrs). Third and fourth columns represent the enhancement 24 hrs post injection compared to 3 hrs. If the voxels remained enhanced it was called “Maintained” and if the voxels had decrease in intensity, they were called “Washout”.

***Specific Comment 58:** P17.L316 – from Fig4c, it seems T2w has higher sensitivity for heterogeneity (difference b/w left-right segments?) than IR. Please define how you are determining sensitivity. How are you quantifying which method (T1 or T2 or IR) is more “sensitive”?*

Response: Based on Fig.5 d and 5e, it is observed that T1 inversion recovery is showing heterogeneity better than T2-wieghted image. This observation is based on the level of enhancement. Based on the color matrix, the number of voxels in T1 inversion recovery that have 100 % < are much higher than T2-wrighted. Approximately 1200 voxels in T1 inversion recovery have 100 % < enhancement, however this number is ~ 600 in T2-weighted image (line 298-300).

Discussion Comments:

***Specific Comment 59:** P17.L323 – What were the data results for correlation with histology, CPA, and Gd3+?*

Response: These data can be found in Fig. 2d, Fig. 3d and Fig. S9.

***Specific Comment 60:** P17.L324 – Is ICP-OES a common term? Not defined anywhere. Was Gd3+ detection at 48hrs?*

Response: ICP-OES stands for “coupled plasma optical emission spectrometry”. We have defined ICP-OES in the manuscript as suggested (line 91-92). Gd³⁺ concentration was detected at 24 hrs post injection.

***Specific Comment 61:** P17.L325 – Is it due to 5x increase in relaxivity, or collagen binding affinity? I would think the latter, since the mechanism seems to be whether the CA is present in liver tissue or not, right?*

Response: The robust detection of early stage fibrosis is due to both collagen-targeting ability and higher relaxivity. As we discussed earlier, upon binding, the relaxivity of the contrast agent can increase even more which leads to even higher sensitivity for early stage detection. ProCA32.collagen1 binds to collagen in the liver, which extends its retention in the liver with much longer half-life in the liver compared with that of blood vessel. Once the contrast agent is injected, liver will be enhanced along with the blood vessels due to the biodistribution of ProCA32.collagen1 in the blood vessel and sinusoid space in the liver. ProCA32.collagen1 gradually binds to collagen in the liver of diseased mice, which extend/elongate the half-time of contrast agent in the liver. No-specifically distributed ProCA32.collagen1 is eliminated and ProCA32.collagen1 remains in the liver regions with overexpression of collagen.

Specific Comment 62: *P17.L326 – IR + long TE may need to be elaborated. Depending on inversion time, IR will enhance T1 differences, while long TE enhances T2 differences.*

Response: We have elaborated short T1 inversion recovery with long TE in the revised manuscript in “Discussion” section (line 360-365). As we demonstrated in Fig. S10, the combination of ProCA32.collagen1 with short T1 inversion recovery with long TE pulse sequence results in higher sensitivity. ProCA32.collagen1 has high accumulation in the liver. As a result, liver has shortened T1, first, inversion time was used to suppress liver signal. Second, since liver also has short T2 (because of r_2 property of the contrast agent), we then use long TE to further suppress the liver signal. As a result, the liver signal is suppressed twice, and collagens will appear as bright signal, since the contrast agent has high r_1 and r_2 , these two steps can be used.

Specific Comment 63: *P17.L327 – It is not clear how the use of enhanced r_2 properties, coupled with various imaging techniques (IR, T1 and T2-weighted) overcome small changes in “liver morphology”*

Response: Please see our response to Comment 60. Clinically approved Gd^{3+} based contrast agents such as Eovist can only apply T1-weighted imaging for in vivo application that has limited sensitivity to detect small changes in liver morphology associated to early stage liver fibrosis. We can combine both r_1 and r_2 properties of ProCA32.collagen1 to increase the detection dynamic range via multiple imaging methodologies as demonstrated in here. In addition, ProCA32.collagen1 provides multiple ways via multiple imaging sequences to observe differential subtle changes in the early stage of diseases.

Specific Comment 64: *P17.L328 – explain how “precision” is “doubled”. Precision of what? Does this mean some coefficient of variation metric is two-times lower by using this agent?*

Response: To avoid any confusion, the sentence was removed from the manuscript. It was meant to show that using multiple imaging sequences provide multiple ways to eliminate MRI artifacts associated with heterogeneous liver background (i.e. false positives) and increase accuracy.

Specific Comment 65: *P17.L333 – Are these statements “likely due to...” an effect? Are they not definite?*

Response: We have modified this sentence to avoid any confusion. There might be several reasons for this high collagen affinity. Addition of a flexible hinge (GGG linker) gives enough freedom to the targeting moiety to bind to collagen (increase binding capacity).

Specific Comment 66: P17.L336 – What is meant by “tissue penetration”? I don’t see correlation values mentioned between CPA and $\Delta R1$ and $\Delta R2$. Is correlation based of visual inspection only?

Response: Collagen proportional area (CPA) values described in Fig. S9 were used to generate Fig. 6e and 6f along with $\Delta R1$ and $\Delta R2$ values. Fig. 2c, Fig. 3c, Fig. S8b, and Fig. S8d, they all demonstrate the correlation between Ishak scores calculated from CPA analysis with $\Delta R1$ and $\Delta R2$ values.

Specific Comment 67: P17.P339 – It is important to precisely define “dynamic molecular imaging (DMI)”. It’s not an intuitive concept. You mention high “spatial and temporal resolution”, but what is considered high for DMI? 0, 3, 24, 48hrs seem very spaced out time points. It will be important to fully discuss how “early” (i.e. +3hrs) time points point to angiogenesis. Where is contrast being “retained” in this scenario vs. normal liver, and why is that indicative to new vessel formation.

Response: Please see our Response to Comment 5. We define this new MRI methodology as dynamic molecular imaging (DMI) to emphasize our novel features. We used the term “dynamic molecular imaging”, simply to convey the point that the contrast agent property is different from other contrast agents, however the word “dynamic” was used because the contrast agent enhancement is different at different time points and has different washout rates based on the stage of the disease (see Fig. 6c, 6d). Furthermore, Fig. 5d and 5e demonstrate this property by showing heterogeneous MRI signal enhancement within 24 h post injection.

Specific Comment 68: P18.L346 – what is SEM? Acronym not defined.

Response: It stands for scanning electron microscopy, we have defined SEM in manuscript (line 380).

Specific Comment 69: P18.L350 – is your claim that a “dramatic” increase in $R2$ and $R2$ at 3hrs indicative of portal hypertension? Why? What increase is considered “dramatic”? Presumably, ProCA32collagen1 concentration in liver is high at 3hrs (vs. normal and early-stage). Doesn’t this mean that contrast agent wash-in and retention (collagen affinity) is greater in cirrhotics vs. early-stage? Do you suspect a more gradual CA uptake in early stage disease? Also, please explain why $R1$ and $R2$ appear to continue increasing in early-stage from 3 to 24 hrs.

Response: $R1$ and $R2$ increase at 3 hrs time point were higher compared to 24 hrs time point mainly due to slow washout originating from decrease in number of fenestrae in liver for late stage of fibrosis as well as collagen binding (Fig 4e, 4f, 6c, 6d). Please also see our Response for Comment 43). The accumulation of contrast agent is predominant for the late stage of fibrosis. However, due to lack of structural changes in fenestrae numbers and size in early stage of fibrosis, we do not have accumulation of protein contrast agent. The slow and gradual increase of $R1$ and

R2 is largely due to the contribution of collagen binding process. The observed enhancement at 24 hours represents the expression level of collagen associated with a disease stage.

***Specific Comment 70:** P18.L352 – This is a one sentence paragraph and seems out of place.*

Response: We have modified and moved this sentence to “Introduction” (line 49-52).

***Specific Comment 71:** P18.L358 – It sounds like you’re distinguishing DCE from DMI. Please explicitly state the difference, since it’s not obvious.*

Response: Dynamic Contrast Enhanced (DCE) imaging measures MRI signal changes such as T1 changes in tissues over time after bolus administration of gadolinium contrast agent. Dynamic refers to the changes as a function of time. It does not imply “short time”. All clinical contrast agents that are based on small molecules (<1 kD in size) have much shorter half-lives (<10 min), and that is why the MR imaging occurs < 1hr post-injection for other MR imaging contrast agents. Their short half-life of clinically approved contrast agents results in limitations in missing the peak signal and large errors in AUC calculation. We did not use the word “DCE” for our contrast agent, and DCE is usually used for clinical contrast agents and shorter time points, but our contrast agent mechanism is clearly different from clinical contrast agents. Since our contrast agent has a half-life of ~9 hours and has collagen binding capability, thus we tailored the acquisition time points to capture the entire curve without missing the peak signal or max R1 time points. ProCA32.collagen1 binds to collagen in the liver, which extends its retention in the liver with much longer half-life in the liver compared with that of blood vessel. Once the contrast agent is injected, liver will be enhanced along with the blood vessels due to the biodistribution of ProCA32.collagen1 in the blood vessel and sinusoid space in the liver. ProCA32.collagen1 gradually binds to collagen in the liver of diseased mice, which extends/elongates the half-time of contrast agent in the liver. Non-specifically distributed ProCA32.collagen1 is eliminated and ProCA32.collagen1 remains in the liver regions with overexpression of collagen.

We defined this new property of the contrast agent as dynamic molecular imaging (DMI) to emphasize our novel features. We used the term “dynamic molecular imaging”, simply to convey the point that the contrast agent property is different from other contrast agents, however the word “dynamic” still can be used because the contrast agent enhancement is different at different time points and has different washout rates based on the stage of the disease (see Fig. 4, 6). Furthermore, Fig. 5d and 5e demonstrate this property by showing heterogeneous MRI signal enhancement within 24 h post injection.

***Specific Comment 72:** P18.L362 – The connection between improved “penetration capability” and the formation of “new vascular structures” is unclear. The agent binds to collagen, so how does this point to vascular information?*

Response: Intrahepatic angiogenesis involves the formation of new vessels and vascular structures with and without complete connection (see Fig. 4g). Such new vascular structural change is packed by overexpress collagen. The significant increase of R1 and R2 at 3-hour time point is a result of accumulation of high concentration of contrast agent due to slow wash out and altered vascular structure as well as collagen binding.

Specific Comment 73: P18.L367 – Please state/disclose these precise pattern differences in the Results, and discuss them here. Are they consistent? Is there a metric to quantitatively determine pattern differences?

Response: We appreciate the recognition of such important finding by the reviewer. As the reviewer pointed out in Comment 47, a qualitative assessment may require a separate investigation and we feel it is beyond the scope of this manuscript. We will report quantitative determination of pattern differences once we establish methodology for such imaging analysis.

Specific Comment 74: P19.L370 – This is a run-on sentence. Also, are you claiming that AUC_0-48 of both R1 and R2 is a robust metric to stage fibrosis? What are the threshold values these need to be to distinguish normal, early and late stage? Are these better than delta R1 and R2 at 24hrs for staging fibrosis?

Response: We have modified this sentence (line 354-355). Fig. 6e and 6f are based on $\Delta R1$ and $\Delta R2$ correlation with CPA. Therefore, $\Delta R1$ and $\Delta R2$ values were used to distinguish early stage fibrosis from cirrhotic liver and normal liver instead of R1 and R2 values. To avoid any confusion, we have modified the labels for these figures. AUC_0-48 of both $\Delta R1$ and $\Delta R2$ can be used to stage fibrosis.

R1 and R2 values were used to show the uptake and washout of the contrast agent (Fig. 6a, b, c, d). Taken together, our reported results using mice models in Fig. 6 suggest the high potential of ProCA32.collagen1 in staging the disease. In the future, we will determine the threshold for differentiation of various stages of fibrosis using large numbers of patient samples with a separate longitudinal study. It is outside the scope of this manuscript.

Specific Comment 75: P19.L376 – Some institutions have migrated from MultiHance to Prohance or Dotarem. Eovist is usually used in specialized cases (low overall usage), and is fairly stable.

Response: Prohance and Dotarem are not liver specific reagents. We have shown that our contrast agent, ProCA32.collagen1 is more selective and stable against competition of endogenous metal ions than several clinically-approved agents including liver specific agent, Eovist (Fig 1c). Eovist is the FDA approved contrast agent with application for liver cirrhosis detection. See reference below:

1. Irene Cruite et al, Gadoxetate Disodium–Enhanced MRI of the Liver: Part 2, Protocol Optimization and Lesion Appearance in the Cirrhotic LiverDOI:10.2214/AJR.10.4538

Methods Comments:

Specific Comment 76: P24L509 – Is CPA calculated over the entire liver, individual slices, or just one segment? In other words, how big was the CPA sample? What was the CPA for the various mouse models?

Response: Collagen Proportional Area (CPA), as determined by the % area stained with Sirius Red, was quantified from the histology images using ImageJ as per standard procedures (line 579-580). The percentage of Sirius red stained areas was calculated based on the entire slide scan. The results are mean values across 10 different slides from different locations of mouse liver (Fig. S9).

Specific Comment 77: P25.L522 – You state 0.02mmol/kg (Line 140). Is 5mM equivalent to that? Perhaps be consistent here.

Response: Yes, 100 µL of 5 mM with mouse weight of 25 g is equivalent to 0.02 mmol/kg. We made sure the injection dosage is consistent throughout the manuscript.

Specific Comment 78: P25.L523 – Please comment on the choice of inversion times. Do you feel there is enough T1 sampling resolution? Your results show that your R1 range is ~1.5 to ~4s-1 (or T1~250ms to 666ms), which means the ideal null point is around 170 to 460ms. Also, were the inversion times acquired in separate acquisitions? Spin echo or gradient echo? What was the scan time? Similar questions for T2 mapping.

Response: Quantitative T1 map images were collected using 8 different inversion recovery times 10, 222, 435, 648, 861, 1074, 1287, and 1500 ms. These IR times have enough sampling resolution and cover the null point and falls in range as compared to liver scanning in animals. The inversion times were not acquired separately; they were part of the T1 map inversion recovery pulse sequence. The sequence is Fast Spin Echo Multi Slice (FSEMS). The scan time was 21 min. T2 map pulse sequence was Multi Echo Multi Slice (MEMS) and the scan time was 8 min. We also selected the Echo Times (TE) in the most efficient way to capture the T2 map difference among different fibrotic livers and normal liver (line 583-599).

Specific Comment 79: P25.L529 – What model was used for curve fitting T1 and T2? Was it performed pixel by pixel?

Response: T1 maps were calculated by mono-exponential recovery fitting of the single TESE signals using a least-square nonlinear algorithm on a pixel-by-pixel basis as follows:

The T1 values are estimated by fitting the curve of signal intensities versus TI. The T1 values determined from the curve fitting are based on the equation $S = S_0 \times (1 - 2 e^{-TI/T1})$, where S represents the signal intensity of the image and S₀ represents the maximum signal intensity at equilibrium condition.

Similarly, T2 maps were computed by mono-exponential fitting of the multi-echo SE signals using a least-square nonlinear algorithm on a pixel-by-pixel basis as follows:
 $SI = SI_0 \times e^{-TE/T2}$.

Specific Comment 80: P25.L534 – What is meant by “intensity MRI images”? Are these the R1 and R2 maps? If so, these are somewhat quantitative, not qualitative.

Response: The histogram analysis was performed for R1 maps (Fig. S14a, S14b, S15b). We corrected this mistake and rephrased those sentences in “Histogram and Voxel Analysis of MRI Images” section (line 600-603).

Specific Comment 81: P25.L536 – *Not sure I follow. Are you saying each image voxel has a min and max R1 value?*

Response: Each image has a maximum and minimum R1 value. We changed the paragraph to “For R1 map histogram analysis, first the R1 value of each voxel was calculated. Then, the minimum and maximum R1 value in each image was determined. The interval between the minimum and maximum value was divided into 500 equal-sized bins. The voxels which were within each bin were counted” for better clarification (line 604-606).

Specific Comment 82: P25.L537 – *Please explain “seeding” over the “interval”. What is the interval?*

Response: Interval here is defined as the range between the minimum R1 and maximum R1 values. Then we divided this range into 500 bins, clusters or categories. We changed “seed” to “bin” for better clarification (line 605).

Specific Comment 83: P26.L542 – *I assume T1, T2, and IR are different acquisitions than R1 and R2 mapping. What were the parameters? So, if I understand, at each time point (0, 3, 24, 48 hrs), T1, T2, IR, R1-mapping, and R2-mapping was acquired? Please state how many scans were performed at each time point, and how long each scan was?*

Response: T1- and T2-weighted pulse sequences were collected to get qualitative information and they are different from spin echo-based inversion recovery (IR) pulse sequence. IR refers to one of the inversion times (TI=10 ms) in T1 inversion recovery which was selected, processed and presented in Fig. 5a. This was not scanned separately but was part of the inversion recovery sequence. The T1-weighted scan time was 2 min. The T2-weighted scan time was 4 min. For T1- and T2 weighted images, T1 and T2 maps, the number of scans was 1 and average=2. All parameters, names of pulse sequences, number of scans and duration, have been added to “Methods” section, “MRI Scan” (line 583-599).

Specific Comment 84: P26.L544 – *awkward sentence (“To make sure the enhanced area remained enhanced or the contrast agent is washed out...”)*

Response: As suggested, we rephrased this sentence to be “To identify areas that remained enhanced or the contrast agent is washed out, another variable was defined” (line 610-611).

Specific Comment 85: P26.L547 – *Not sure if you can conclude this; it depends on imaging technique. For T2-weighting, more negative contrast may indicate more T2 effect, hence greater agent concentration. A negative T1 contrast may also indicate increase concentration, since a T2 effect dominates. I can presume you may not have to worry about the latter case, since delta R2 were not extremely high, but you may want to acknowledge that this is possible.*

Response: We believe both T1- and T2-weighted imaging techniques will have enhancement upon injection of the contrast agent, therefore they both have positive effects post-injection, as observed in Fig. 5a. Collagen binding of the contrast agent will cause this positive contrast in both T1- and T2-weighted imaging. Then, any negative values will be an indication of contrast agent washout. When comparing 3 and 24 hrs time points with Prescan, we observed either enhancement in certain regions or no enhancement within 3 and 24 hrs. The negative value that we described here means that the intensity was lower at 24 hrs time point compared to 3 hrs time point.

Specific Comment 86: P26.L549 – It’s ok to formulate a unique analysis like this. But since it’s challenging for readers to grasp initially, please summarize the overall results concisely (in the Results). What significance does it have?

Response: We have summarized the results as suggested in the “Results” section (line 290-300). Detection of collagen heterogeneity with this analysis can facilitate biopsy procedure and reduce complications associated with this invasive technique. In addition, it can provide information regarding the regions of the cirrhotic liver that are likely to develop HCC.

Specific Comment 87: P26.L560 – relaxivities, not “relaxation rates”. “GdCl₃ and protein (2:1)”: how does this mimic ProCA32collagen in vivo?

Response: We corrected the mistake and changed it to “relaxivities”. The protein has two Gd³⁺ binding sites, gadolinium was loaded in 2:1 ratio. This ratio was used for all *in vitro* and *in vivo* experiments. ProCA32.collagen1 relaxivity is expected to increase due to collagen binding based on the influence of binding on rotational correlation time (τ_R).

Specific Comment 88: P26.L563 – How were Eovist and ProCA32collagen dilutions prepared for ex vivo measurements? In saline, plasma, serum? How was it different than the 1.4T preparation?

Response: 5 mM of both Eovist and ProCA32.collagen1 were prepared in 10 mM HEPES buffer with pH of 7.2 for both *ex vivo* and *in vivo* experiments.

Specific Comment 89: P27.L577 – euthanasia at 48hr post-injection?

Response: It was performed at 24 hrs post injection.

P27.L583 – this stats section only deals with AUC analysis. Please re-state section heading.

Response: We modified the section heading to “AUC Analysis” as suggested (line 648).

Specific Comment 90: P27.L584 – PSE was already defined and stated earlier in Methods (line 543).

Response: We have defined Percentage of R1 and R2 enhancement in “Methods” section in “AUC Analysis” section (line 647).

Specific Comment 91: P27.L586 – why only NASH model? How was AUC measured... trapezoidal rule?

Response: We used R package "pROC" to calculate AUC by trapezoidal rule. The same analysis was performed for TAA/Alcohol model as well (See Fig. S12), to show the ability of contrast agent to distinguish different stages of the disease, and the AUROC was calculated. Fig. S12 is another method of representing AUC. Fig. 4b and 4d show the same pattern for uptake and washout of the contrast agent, therefore, only NASH model was presented.

Specific Comment 92: P27.L587 – why wouldn't AUC_3-24, slope_3-24, or slope_0-3 show differences too? Were all possibilities tested to determine the appropriate metric?

Response: They all show the difference; however, we believe this analysis is the best representation of the effect of the contrast agent over 48 hrs time point (Fig. 6).

Specific Comment 93: P28.L591 – Most of the statements in this section are results, not methods.

Response: We have modified this section as suggested to just show the methods and not any results (line 647-656).

Specific Comment 94: P28.L598 – Were the values from an ROI of the entire liver? Please state ROI size and any segmentation routines (automatic or manual).

Response: The ROI was from the entire liver in all 12 image slices and a manual segmentation was performed.

Specific Comment 95: P28.L600 – some analysis was paired, correct? Such as RI time course within the same animal model.

Response: Yes, Fig. 4b and 4d data are paired.

Specific Comment 96: P28.L603 – why weren't the pharmacokinetic results included in the main document?

Response: We have included the pharmacokinetic data in the main text as suggested in Fig. 1f and 1g.

Specific Comment 97: P28.L605 – how many sampling points? What is meant by n=3-6?

Response: There were six mice in each sampling point (n=6).

Supplemental Figures Comments:

Specific Comment 98: Figure S4 – a. show colorbar scale on image; c. which tissues/values represent CNR calculation? Normal vs. cirrhotic? Left vs. right? Which ROIs?

Response: We have removed this Figure from the manuscript, as it was repetitive data. Fig. S15 (R1 map) shows the same data and we have added a color bar for this figure.

***Specific Comment 99:** Figure S6 – Is clearance via the kidney? Also consider comparing Table c with current clinical agents from literature. I suggest including pharmacokinetic results in the main document.*

Response: We believe the main clearance is through kidney, however there is some liver clearance as well. We have added pharmacokinetic data into the main text, in Fig. 1f and 1g. There is not much information available regarding the pharmacokinetic parameters of clinical contrast agents in mice, as the majority of the data available in the literature are for application in human, therefore comparison with human data was not possible. Elimination half-life of clinical contrast agents in mice varies from 30-50 min.

REVIEWERS' COMMENTS:

Reviewer #2 (Remarks to the Author):

Most of my comments have been addressed, and the manuscript has been greatly improved. However, the authors need to include the fibrosis regression data in the manuscript, as this is a clinically very relevant condition.

Reviewer #4 (Remarks to the Author):

Reviewer #1 asked for a complete description of the protein including animal or human source. The animal or human source was not provide on line 78 as stated by authors in the response.

Fig two caption line 173. Change map to maps. Line 175 “ from R1 map” to “a R1 map”. The SI abbreviation for hours is “h” not hrs” The alternative abbreviation is hr, regardless of the number of hours.

Other Reviewer 1 comments are all well addressed.

Reviewer 2.

The authors' response to comments 1 and 2 are thoughtful and correct, in my opinion.

Reviewer 2's idea that further studies are needed to support translation are asking too much of a preclinical paper of such breadth and depth. Such studies would fill another paper. Authors' addition of the histology (Fig 7C) are reasonable.

Reviewer 3.

I don't fully agree with Reviewer 3's comment on limiting the data and discussion in this preclinical paper. The authors addressed the comment in revisions to a reasonable extent. Refinement to a clinical protocol is suggested by authors, even if it is not proven that their suggestions will be the likely clinically adopted protocols. Clinical outcomes research will certainly follow.

Comment 5. Regarding the “dynamic” definition and arguments around using “dynamic “ to create a name, DMI, the key concept for differentiating dynamic as a word is “continuous” so the Reviewer 3 is correct. Moreover, clinicians that use Gd agents in DCE MRI already use “dynamic” to mean a rapidly acquired continuous SI change in voxels whose several parameters (height of curve etc.) are diagnostic of disease states. “Serial MRI” is what authors performed, not “dynamic” MRI. In my opinion, “DMI” coined by the authors will not decrease confusion but add to it in the minds of clinical users. No new term is required, but if authors see a need for a new term to describe an

element of novelty in how they interpret the changes in SI with time for their new agents, then SMI could be used. It is just serial MRI scans, that are often used with or without added Gd agents to document disease processes, and here document a Gd agent's changing effect on relaxation times and MRI images over hours and days.

Comment 6. The reviewer is correct. All Supplemental data should be referred to somewhere in the text.

Comment 23. MRI cannot really differentiate retention of the Gd agent in the blood, sinusoids, extracellular space or bound to collagen or anything else. It just measures the average relaxation time in a voxel (and some other things) and that T1 and T2, are known to be affected by the presence of Gd (anywhere in the voxel where water is exchangeable on a microsecond time scale). Hence the authors' speculation about the dynamic intra-tissue compartment biodistribution is just that. In fact all comments by reviewers and authors that rely on the assumption that in vivo MRI signal and signal changes are representative of [Gd] are unproven and inherently not expected to be linear as SI vs [Gd], especially in a protein agent. This does not mean that changes in MRI signal are not useful in diagnosis nor clearly related in some way to the dynamics of contrast agent concentration changes, just that quantitatively, one cannot know that the relationship is constant.

Comment 25. Authors response is not unreasonable and some time points simply have to be chosen as starting points, but the only way to know where the peak signal is (not where peak [Gd] is) is to measure signal vs time in small increments relative to the overall area under the time- signal curve.

The comments and responses around mechanism necessarily are speculative on the part of the authors. It is fine to speculate, as long as it is acknowledged. I think some of reviewers' questions are stimulated by the apparent confidence of authors in the truth of their hypotheses, Fig 7.

Discussion. The whole discussion of NSF etc. is mute because the macrocyclic agents already are dominating the Gd agent markets due to their lack of NSF cases. Authors can make a single sentence statement that ProCa obviously passes this test with stability in serum equivalent to the macrocyclic agents. Same for brain deposition, maybe in the same sentence.

I recommend publication with due consideration to the above comments. Reviewers concerns have been adequately addressed (except the first coment above of Reviewer 1, which is easily fixed).

Reviewer #5 (Remarks to the Author):

The report on this interesting new contrast media which was named ProCA32collagen1 by your group is very informative. Several aspects (physicochemical, relaxivity) are mixed with interesting

correlation studies with other contrast media, histopathology as well as the interpretation of experimental MR imaging data to ensure that the potential use of this contrast media will be underlined. Congratulation to your work on this interesting field.

REVIEWERS' COMMENTS:

All the changes in the text based on Reviewers comments are highlighted in yellow.

Reviewer #2 (Remarks to the Author):

Most of my comments have been addressed, and the manuscript has been greatly improved. However, the authors need to include the fibrosis regression data in the manuscript, as this is a clinically very relevant condition.

Response: Based on Reviewer 2 comment, liver fibrosis treatment data have been added to the manuscript (Figure S17).

Reviewer #4 (Remarks to the Author):

Reviewer #1 asked for a complete description of the protein including animal or human source. The animal or human source was not provide on line 78 as stated by authors in the response.

Response: The protein based on rat α -parvalbumin was expressed and purified in bacteria (*Escherichia coli*). We have mentioned the source of the protein in the manuscript in the "Introduction" section (highlighted in yellow).

Fig two caption line 173. Change map to maps. Line 175 " from R1 map" to "a R1 map". The SI abbreviation for hours is "h" not hrs" The alternative abbreviation is hr, regardless of the number of hours.

Response: Mentioned changes are made to the manuscript according to the comment.

Other Reviewer 1 comments are all well addressed.

Reviewer 2. The authors' response to comments 1 and 2 are thoughtful and correct, in my opinion.

Reviewer 2's idea that further studies are needed to support translation are asking too much of a preclinical paper of such breadth and depth. Such studies would fill another paper. Authors' addition of the histology (Fig 7C) are reasonable.

Reviewer 3.

I don't fully agree with Reviewer 3's comment on limiting the data and discussion in this preclinical paper. The authors addressed the comment in revisions to a reasonable extent. Refinement to a clinical protocol is suggested by authors, even if it is not proven that their suggestions will be the likely clinically adopted protocols. Clinical outcomes research will certainly follow.

Comment 5. Regarding the "dynamic" definition and arguments around using "dynamic " to create a a name, DMI, the key concept for differentiating dynamic as a word is "continuous" so the Reviewer 3 is correct. Moreover, clinicians that use Gd agents in DCE MRI already use "dynamic" to mean a rapidly acquired continuous SI change in voxels whose several parameters (height of curve etc.) are diagnostic of disease states. "Serial MRI" is what authors performed, not "dynamic" MRI. In my opinion, "DMI" coined by the authors will not decrease confusion but add to it in the minds of clinical users. No new term is required, but if authors see a need for a new term to describe an element of novelty in how they interpret the changes in SI with time for their new

agents, then SMI could be used. It is just serial MRI scans, that are often used with or without added Gd agents to document disease processes, and here document a Gd agent's changing effect on relaxation times and MRI images over hours and days.

Response: Based on the comment, we have removed “dynamic” from the text and replaced it with “serial molecular imaging (SMI)” where applicable.

Comment 6. The reviewer is correct. All Supplemental data should be referred to somewhere in the text.

Response: All Supplemental data have been referred to in the text where appropriate.

Comment 23. MRI cannot really differentiate retention of the Gd agent in the blood, sinusoids, extracellular space or bound to collagen or anything else. It just measures the average relaxation time in a voxel (and some other things) and that T1 and T2, are known to be affected by the presence of Gd (anywhere in the voxel where water is exchangeable on a microsecond time scale). Hence the authors' speculation about the dynamic intra-tissue compartment biodistribution is just that. In fact all comments by reviewers and authors that rely on the assumption that in vivo MRI signal and signal changes are representative of [Gd] are unproven and inherently not expected to be linear as SI vs [Gd], especially in a protein agent. This does not mean that changes in MRI signal are not useful in diagnosis nor clearly related in some way to the dynamics of contrast agent concentration changes, just that quantitatively, one cannot know that the relationship is constant.

Response: According to the comment, we have removed some of the speculations made in the manuscript for contrast agent retention.

Comment 25. Authors response is not unreasonable and some time points simply have to be chosen as starting points, but the only way to know where the peak signal is (not where peak [Gd] is) is to measure signal vs time in small increments relative to the overall area under the time-signal curve. The comments and responses around mechanism necessarily are speculative on the part of the authors. It is fine to speculate, as long as it is acknowledged. I think some of reviewers' questions are stimulated by the apparent confidence of authors in the truth of their hypotheses

Response: We have acknowledged that some statements made in the manuscript regarding the mechanism of the contrast agent are speculations.

Fig 7. Discussion. The whole discussion of NSF etc. is mute because the macrocyclic agents already are dominating the Gd agent markets due to their lack of NSF cases. Authors can make a single sentence statement that ProCa obviously passes this test with stability in serum equivalent to the macrocyclic agents. Same for brain deposition, maybe in the same sentence. I recommend publication with due consideration to the above comments. Reviewers concerns have been adequately addressed (except the first coment above of Reviewer 1, which is easily fixed).

Response: According to the comment, we have modified statements regarding NSF. We added a short paragraph instead in “Discussion” section. The paragraph can be found bellow:
“Our developed ProCA32.collagen1 have addressed concerns about metal toxicity and Gd³⁺ deposition associated with liver linear contrast agents such as Eovist and MultiHance by exhibiting reduced dose due to higher relaxivity and strong resistance against transmetallation. Our studies

demonstrated no Gd³⁺ deposition in brain associated with ProCA32.collagen1 (Fig. S6d) despite some recent reports of potential brain deposition in some clinical contrast agents”

Reviewer #5 (Remarks to the Author):

The report on this interesting new contrast media which was named ProCA32collagen1 by your group is very informative. Several aspects (physicochemical, relaxivity) are mixed with interesting correlation studies with other contrast media, histopathology as well as the interpretation of experimental MR imaging data to ensure that the potential use of this contrast media will be underlined. Congratulation to your work on this interesting field.

Response: Thank you.